# STING induces ZBP1-mediated necroptosis independently of TNFR1 and FADD

Konstantinos Kelepouras[1,2,3], Julia Saggau[1,2,3,4,5,11], Debora Bonasera[1,2,3,4,5,11], Christine Kiefer[1,2,3], Federica Locci[6], Hassan Rakhsh-Khorshid[1,2,3], Louisa Grauvogel[7], Ana Beatriz Varanda[4,5], Martin Peifer[7], Elena Loricchio[8], Antonella Montinaro[9], Marijana Croon[3,10], Aleksandra Trifunovic[3,10], Giusi Prencipe[8], Antonella Insalaco[8], Fabrizio De Benedetti[8], Henning Walczak[4,5,9] & Gianmaria Liccardi[1,2,3]✉

Conditional deletion of caspase-8 in mouse epidermal keratinocytes ($Casp8^{E-KO}$) causes necroptosis-driven lethal dermatitis[1–7]. Here we find that the loss of $Casp8$ leads to an accumulation of cytosolic DNA that is responsible for the activation of a cyclic GMP-AMP synthase (cGAS)–stimulator of interferon genes (STING)-mediated transcriptional program. Genetic and biochemical evidence indicate that STING upregulates both Z-DNA-binding protein 1 (ZBP1) and mixed lineage kinase domain-like pseudokinase. Combined caspase-8-deficiency- and STING-activation-driven accumulation of Z-nucleic acids activates ZBP1 and triggers the formation of a ZBP1–RIPK1–RIPK3 complex independently of the FADD–RIPK1–RIPK3 complex, enabling execution of necroptosis. Genetically, we reveal a functional overlap between STING and ZBP1 as drivers of lethal dermatitis independently of tumour necrosis factor receptor 1 (TNFR1), identifying an aetiology of necroptotic inflammation. As gain-of-function mutations in human STING cause STING-associated vasculopathy with onset in infancy (SAVI), we assessed the role of STING-induced necroptosis in SAVI's aetiology. Chronic activation of STING in patients orchestrates a necroptotic transcriptional program that is confirmed in the $Sting1^{N153S}$ SAVI preclinical mouse model in which immune-cell-driven pathology and lethality are rescued by receptor-interacting serine/threonine-protein kinase 3 ($Ripk3$) co-deletion. These findings establish STING-driven ZBP1-mediated necroptosis as a central pathogenic mechanism in both caspase-8-deficient inflammation and SAVI and suggest that targeting the ZBP1–RIPK3–MLKL axis holds therapeutic potential for interferonopathies characterized by excessive necroptosis.

In contrast to deletion of $Ripk3$ or $Mlkl$, loss of $Tnfr1$ does not prevent, and only delays, the embryonic lethality and the lethal dermatitis of $Casp8^{-/-}$ and $Casp8^{E-KO}$ mice, respectively[1,8,9]. Thus, RIPK3–MLKL-induced signalling beyond TNFR1-mediated necroptosis is responsible for aberrantly activated necroptosis. Similarly, removal of $Tnfr1$ only minimally delays the observed lethal dermatitis induced by loss of receptor-interacting serine/threonine-protein kinase 1 ($Ripk1$) or FAS-associated death domain protein ($Fadd$) in epidermal keratinocytes ($Ripk1^{E-KO}$ or $Fadd^{E-KO}$)[10–12]. Notably, $Ripk1^{E-KO}$ mice can survive through adulthood already with single co-deletion of $Zbp1$, and $Fadd^{E-KO}$ mice are completely rescued by co-deletion of $Zbp1$ and $Tnfr1$, suggesting ZBP1-dependent but TNFR1-independent necroptosis involvement[7,13,14].

Although it was shown that activation of ZBP1 depends on its binding to cytosolic Z-DNA or Z-RNA[15–20], the pathway responsible for ZBP1 upregulation and its consequent role in necroptosis remained unclear.

Previous studies have indicated that the loss of caspase-8 induces upregulation of ZBP1 in mouse embryonic fibroblasts (MEFs)[21]. Histological examination of skin obtained from $Casp8^{-/-}$ mice crossed with either RIPK3- or MLKL-deficient mice ($Casp8^{-/-}Ripk3^{-/-}$ and $Casp8^{-/-}Mlkl^{-/-}$) revealed a significant upregulation of ZBP1 and signal transducer and activator of transcription 1 (STAT1) phosphorylation (p-STAT1) also in vivo (Fig. 1a). ZBP1 upregulation was indeed detectable despite the absence of any visible necroptosis-induced skin lesions or histologically detectable immune cell infiltration (Fig. 1a).

[1]Genome Instability, Inflammation and Cell Death Laboratory, Institute of Biochemistry I, Centre for Biochemistry, Faculty of Medicine, University of Cologne, Cologne, Germany. [2]Centre for Molecular Medicine Cologne (CMMC), University of Cologne, Cologne, Germany. [3]Cologne Excellence Cluster on Cellular Stress Responses in Ageing-Associated Diseases (CECAD), Medical Faculty, University of Cologne, Cologne, Germany. [4]Cell Death, Inflammation and Immunity Laboratory, CECAD Cluster of Excellence, University of Cologne, Cologne, Germany. [5]Cell Death, Inflammation and Immunity Laboratory, Institute of Biochemistry I, Centre for Biochemistry, Faculty of Medicine, University of Cologne, Cologne, Germany. [6]Department of Plant–Microbe Interactions, Max Planck Institute for Plant Breeding Research, Cologne, Germany. [7]Department of Translational Genomics, Centre of Integrated Oncology Cologne-Bonn, Medical Faculty, University of Cologne, Cologne, Germany. [8]Division of Rheumatology, Ospedale Pediatrico Bambino Gesù, IRCCS, Rome, Italy. [9]Centre for Cell Death, Cancer and Inflammation, UCL Cancer Institute, University College London, London, UK. [10]Institute for Mitochondrial Diseases and Ageing, Medical Faculty and Centre for Molecular Medicine Cologne (CMMC), University of Cologne, Cologne, Germany. [11]These authors contributed equally: Julia Saggau, Debora Bonasera. ✉e-mail: gianmaria.liccardi@uk-koeln.de

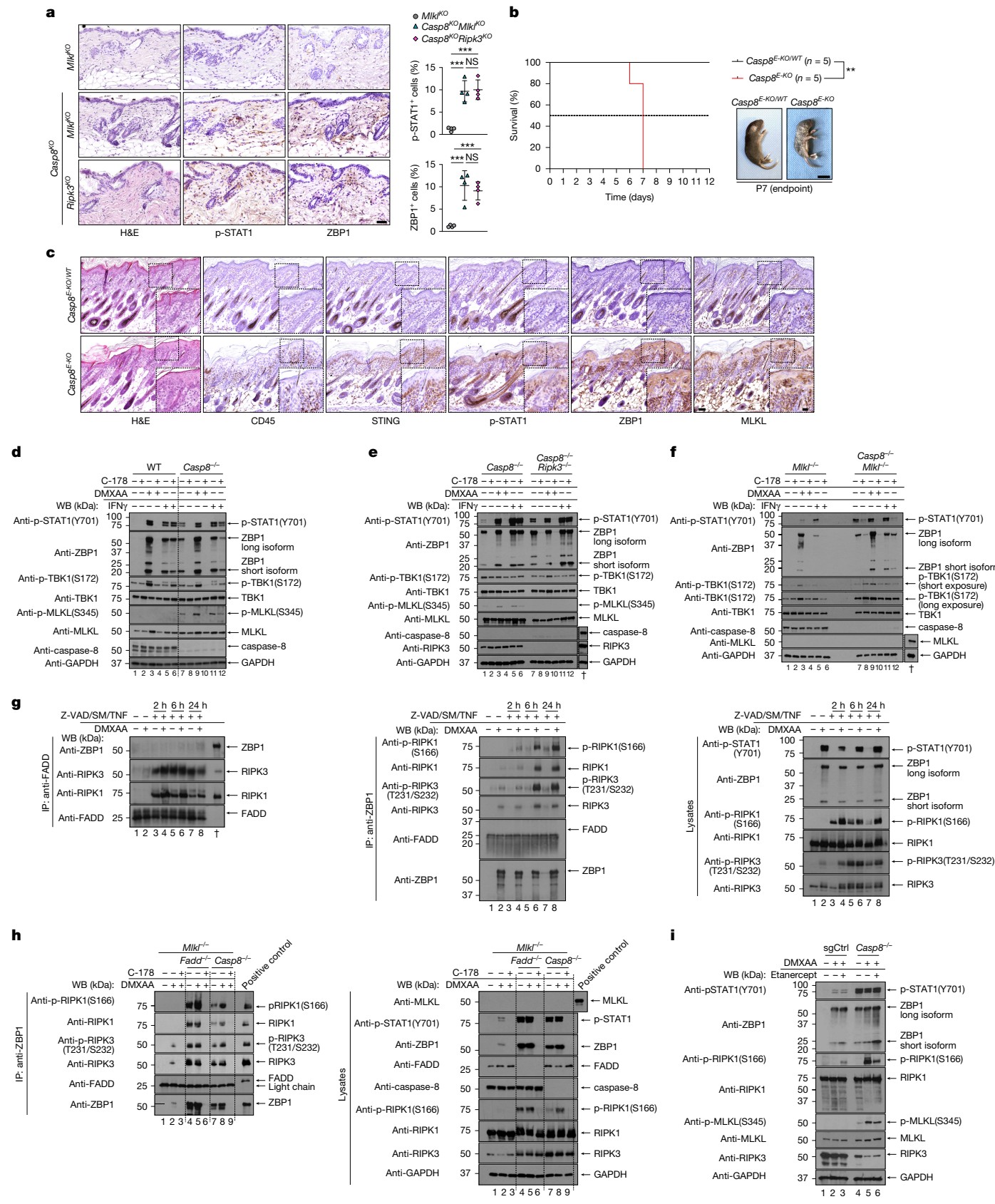

**Fig. 1 | See next page for caption.**

This suggested that caspase-8 prevents the sustained tonic activation of interferon (IFN) signalling, probably responsible for driving ZBP1 expression independently of cell death.

Given that *Zbp1* deletion is required to neutralize aberrantly activated necroptosis[14–16,20,22–24], we hypothesized that the activation of this IFN-mediated transcriptional response would have

**Fig. 1 | STING induces ZBP1–MLKL upregulation and ZBP1–RIPK1–RIPK3 complex formation. a**, Representative images of consecutive skin sections from 12-week-old mice of the indicated genotypes stained with haematoxylin and eosin (H&E) and the indicated antibodies. $n = 4$ per group. Scale bar, 50 μm. Immunostaining quantification is shown. Data are mean ± s.e.m. Each dot represents one mouse. $P$ values were calculated using one-way analysis of variance (ANOVA) followed by Tukey's multiple-comparison test. **b**, Kaplan–Meier survival graph of mice of the indicated genotypes with representative images. $n = 5$ per group. Scale bars, 1 cm. $P$ values were calculated using two-sided Gehan–Breslow–Wilcoxon tests. **c**, Representative images of consecutive skin sections from P5 mice of the indicated genotypes stained with the indicated antibodies. $n = 5$ per group. Scale bars, 50 μm (main images) and 20 μm (insets). **d–f**, Immunoblot analysis with the indicated antibodies of lysates from immortalized WT and $Casp8^{-/-}$ (**d**), $Casp8^{-/-}$ and $Casp8^{-/-}Ripk3^{-/-}$ (**e**), and $Mlkl^{-/-}$ and $Casp8^{-/-}Mlkl^{-/-}$ (**f**) MEFs treated with C-178 (10 μM), DMXAA (20 μg ml$^{-1}$) and IFNγ (1,000 U ml$^{-1}$) for 48 h. The dagger symbols indicate lysates from WT (**e**) or $Casp8^{-/-}$ (**f**) MEFs. **g**, Lysates from immortalized $Mlkl^{-/-}$ MEFs pretreated with DMXAA overnight, followed by treatment with recombinant TNF (100 ng ml$^{-1}$), Z-VAD-FMK (20 μM) and BV6 (2 μM) for 48 h were immunoprecipitated with anti-FADD and anti-ZBP1 antibodies. An immunoblot analysis using the indicated antibodies is shown. SM, Smac mimetics. **h**, Lysates from immortalized $Mlkl^{-/-}$, $Fadd^{-/-}Mlkl^{-/-}$ and $Casp8^{-/-}Mlkl^{-/-}$ MEFs treated with DMXAA and/or C-178 for 48 h were immunoprecipitated with anti-ZBP1 antibodies. An immunoblot analysis using the indicated antibodies is shown. **i**, Immunoblot analysis with the indicated antibodies of lysates from $Casp8^{-/-}$ and sgCtrl MEFs treated with DMXAA and/or etanercept (50 μg ml$^{-1}$) for 48 h. Representative immunoblot data are shown from two independent experiments. **$P \leq 0.01$, ***$P \leq 0.001$; NS, not significant ($P > 0.05$).

a pathological role in regulating an arm of inflammatory necroptosis. We therefore aimed to identify the critical pathway component responsible for this IFN-mediated transcriptional ZBP1 upregulation. By doing so, we would establish a mechanistic link between necroptosis activation and the clinical manifestations of interferonopathies specifically driven by untoward activation of this upstream regulator.

## STING orchestrates necroptosis through ZBP1

To address this question, we generated $Casp8^{E-KO}$ mice (Fig. 1b). As expected, these mice succumbed to keratinocyte necroptosis-driven dermatitis (Fig. 1b and Extended Data Fig. 1a,b) and systemic inflammation (Extended Data Fig. 1c–f) by postnatal day 7 (P7)[1]. Transcriptional analysis of P5 $Casp8^{E-KO}$ lesion-containing skin using bulk 3′ mRNA sequencing (RNA-seq) identified the upregulation of 1,808 genes, including $Mlkl$, $Zbp1$ and, notably, numerous other IFN-stimulated genes (ISGs) (Extended Data Fig. 1g,h). Functional grouping associated these genes with inflammation, cell death and IFN response pathways (Extended Data Fig. 1i–n). Unexpectedly, we observed that the IFN response signature was associated with innate immune signalling, cytosolic DNA sensing and, specifically, the STING signalling pathway (Extended Data Fig. 1i–n). Histological examination of $Casp8^{E-KO}$ skin at P5 confirmed an expected infiltration of CD45$^+$ immune cells in the dermis in response to keratinocyte necroptosis; however, it also revealed elevated STING levels in epidermal cells which were causative of necroptotic cell death (Fig. 1c). Notably, these same cells also exhibited increased p-STAT1, MLKL and, decisively, de novo upregulation of ZBP1 (Fig. 1c). We therefore hypothesized that caspase-8 deficiency drives STING-mediated IFN response in keratinocytes that, in turn, was responsible for ZBP1 and MLKL upregulation, thereby priming these cells to necroptosis.

To test this hypothesis, we used caspase-8-deficient, caspase-8 and RIPK3-deficient and caspase-8 and MLKL-deficient MEFs. As previously reported[21], caspase-8 deficiency resulted in endogenous upregulation of ZBP1 that was accompanied by basal activation of p-STAT1, confirming activation of the IFN response (Fig. 1d–f). Notably, treatment of caspase-8-deficient MEFs with the mouse STING antagonist C-178 (ref. 25) completely abolished the activation of STAT1 and the increased expression of ZBP1 (Fig. 1d–f), thereby providing a functional link between STING activation induced by caspase-8 deficiency and IFN-response signalling. Consistently, treatment with the mouse STING agonist (DMXAA) or IFNγ enhanced the basal level of ZBP1, which was completely abolished or slightly reduced, respectively, by co-treatment with C-178 or co-deletion of $Sting1$ (Fig. 1d–f and Extended Data Fig. 2a). Thus, these data suggest that the loss of caspase-8 leads to aberrant STING activation, which drives IFN-dependent transcriptional upregulation of both ZBP1 and MLKL and therefore necroptosis execution.

## STING induces ZBP1–RIPK1–RIPK3 complex

As DMXAA treatment could also lead to STING-dependent phosphorylation of MLKL (Fig. 1d,e), we next investigated STING-mediated sensitization to necroptosis in WT, $Casp8^{-/-}$, $Ripk3^{-/-}$, $Mlkl^{-/-}$, $Casp8^{-/-}Ripk3^{-/-}$, $Casp8^{-/-}Mlkl^{-/-}$, $Zbp1^{-/-}$ and $Casp8^{-/-}Zbp1^{-/-}$ MEFs, primary cells, and human HT29 and HaCaT cell lines. STING agonism (ADU-S100 for human cells) enhanced TNF-induced cell death, which was completely prevented in MLKL- or RIPK3-deficient or -inhibited cells, confirming that STING activation predominantly sensitizes cells to necroptosis (Extended Data Fig. 2b–f). However, $Zbp1^{-/-}$ cells still underwent TNF-induced necroptosis but were unresponsive to STING-activation-induced necroptosis (STAIN), indicating that ZBP1 is essential for enhanced cell death induction by STING (Extended Data Fig. 2b,e). STING-agonist-induced sensitization was further exacerbated by $Casp8$ deletion alone (Extended Data Fig. 2c,e); however, co-treatment with the TNF blocker etanercept only partially inhibited this sensitization (Extended Data Fig. 2g). Complete inhibition was achieved only after $Zbp1$ co-deletion, confirming that STING-induced necroptosis is dependent on ZBP1 expression (Extended Data Fig. 2h). Together, these results suggested also a STING-dependent TNFR1 activation alongside a TNFR1-independent pathway of STAIN through ZBP1.

Investigation of complex-II and necrosome formation revealed the formation of a FADD–RIPK1–RIPK3 complex after treatment with SM, Z-VAD and TNF, with only a modest increase in RIPK1 and RIPK3 recruitment after DMXAA co-treatment (Fig. 1g). Notably, despite input lysates showing significantly elevated levels of p-RIPK1 and p-RIPK3 after DMXAA co-treatment, this did not correspond to an increased recruitment of the activated RIP kinases to FADD. Similarly, although ZBP1 was upregulated in a STING-dependent manner, it was not detected in FADD pull-downs (Fig. 1g). Notably, endogenous ZBP1 pull-down demonstrated that STING-induced upregulation of ZBP1 resulted in the formation of a FADD-independent complex, which was also formed even after DMXAA treatment alone (Fig. 1g). Consistently, in $Casp8^{-/-}Mlkl^{-/-}$ and $Fadd^{-/-}Mlkl^{-/-}$ MEFs, we detected constitutive formation of the ZBP1–RIPK1–RIPK3 complex under basal conditions, correlating with sustained endogenous ZBP1 expression (Fig. 1h). After DMXAA treatment, both the recruitment and phosphorylation of RIPK1 and RIPK3 within this complex were markedly enhanced, confirming that STING activation amplifies ZBP1-driven complex formation and kinase activation even in the absence of FADD (Fig. 1h). Importantly, co-treatment with C-178 potently suppressed ZBP1 expression and consequently ZBP1-complex assembly (Fig. 1h). These findings establish that STING signalling is sufficient to drive the formation of a FADD- and TNFR1-independent ZBP1–RIPK1–RIPK3 platform, capable of initiating necroptosis in the absence of additional inflammatory inputs. Consistently, co-treatment with the TNF inhibitor etanercept did not diminish p-MLKL in caspase-8-deficient MEFs, and the levels of p-RIPK1 were only partially affected after co-treatment with DMXAA (Fig. 1i).

Collectively, these results identify a pathway of STAIN mediated by a ZBP1–RIPK1–RIPK3-containing complex that forms independently of the TNF-induced FADD–RIPK1–RIPK3 complex-II, while also contributing to autocrine TNF release.

## STAIN occurs independently of TNFR1

To investigate STAIN in vivo, we generated *Sting1*[E-KO] mice, which were viable and fertile (Supplementary Fig. 2), and *Casp8*[E-KO]*Sting1*[E-KO] mice. Moreover, we generated *Casp8*[E-KO]*Tnfr1*[−/−] mice, which, contrary to published reports[1], succumbed to lethal dermatitis at P10 (Fig. 2a). Notably, *Casp8*[E-KO]*Sting1*[E-KO] mice survived until P11, demonstrating a clear similarity and a small, albeit significant, survival advantage to *Tnfr1* full-body co-deletion (Fig. 2a and Supplementary Fig. 2).

Macroscopically, both *Casp8*[E-KO]*Tnfr1*[−/−] and *Casp8*[E-KO]*Sting1*[E-KO] mice maintained a normal body weight and exhibited no or significantly fewer lesions at P3 and P5, respectively, compared with *Casp8*[E-KO] mice (Fig. 2b). This was corroborated by a significant decrease in skin inflammation (keratin-6 (K6), K10, K14) (Fig. 2c) and by the rescue of organ inflammation (Extended Data Fig. 3a–c). Immunohistochemical analysis of P3 skin sections confirmed the observed delay in necroptosis activation, evidenced by diminished numbers of MLKL phosphorylated at Ser345 (p-MLKL(Ser345))[26], and ZBP1-positive epidermal keratinocytes, particularly in the skin of *Casp8*[E-KO]*Sting1*[E-KO] mice (Fig. 2d). Similarly, skin sections from P5 mice showed clear reductions in p-MLKL(Ser345)[26], cleaved CASP3 (Cl-CASP3) and TUNEL staining in both genotypes (Fig. 2e). Loss of STING in keratinocytes coincided with the absence of p-STAT1 and ZBP1 in the epidermis, whereas, in *Casp8*[E-KO]*Tnfr1*[−/−] mice, staining for these markers was partially reduced in both stromal and immune cells (Fig. 2f). This indicated a tissue-compartment-specific difference in early necroptosis inhibition between keratinocyte-specific STING and constitutive TNFR1 absence.

We next assessed the following three manifestations of inflammation in *Casp8*[E-KO]*Tnfr1*[−/−] and *Casp8*[E-KO]*Sting1*[E-KO] mice: (1) immune cell infiltration; (2) cytokine profiles from whole skin homogenates; and (3) differential gene expression using quantitative PCR with reverse transcription (RT–qPCR) and RNA-seq. No major differences were observed in immune cell recruitment with the exception of an enhanced infiltration of CD3[+] T cells in both genotypes, albeit more significantly after *Sting1* co-deletion in keratinocytes, and a significant reduction in F4/80[+] macrophages in *Casp8*[E-KO]*Tnfr1*[−/−] mice (Extended Data Fig. 3d). The levels of most cytokines were similarly lower in skin lysates from both genotypes with only a few showing significant differences in downregulation (that is, IL-10, CXCL13 and M-CSF) (Extended Data Fig. 3e,f). Importantly, TNF levels were similarly downregulated, indicating analogous inhibition of immune cell infiltration and overall inflammatory response (Extended Data Fig. 3e,f). Notably, RT–qPCR analysis revealed that IFNγ and IFNβ were significantly attenuated by keratinocyte-specific STING deficiency, implying that STING in these cells drives the intrinsic cytokine expression needed for immune cell infiltration through IFN-response activation (Extended Data Fig. 3g). Comparative RNA-seq analysis of skin tissue from *Casp8*[E-KO], *Casp8*[E-KO]*Tnfr1*[−/−] and *Casp8*[E-KO]*Sting1*[E-KO] mice revealed that genes specifically downregulated after constitutive TNFR1 absence were predominantly involved in TNF-, TLR- and NF-κB-driven inflammatory pathways (Fig. 2g). By contrast, keratinocyte-specific deletion of *Sting1* resulted in specific downregulation of many ISGs, as well as *Mlkl* and *Gsdmd* (Fig. 2h). Furthermore, genes such as *Eif2ak2*, *Gsdmc*, *Ninj1*, *Il6* and *Il1b* were significantly downregulated in both models, suggesting their involvement in amplifying inflammatory cell death rather than initiating disease-causing necroptosis (Fig. 2i). Importantly, *Zbp1* was downregulated in both *Casp8*[E-KO]*Tnfr1*[−/−] and *Casp8*[E-KO]*Sting1*[E-KO] mice (Fig. 2i and Extended Data Fig. 3g), suggesting that, while *Sting1* deletion in keratinocytes contributes to intrinsic *Zbp1* levels, a substantial portion of *Zbp1* expression is keratinocyte-independent, probably

mediated by immune cell infiltration. Moreover, the ability of STING to initiate IFN response, transcriptionally upregulate ZBP1 and, notably, also MLKL (Fig. 2h,i and Extended Data Fig. 3g) underpins its central role in sustaining necroptosis initiation and propagation by providing essential components required for necroptosis execution.

To further elucidate the complementary, independent roles of TNFR1 and STING in *Casp8*[E-KO]-mediated necroptosis and their functional interplay with ZBP1, we generated *Casp8*[E-KO]*Tnfr1*[−/−]*Zbp1*[−/−] and *Casp8*[E-KO]*Tnfr1*[−/−]*Sting1*[E-KO] mice. Notably, heterozygous deletion of *Zbp1* or *Sting1* prevented equally the postnatal lethality observed in *Casp8*[E-KO]*Tnfr1*[−/−] mice (Fig. 3a). These mice reached the end-point criteria at around 4 weeks of age owing to severe dermatitis (Fig. 3a–c), accompanied by reduced body weight, multiorgan inflammation and haematological abnormalities (Fig. 3d,e and Extended Data Fig. 4a–d). Notably, homozygous constitutive co-deletion of *Zbp1* or keratinocyte-specific co-deletion of *Sting1* in *Casp8*[E-KO]*Tnfr1*[−/−] mice rescued all of the pathological phenotypes observed in *Casp8*[E-KO]*Tnfr1*[−/−] mice. Both *Casp8*[E-KO]*Tnfr1*[−/−]*Zbp1*[−/−] and *Casp8*[E-KO]*Tnfr1*[−/−]*Sting1*[E-KO] mice appeared completely normal and were indistinguishable from control mice up until 4 weeks of age (Fig. 3a–e and Extended Data Fig. 4a–d).

This result was further corroborated by the histological analysis of the skin of both *Casp8*[E-KO]*Tnfr1*[−/−]*Zbp1*[−/−] or *Casp8*[E-KO]*Tnfr1*[−/−]*Sting1*[E-KO] mice at 4 weeks of age, which did not reveal any signs of dermatitis (Fig. 3f). By contrast, severe epidermal thickening could be readily observed in both *Casp8*[E-KO]*Tnfr1*[−/−]*Zbp1*[−/+] and *Casp8*[E-KO]*Tnfr1*[−/−]*Sting1*[E-KO/+] mice. These mice displayed elevated immune cell infiltration, increased levels of p-STAT1 and ZBP1 as well as p-MLKL(Ser345)-positive cells in the epidermis. As expected, Cl-CASP3 was also detected due to immune cell infiltration, leading to secondary apoptosis (Fig. 3f).

Constitutive absence of ZBP1 completely rescued the lethal dermatitis observed in *Casp8*[E-KO]*Tnfr1*[−/−] mice through adulthood (Extended Data Fig. 4e,f). Yet, keratinocyte-specific absence of STING significantly extended the survival, with approximately 50% of *Casp8*[E-KO]*Tnfr1*[−/−]*Sting1*[E-KO] mice surviving at least up to 14 weeks (Fig. 3a).

*Casp8*[E-KO]*Tnfr1*[−/−]*Sting1*[E-KO] mice at 12–14 weeks of age showed epidermal thickening, immune cell infiltration and elevated levels of ZBP1, predominantly located in the dermis, accompanied by increased levels of Cl-CASP3 (Extended Data Fig. 4f). These mice also exhibited haematological abnormalities and systemic inflammation (Extended Data Fig. 4g and Supplementary Fig. 3). Notably, a few cells in the epidermis of *Casp8*[E-KO]*Tnfr1*[−/−]*Sting1*[E-KO] mice remained positive for ZBP1 staining (Extended Data Fig. 4f), probably owing to incomplete STING depletion from epidermal keratinocyte.

Collectively, the results presented so far identify a role of aberrant STING activation as a driver of a transcriptional programme responsible for the upregulation of ZBP1 and MLKL in addition to that of many other ISGs. This is both necessary and sufficient to trigger lethal inflammatory disease caused by ZBP1–MLKL-dependent but TNFR1–FADD-independent necroptosis in a cell-intrinsic manner.

## Compartmental STING shapes necroptosis

Next, we sought to determine whether constitutive co-deletion of *Sting1* would mimic the protective benefits observed after ZBP1 co-deletion in *Casp8*[E-KO]*Tnfr1*[−/−] mice. To this end, we first generated *Casp8*[E-KO]*Sting1*[−/−] mice which, notably, survived substantially longer than *Casp8*[E-KO]*Tnfr1*[−/−] mice (Fig. 4a). These mice displayed delayed onset and reduced severity of skin pathology, as evidence by the levels of CD45[+] cells, p-STAT1, ZBP1 and p-MLKL(S345), which were indistinguishable from those of healthy controls at P5 (Fig. 4b and Supplementary Fig. 4). Systemically, by P5, *Casp8*[E-KO]*Sting1*[−/−] mice, similarly to *Casp8*[E-KO]*Tnfr1*[−/−] mice, showed no signs of inflammation (Extended Data Fig. 5a–c). Compared with *Casp8*[E-KO]*Sting1*[E-KO] mice, *Casp8*[E-KO]*Sting1*[−/−] mice not only survived longer (Fig. 4c) but, by P11 (the endpoint of *Casp8*[E-KO]*Sting1*[E-KO] mice), they also showed no significant

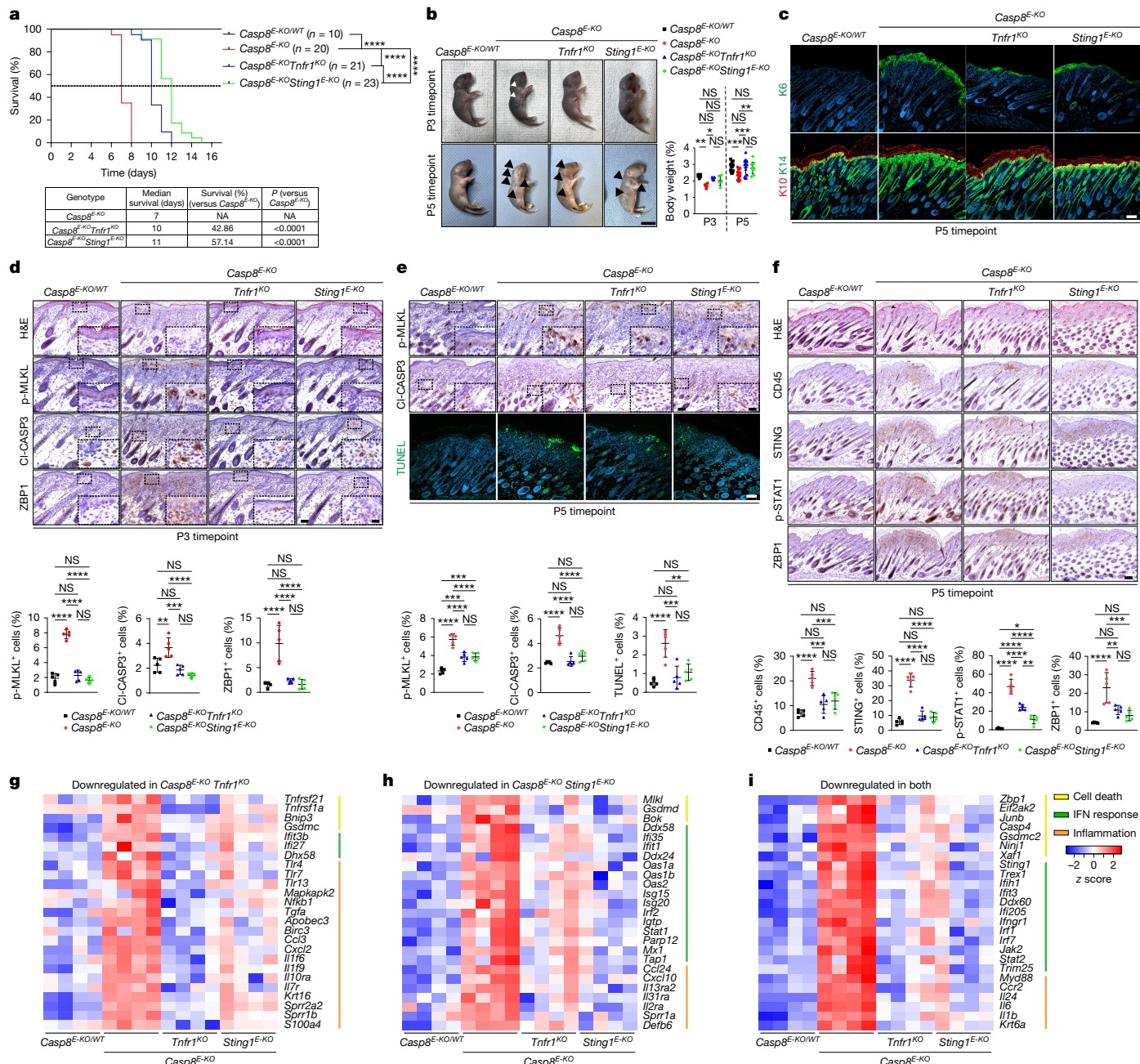

**Fig. 2 | STING-induced necroptosis occurs independently of TNF–TNFR1 signalling. a**, Kaplan–Meier survival curves of *n* mice of the indicated genotypes. The table shows the median survival and the relative survival advantage (%). *P* values were calculated using two-sided Gehan–Breslow–Wilcoxon tests. **b**, Representative images and quantification of the body weight of mice of the indicated genotypes at P3 (*n* = 5 per group) and P5 (*n* = 15, 15, 12 and 10 for the respective genotypes). The arrows indicate areas with lesions. Scale bar, 1 cm. **c**, Representative images of skin sections from P5 mice of the indicated genotypes (*n* = 5 per group) stained for K6 (Alexa Fluor 488, green), K10 (Alexa Fluor 594, red) and K14 (Alexa Fluor 488, green) and nuclei (DAPI, blue). Scale bar, 100 μm. **d**–**f**, Representative images of consecutive skin sections from mice of the indicated genotypes (*n* = 5 per group) at P3 stained with H&E and the indicated antibodies (**d**), at P5 stained with H&E, the indicated antibodies and TUNEL (fluorescein, green) (**e**) and at P5 stained with H&E and the indicated

antibodies (**f**). Scale bars, 50 μm (main bright-field images; **d** and **e** (top and middle)), 20 μm (insets in **d** and **e**), 100 μm (confocal images; **e** (bottom)) and 100 μm (**f**). Immunostaining quantification is shown. **g**–**i**, Significantly downregulated genes in the skin from P5 *Casp8^E-KO^Tnfr1^KO^* (**g**), *Casp8^E-KO^Sting1^E-KO^* (**h**) and both *Casp8^E-KO^Tnfr1^KO^* and *Casp8^E-KO^Sting1^E-KO^* (**i**) mice compared with *Casp8^E-KO^* mice. Representative genes involved in cell death (yellow), IFN response (green) and inflammation (orange) categories are highlighted. Expression values were normalized using DESeq2, row-scaled and colour coded by intensity (red, high; blue, low). Differential expression analysis was performed using the Wald test as implemented in DESeq2, with *P* values adjusted for multiple testing using the Benjamini–Hochberg method (*P* < 0.05). In **b** and **d**–**f**, data are mean ± s.e.m.; *P* values were calculated using two-way ANOVA followed by Tukey's multiple-comparison test; *$P \leq 0.05$, ****$P \leq 0.0001$.

epidermal thickening, a marked reduction in p-MLKL(Ser345) levels in the epidermis, early infiltration of ZBP1-expressing CD45⁺ cells, probably driven by TNF-mediated immune cell recruitment (Fig. 4d and Supplementary Fig. 4), and reduced systemic inflammation (Extended

Data Fig. 5d,e). To determine whether the observed prolonged survival was solely due to suppression of ZBP1-driven necroptosis, we also generated *Casp8^E-KO^Zbp1^−/−^* mice, which we predicted would phenocopy the survival of *Casp8^E-KO^Sting1^−/−^* mice. Unexpectedly, these mice developed

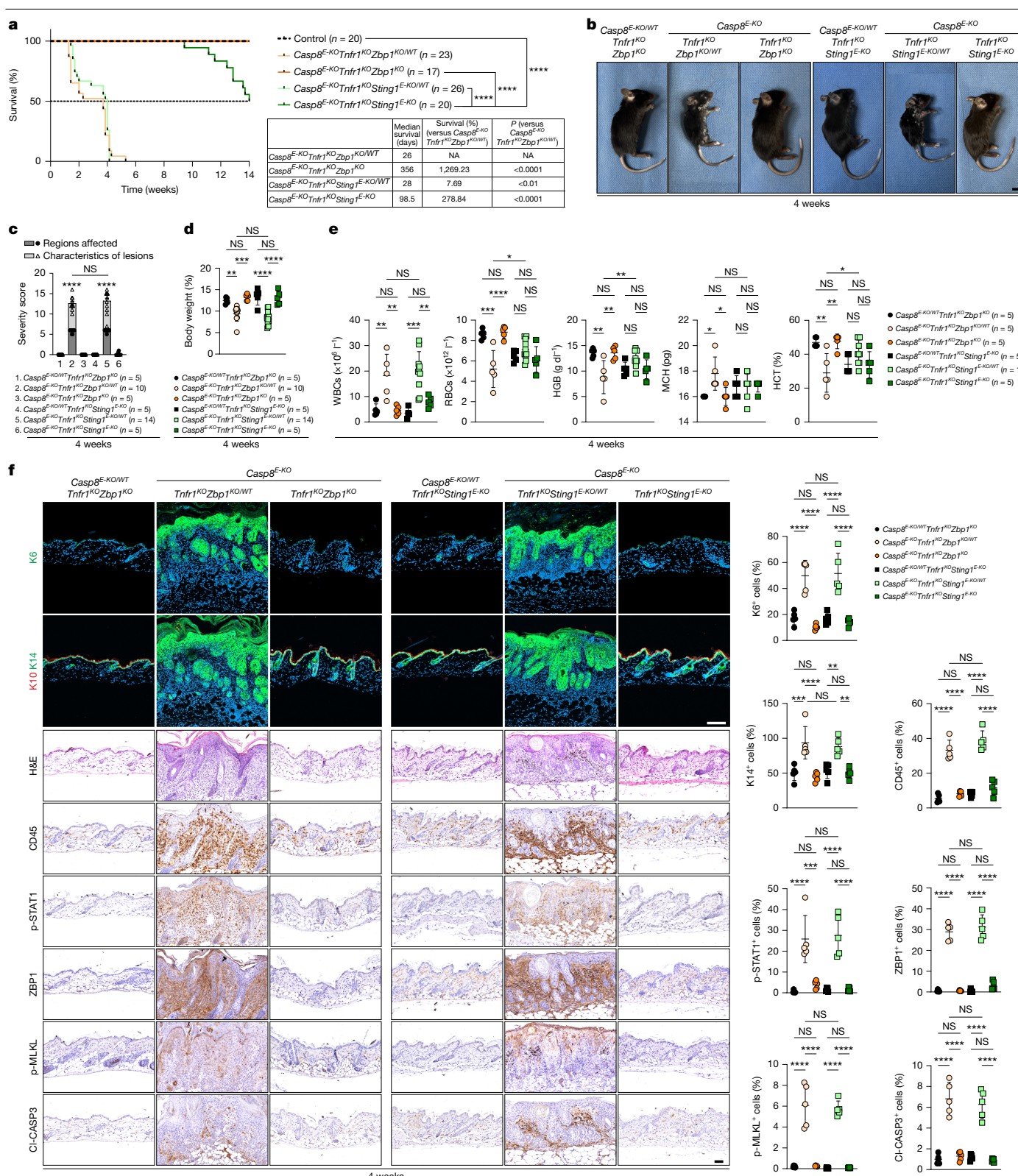

**Fig. 3 | Keratinocyte STING licensing of ZBP1 mediates TNF-independent necroptosis. a**, Kaplan–Meier survival curves of *n* mice of the indicated genotypes. The table shows the median survival and the relative survival advantage (%). *P* values were calculated using two-sided Gehan–Breslow–Wilcoxon tests. NA, not applicable. **b**–**e**, Representative images (**b**) and quantification of the dermatitis severity score (**c**), body weight (**d**) and white blood cell (WBC), red blood cell (RBC), haemoglobin (HGB), mean corpuscular haemoglobin (MCH) and haematocrit (HCT) values in the peripheral blood (**e**) of 4-week-old mice of the indicated genotypes. *n* values are indicated. Control mice included age-matched *Casp8^E-KO/WT^Tnfr1^KO^Zbp1^KO* and *Casp8^E-KO/WT^Tnfr1^KO^Sting1^E-KO*

littermates. Scale bar, 1 cm (**b**). **f**, Representative images of consecutive skin sections from 4-week-old mice of the indicated genotypes stained for K6 (Alexa Fluor 488, green), K10 (Alexa Fluor 594, red), K14 (Alexa Fluor 488, green), nuclei (DAPI, blue), H&E and the indicated antibodies (*n* = 5 per group). Scale bars, 50 μm (bright-field images, bottom 6 rows) and 100 μm (confocal images, top two rows). Immunostaining quantification is shown. For **c**–**f**, data are mean ± s.e.m.; each dot represents one mouse; *P* values were calculated using two-way ANOVA followed by Tukey's multiple-comparison test; *$P \leq 0.05$, **$P \leq 0.01$, ***$P \leq 0.001$, ****$P \leq 0.0001$; NS, not significant ($P > 0.05$).

severe, lethal dermatitis and systemic inflammation indistinguishable in timing and severity from $Casp8^{E-KO}$ mice (Fig. 4a,b and Extended Data Fig. 5a–c), underscoring that ZBP1 loss alone was insufficient to delay disease onset, progressions and severity. As $Casp8^{E-KO}Tnfr1^{-/-}Zbp1^{-/-}$ mice exhibited full rescue with complete absence of dermatitis and systemic inflammation (Fig. 3 and Extended Data Fig. 4), we concluded that (1) TNF–TNFR1 acts as the initial necroptosis checkpoint that must be bypassed to engage into ZBP1-mediated necroptosis; (2) ZBP1-mediated necroptosis acts as a second, independent, necroptotic checkpoint; (3) given the survival of the $Casp8^{E-KO}Sting1^{-/-}$ mice encompassing and surpassing the survival of the $Casp8^{E-KO}Tnfr1^{-/-}$ mice (Fig. 4e), STING signalling not only licenses the ZBP1 necroptotic checkpoint but also partially contributes to TNF–TNFR1-driven cell death and inflammation, probably through autocrine TNF production and NF-κB pathway engagement (Fig. 4f). Consistently, transcriptional upregulation of several inflammatory genes, including $Tnf$, $Zbp1$ and $Mlkl$, was completely suppressed in $Casp8^{E-KO}Sting1^{-/-}$ mice, and RNA-seq analysis revealed a similar downregulation for NF-kB signature genes compared with in $Casp8^{E-KO}Tnfr1^{-/-}$ mice (Extended Data Fig. 5f,g). This confirmed the requirement for STAIN, as well as a functional overlap between STING and TNFR1 in regulating NF-kB-driven inflammatory programme in the absence of caspase-8. Thus, the extended survival observed in $Casp8^{E-KO}Sting1^{-/-}$ mice compared with $Casp8^{E-KO}Tnfr1^{-/-}$ mice is best explained by STING's dual contribution to both necroptotic arms. Collectively, this establishes the STING axis as an independent and genetically separable checkpoint downstream of caspase-8 loss, which also overlaps with the TNFR1-dependent pathway to amplify inflammatory necroptosis (Fig. 4f).

To fully disentangle the functional overlap between STING and ZBP1, we generated $Casp8^{E-KO}Tnfr1^{-/-}Sting1^{-/-}$ mice. Notably, these mice remained viable, healthy and phenotypically indistinguishable from control littermates and $Casp8^{E-KO}Tnfr1^{-/-}Zbp1^{-/-}$ mice (Fig. 4g). At 12 weeks of age (shortly before $Casp8^{E-KO}Tnfr1^{-/-}Sting1^{E-KO}$ mice reach their survival endpoint), $Casp8^{E-KO}Tnfr1^{-/-}Sting1^{-/-}$ mice showed no signs of lethal dermatitis (Fig. 4h, Extended Data Fig. 5h and Supplementary Fig. 4). Moreover, whereas $Casp8^{E-KO}Tnfr1^{-/-}Sting1^{E-KO}$ mice at 12 weeks of age revealed severe systemic disease, $Casp8^{E-KO}Tnfr1^{-/-}Sting1^{-/-}$ mice appeared indistinguishable from the littermate controls (Extended Data Fig. 5i–k and Supplementary Fig. 4). Histological examination of skin sections showed suppression of lethal dermatitis as evidenced by normalized epidermal thickness, loss of p-MLKL and ZBP1 expression and CD45⁺, and Cl-CASP3-positive cells (Fig. 4h). This marked phenotypic divergence confirms that the residual necroptosis observed in $Casp8^{E-KO}Tnfr1^{-/-}Sting1^{E-KO}$ mice was possibly due to incomplete $Sting1$ deletion. Although STING expression in non-keratinocyte compartments might also have a pivotal role in sustaining and propagating systemic necro-inflammation, STING activation in keratinocytes is essential to initiate ZBP1-mediated necroptosis and early inflammation. Gene dosage analysis revealed that the survival of $Casp8^{E-KO}Tnfr1^{-/WT}$ mice was significantly prolonged after homozygous constitutive co-deletion of $Sting1$, particularly when compared to homozygous co-deletion of $Zbp1$ or keratinocyte-specific $Sting1$ deletion (Fig. 4i). This confirms a role of STING also in regulating TNFR1-induced necroptosis and cell-extrinsic necroptotic inflammation. This dual role of STING, as both a trigger and amplifier of necroptosis, highlights its essential and non-redundant function in TNFR1-independent, ZBP1-mediated pathology.

## gDNA activates cGAS–STING in $Casp8^{E-KO}$

To determine the source of STING activation downstream of $Casp8$ loss, we generated $Casp8^{E-KO}Cgas^{-/-}$ mice. $Cgas$ co-deletion fully phenocopied the protection observed in $Casp8^{E-KO}Sting1^{-/-}$ mice, as evidence by the survival analysis and comparable suppression of tissue inflammation through histological analysis at P5 (Fig. 4j,k and Supplementary Fig. 4).

To exclude alternative nucleic-acid-sensing pathways, we also generated $Casp8^{E-KO}Mavs^{-/-}$ mice. $Mavs$ deletion did not rescue lethality or reduce tissue pathology (Fig. 4j,k and Supplementary Fig. 4). These results indicate that activation of STING after $Casp8$ loss is entirely dependent on cGAS, placing cytosolic DNA sensing upstream of the type I IFN response signature observed in $Casp8^{E-KO}$ mice.

To identify the origin of the DNA species activating cGAS, we first examined the potential role of mitochondrial DNA (mtDNA), as a cGAS agonist[27]. Immunofluorescence staining in $Casp8^{-/-}$, $Casp8^{-/-}Mlkl^{-/-}$ and $Casp8^{-/-}Ripk3^{-/-}$ MEFs revealed cytosolic accumulation of DNA, but no detectable co-localization with the mitochondrial transcription factor A (TFAM) (Extended Data Fig. 6a). mtDNA accumulation was excluded as treatment with 2′,3′-dideoxycytidine (ddC) or IMT1B showed incomplete reduction in DNA staining (Extended Data Fig. 6b), of which the successful depletion was confirmed by RT–qPCR (Extended Data Fig. 6c). By contrast, there was no reduction in ISG transcription (including $Zbp1$) (Extended Data Fig. 6d). These results were further corroborated at the protein level–ZBP1 expression and p-STAT1 levels remained unchanged and could be completely abrogated only after treatment with the STING antagonist C-178 (Extended Data Fig. 6e).

Previous research showed that caspase-8 deficiency induces chromosomal instability[28] and DNA damage[29], which can result in nuclear DNA leakage into the cytosol, known to be a potent activator of cGAS. In agreement with this, we observed that skin obtained from $Casp8^{E-KO}$, $Casp8^{-/-}Ripk3^{-/-}$, $Casp8^{-/-}Mlkl^{-/-}$ as well as $Casp8^{E-KO}Tnfr1^{-/-}Sting1^{-/-}$ and $Casp8^{E-KO}Tnfr1^{-/-}Zbp1^{-/-}$ mice exhibited mitotic abnormalities, with many phospho-histone H3 (p-HH3)-positive cells displaying aberrant chromosome segregation and elevated γ-histone 2A..X (γH2A..X) foci (Fig. 4l and Extended Data Fig. 6f). These findings suggest that caspase-8 deficiency compromises genome integrity independently of necroptosis, as previously reported[28–31], supporting a model in which genomic DNA released due to caspase-8-loss-induced genome instability acts as a primary trigger of cGAS–STING activation in this context.

## Z-NAs and ZBP1 are amplified by STING

To understand how ZBP1 is activated after its upregulation by STING, we used $Casp8^{-/-}$, $Casp8^{-/-}Mlkl^{-/-}$ and $Casp8^{-/-}Ripk3^{-/-}$ MEFs and performed immunofluorescence staining to specifically detect Z-form nucleic acids (Z-NAs)[22]. Our results revealed specific cytosolic accumulation of both Z-DNA and Z-RNA in these cells, with the Z-NA signal being completely abolished only after treatment with both DNase I and RNase A, confirming the presence of both nucleic acid species in the Z-conformation (Fig. 4m,n and Extended Data Fig. 7a,b).

We next assessed the co-localization between ZBP1 and Z-NAs. Clear spatial overlap was observed, indicating physical proximity and probable interaction (Extended Data Fig. 7c–e). Importantly, treatment with the STING agonist DMXAA further enhanced the Z-NA signal intensity, suggesting that STING activation promotes the accumulation and/or stabilization of Z-NAs, the natural ligands of ZBP1 (Extended Data Fig. 7c–e). Given the consistent co-localization with Z-NA under conditions of STING agonism, we conclude that STING enhances ZBP1 activation through increased generation or exposure to Z-NAs. This correlates well with our biochemical findings indicating that STING activation leads to increased recruitment and phosphorylation of RIPK1 and RIPK3 in ZBP1 immunoprecipitates, further supporting functional activation of ZBP1 in this context.

## STAIN is functionally linked to SAVI

In the genetic model we studied so far, aberrant STING activation is caused by caspase-8 deficiency. Yet, constitutive STING signalling can also result from gain-of-function (GOF) mutations in $STING1$ (refs. 32,33) independently of caspase-8, as evident in patients with

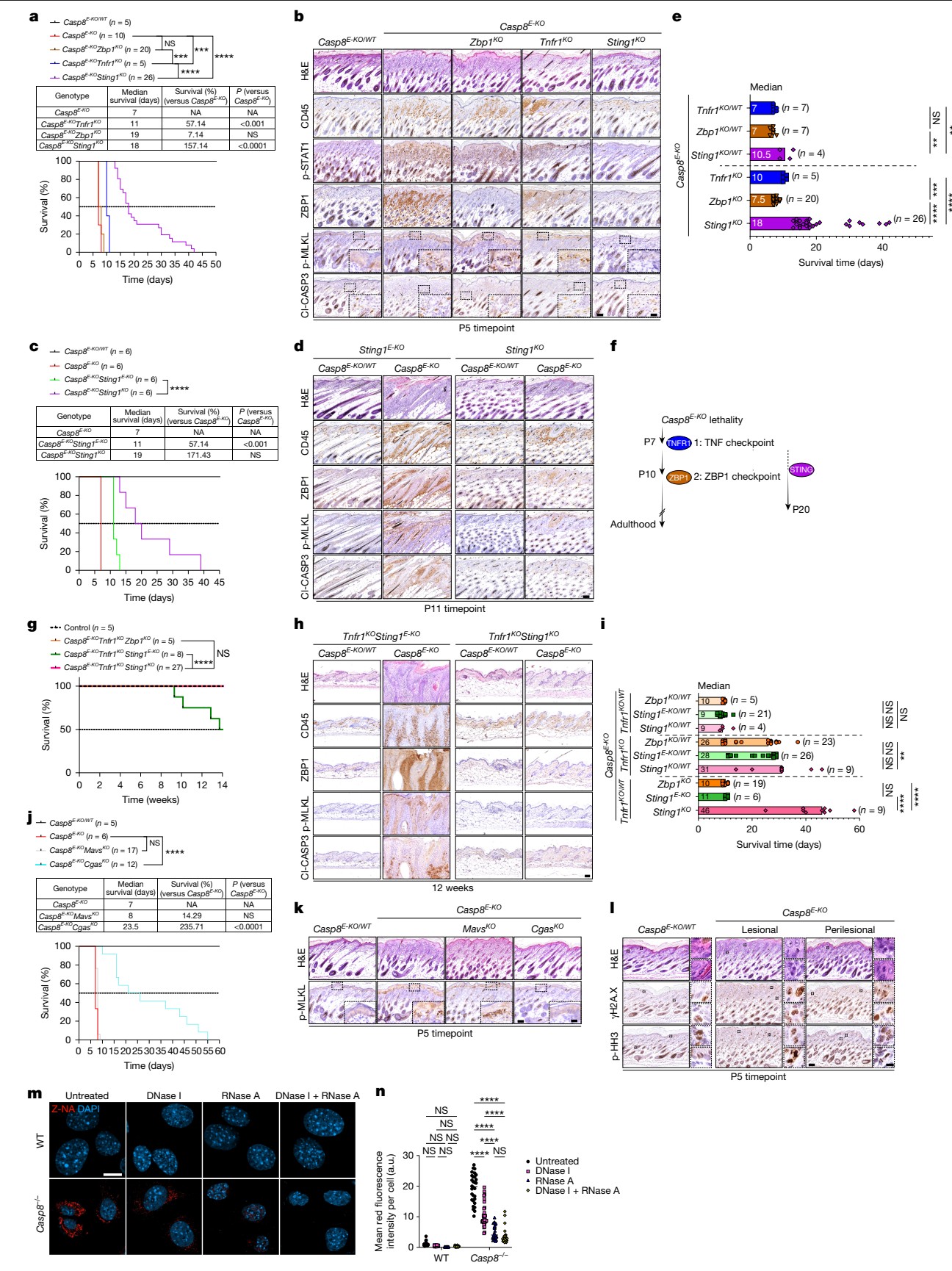

**Fig. 4 | See next page for caption.**

**Fig. 4 | STING regulates necroptosis checkpoints through cGAS–gDNA and ZBP1/Z-NA sensing. a–d**, Kaplan–Meier survival curves of *n* mice of the indicated genotypes. **a,c**, The median survival and the relative survival advantage (%). **b,d**, Representative images of consecutive skin sections from P5 (**b**) or P11 (**d**) mice of the indicated genotypes stained with the indicated antibodies. *n* = 5 (**b**) and *n* = 3 (**d**) per group. For **b** and **d**, scale bars, 100 µm (main images) and 20 µm (insets). **e**, The median survival of *n* mice of the indicated genotypes. **f**, Graphical representation of the TNFR1 and ZBP1 distinct necroptotic checkpoints and their dependence on STING. **g**, Kaplan–Meier survival curves of *n* mice of the indicated genotypes. Control mice included age-matched *Casp8^{E-KO/WT}Tnfr1^{KO}Zbp1^{KO}*, *Casp8^{E-KO/WT}Tnfr1^{KO}Sting1^{E-KO}* and *Casp8^{E-KO/WT}Tnfr1^{KO}Sting1^{KO}* littermates. **h**, Representative images of consecutive skin sections from 12-week-old mice of the indicated genotypes stained with the indicated antibodies. *n* = 3 per group. Scale bar, 50 µm. **i**, The median survival of *n* mice of the indicated genotypes. **j**, Kaplan–Meier survival curves of *n* mice of the indicated genotypes. The table shows the median survival and the relative survival advantage (%). **k,l**, Representative images of consecutive skin sections from P5 mice of the indicated genotypes stained as indicated. *n* = 3 (**k**) and *n* = 5 (**l**). For **k** and **l**, scale bars, 100 µm (main images) and 20 µm or 10 µm (insets). **m,n**, Representative confocal images of the indicated MEFs stained for Z-NA (Alexa Fluor 594, red) and nuclei (DAPI, blue) under basal conditions or after treatment with DNase I (25 U ml⁻¹) and/or RNase A (1 mg ml⁻¹) (**m**) and quantification of mean Z-NA (red) fluorescence intensity (**n**). Data are mean ± s.e.m. *P* values were calculated using two-way ANOVA followed by Tukey's multiple-comparisons test. For **a**, **c**, **e**, **g**, **i** and **j**, *P* values were calculated using two-sided Gehan–Breslow–Wilcoxon tests; each dot in the graphs represents one mouse; **$P \leq 0.01$, ***$P \leq 0.001$, ****$P \leq 0.0001$; NS, not significant ($P > 0.05$).

---

SAVI. SAVI is an autosomal-dominant genetic interferonopathy characterized by early-onset systemic inflammation, chronic anaemia, cutaneous lesions, polyarthritis-like symptoms and interstitial lung disease. Conventional immunosuppressive treatments, anti-malarials, TNF inhibitors and B-cell-depleting biologics have provided limited to no therapeutic effect[34,35]. Recently, new regimens have included JAK1/2 inhibitors (such as ruxolitinib); however, their efficacy has yielded only a temporary reduction in symptoms[32,36].

Considering the identification of aberrant STAIN as a critical driver of lethal inflammatory necroptosis, we hypothesized that STAIN could contribute or even cause the clinical manifestation of SAVI. We therefore examined a set of longitudinal samples obtained from paediatric patients affected by SAVI[37]. For each patient, samples were collected before and after conventional treatment, which included ruxolitinib for an average duration of 18 months (Extended Data Fig. 8a). Assessment of cell-death-involved inflammatory cytokines in the plasma of these patients (Supplementary Table 1) revealed that nearly all of the assessed cytokines were elevated in all samples, independently of the treatment and duration, suggesting a potential contribution of cell-death-induced inflammation to clinical manifestations and persistence of SAVI (Extended Data Fig. 8b and Supplementary Fig. 5). This analysis confirmed known cytokines[38] and identified additional top-ranked candidates (IL-21, CXCL1, IL-1β, IFNα and CD40L) as potential circulating biomarkers (Extended Data Fig. 8b). RNA-seq analysis in whole blood cells obtained from plasma-matching patients samples revealed a significant upregulation of genes involved in both type I and/or type II IFN responses especially in 11 out of 15 samples, which included almost all post-treatment samples (Fig. 5a). Further transcriptional and functional investigation confirmed sustained IFN response activation and, notably, the differential upregulation of *MLKL* and *ZBP1* (Fig. 5b). We further functionally grouped these genes and categorized them into three categories—cell death, IFN response and inflammation (Extended Data Fig. 8c and Supplementary Fig. 6)—and evaluated the levels of executioners, inhibitors and modulators of all functionally upregulated cell death modalities (Extended Data Fig. 8d–f). By doing so, we excluded transcriptional priming of mitochondrial apoptosis due to high expression of inhibitors of apoptosis. Pyroptosis appeared to be predominantly executed in patient 2. Cell death pathways triggered by TNF or other death ligands did not appear to be significantly transcriptionally upregulated or were accompanied by concomitant upregulation of checkpoint genes such as cIAPs and the linear ubiquitin chain assembly complex (LUBAC). Importantly, *RIPK3*, *MLKL* and *ZBP1* were consistently overexpressed, indicative of priming towards necroptosis execution in all patient samples (Extended Data Fig. 8d–f). This was confirmed by RT–qPCR analysis, in which the expression of a small panel of genes used to classify interferonopathies clinically and its fluctuations before and after treatment mirrored the expression levels of *MLKL*, *ZBP1*, and the inflammatory chemokines *CXCL1* and *CXCL8* (ref. 39) (Supplementary Fig. 7). This suggested a functional correlation between STING-mediated IFN response, the transcriptional expression of *ZBP1* and *MLKL*, and the cell-autonomous cytokines known to be released by necroptosis.

Importantly, transcriptomic comparison of upregulated pathways in the *Casp8^{E-KO}* mice (versus the control) with the pathways upregulated in IFN^{high} SAVI samples revealed a significant functional overlap of 73% of all pathways (Fig. 5c). Within these, we identified a set of key genes overlapping between *Casp8^{E-KO}* mice and IFN^{high} SAVI samples (Supplementary Table 2), suggesting that these pathways and genes are descriptive of an established necroptotic program within the SAVI transcriptome and vice versa of the newly characterized STING-driven necroptosis.

## STAIN leads to lethality in SAVI mice

We next turned to the established preclinical SAVI mouse model bearing the heterozygous *Sting1^{N153S}* GOF mutation (hereafter *Sting1^{N153S}* mice), which closely phenocopies human SAVI, faithfully recapitulating key clinical features of this disease[40] (Extended Data Fig. 9a–f).

Histological analysis of skin, lung, thymus and spleen collected from *Sting1^{N153S}* mice, presenting the complete clinical manifestations of SAVI, revealed robust ZBP1 upregulation (Extended Data Fig. 9g). Importantly, necroptosis, detected through p-MLKL(Ser345) staining, was restricted to the skin, while the lung and thymus displayed both p-MLKL(Ser345)- and Cl-CASP3-positive staining (Extended Data Fig. 9g). Spleen exhibited strong Cl-CASP3 staining only, indicative of dominant apoptosis in that compartment at the endpoint (Extended Data Fig. 9g). To determine the role of necroptosis in disease manifestation and pathogenesis and confirm preclinically the translational relevance of our findings, we generated *Sting1^{N153S}Ripk3^{-/-}* mice. These mice showed no signs of disease and exhibited unaffected survival at least up until 35 weeks of age, which was the median survival of the *Sting1^{N153S}* mice (Fig. 5d,e). Histological analysis of 35-week-old *Sting1^{N153S}Ripk3^{-/-}* mice showed a complete absence of p-MLKL(Ser345) staining with a clear rescue in skin, lung, thymus and splenic inflammation and architecture, providing conclusive evidence for the role of STAIN in the inflammatory manifestations of SAVI (Fig. 5f).

The complex multiorgan inflammation characteristic of the SAVI mice is known to derive from STING-GOF-induced triggering of pathogenic early-onset immune dysregulation (6–8 weeks)[41,42]. High-resolution immunophenotyping of the lung, spleen and thymus revealed that co-deletion of *Ripk3* significantly ameliorates the immune abnormalities induced by STING GOF (Fig. 5g–i and Extended Data Fig. 10). In the thymus, *Sting1^{N153S}Ripk3^{-/-}* mice showed recovery of double-negative thymocytes (Extended Data Fig. 10b). CD8⁺ and double-positive (DP; CD4⁺CD8⁺) thymocytes, along with their activation status (CD69⁺), were significantly normalized, pointing to a contributing (but not complete) role for necroptosis in thymic T cell attrition (Fig. 5g and Extended Data Fig. 10b). In the lung, where SAVI pathology often manifests with both inflammation and fibrosis, *Ripk3* co-deletion mitigated the expansion of inflammatory monocytes and

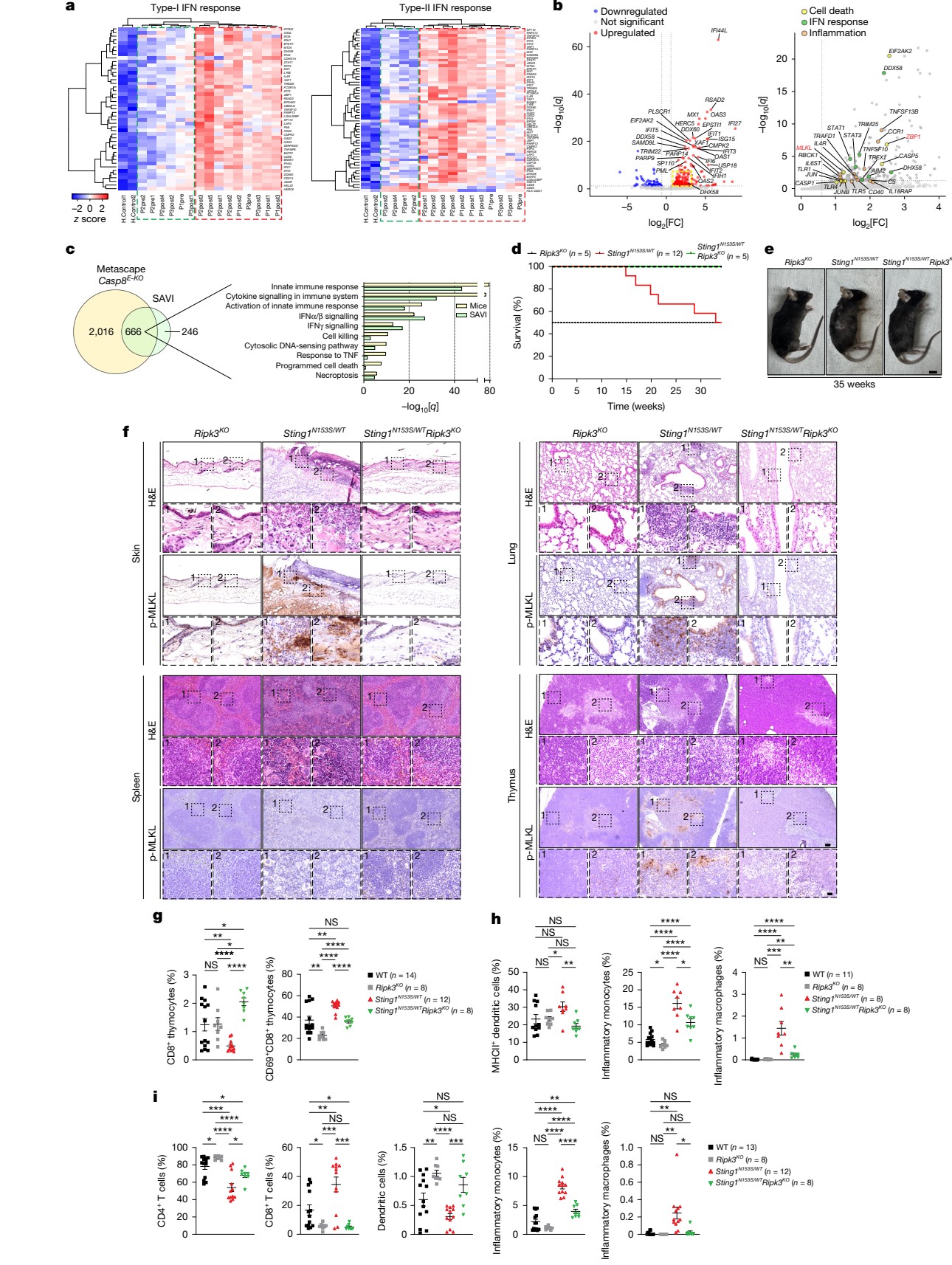

**Fig. 5 | See next page for caption.**

**Fig. 5 | STAIN shapes SAVI transcriptional signatures immune dysregulation and lethality. a**, Type I and type II IFN response signatures in SAVI samples (P(x) post(y) or pre(y), where x = patient number, y = sample number, pre = pre-treatment and post = post-treatment; see Extended Data Fig. 8a). Genes upregulated in SAVI versus controls ($P < 0.05$) that overlap with Hallmark IFNα and IFNγ response gene sets (GSEA) are shown. Expression values are row-scaled and colour coded (red, high; blue, low). The green and red boxes indicate IFN$^{low}$ and IFN$^{high}$ signature samples, respectively. **b**, Differentially expressed genes in type II IFN$^{high}$ SAVI samples compared with controls ($P_{adj} < 0.05$ and fold change (FC) > 1.5). IFN-responsive genes (left) and genes belonging to cell death (yellow), IFN response (green) and inflammation (orange) are annotated. **c**, Pathway comparative analysis from Metascape data. The pathway overlap between *Casp8$^{E-KO}$* mice and SAVI IFN$^{high}$ samples is shown. The bar plot displays $-\log_{10}[q]$ values for ten representative overlapping pathways. **d,e**, Kaplan–Meier survival curves (**d**) and representative images (**e**) of *n* mice of the indicated genotypes. *P* values were calculated using two-sided Gehan–Breslow–Wilcoxon tests.

**f**, Representative images of the indicated consecutive tissue sections from 35-week-old mice of the indicated genotypes stained with H&E and p-MLKL (Ser345). *n* = 3 per group. Scale bars, 100 μm (main images) and 20 μm (insets). **g**–**i**, Flow cytometry analysis of thymus (**g**), lung (**h**) and spleen (**i**) from 6–8-week-old mice of the indicated genotypes (*n* values are indicated). CD8$^{+}$ thymocytes are plotted as the percentage of viable cells, and CD69$^{+}$CD8$^{+}$ thymocytes, MHCII$^{+}$ dendritic cells, and CD4$^{+}$ and CD8$^{+}$ T cells are depicted as the percentage of their respective parental populations; all other populations were plotted as the percentage of CD45$^{+}$ cells. For **a** and **b**, differential expression analysis was performed using the Wald test as implemented in DESeq2; *P* values were adjusted for multiple testing using the Benjamini–Hochberg method. For **g**–**i**, age-matched WT and *Ripk3*-KO littermates were used as controls; data are mean ± s.e.m.; each dot represents one mouse; *P* values were calculated using two-tailed unpaired *t*-tests; *$P \le 0.05$, **$P \le 0.01$, ***$P \le 0.001$, ****$P \le 0.0001$; NS, not significant ($P > 0.05$).

macrophages, antigen-presenting dendritic cells and granulocytes (Fig. 5h and Extended Data Fig. 10d). Although T cell restoration was more modest, the overall pattern again indicated a broad rescue of immune homeostasis. In the spleen, RIPK3 co-deficiency led to marked improvements in both the lymphoid and myeloid compartments. *Sting1$^{N153S}$* mice exhibited severe T cell and natural killer cell lymphopaenia accompanied by expansion of antigen-presenting dendritic cells, granulocytes, inflammatory monocytes and macrophages, all of which are hallmarks of chronic STING activation (Fig. 5i and Extended Data Fig. 10f). These aberrations were consistently and significantly rescued in *Sting1$^{N153S}$Ripk3$^{-/-}$* mice. A similar normalization was observed in bone-marrow-derived and splenic myeloid subsets of the spleen, confirming that RIPK3-dependent necroptosis is a major driver of haematopoietic dysregulation (Extended Data Fig. 10f). Importantly, the frequency of immature erythroid precursors (Ter119$^{+}$CD71$^{+}$) was also reduced, indicating a reversal of extramedullary erythropoiesis and systemic inflammation (Extended Data Fig. 10g).

These findings propose that early necroptosis of immune cells, particularly T cells, contributes directly to the loss of immune homeostasis and the unleashing of systemic inflammation. This cascade probably facilitates the organ-specific inflammatory pathologies observed in SAVI, including dermatitis, interstitial lung inflammation and splenomegaly. In this way, necroptosis acts not only as a mechanism of immune cell loss, but also as a driver of systemic disease progression, reinforcing the functional and translational relevance of our findings.

Collectively, our findings reveal that caspase-8 deficiency in keratinocytes aberrantly activates a STING-mediated upregulation of ZBP1 and MLKL that, in turn, leads to ZBP1–MLKL-mediated necroptosis. We find that caspase-8 deficiency leads to the accumulation of genomic instability and activation of cGAS. Importantly, aberrant STING activation enables necroptosis by facilitating the formation of a ZBP1–RIPK1–RIPK3 complex, which forms and drives necroptosis independently of the TNF–TNFR1-induced FADD–RIPK1–RIPK3 complex. Activation of STING also induces further accumulation and stabilization of Z-NAs, which in turn serve as ligands for ZBP1 activation (Supplementary Fig. 8). Co-ablation of STING in keratinocytes or constitutively together with full-body depletion of TNFR1 revealed a substantial or complete additional survival advantage to *Casp8$^{E-KO}$* mice, respectively. This provides genetic proof that aberrantly activated STING is capable of driving a TNFR1-independent necroptosis pathway in vivo responsible for ZBP1–RIPK3–MLKL-mediated necroptosis. Our analysis provides compelling evidence that STING enables necroptosis not only by inducing the transcriptional upregulation of ZBP1 but also by contributing to autocrine TNF production potentially through NF-κB activation. This dual role positions STING as a central inducer and amplifier of necroptotic signalling. Importantly, the extended survival and broader rescue observed after systemic *Sting1* deletion, compared with the deletion of either *Zbp1* or *Tnfr1* alone, suggest that STING governs both the TNFR1

and ZBP1 checkpoints. Thus, STING not only licenses the ZBP1 axis and its activation but also reinforces the TNFR1-dependent pathway, underscoring its unique role as a shared upstream regulator of inflammatory necroptosis.

The necroptosis-driven lethal inflammation observed in *Casp8$^{E-KO}$* mice shares transcriptional commonalities with SAVI, providing further functional links between aberrant STAIN and development of SAVI. In the *Sting1$^{N153S}$* mice, *Ripk3* co-deletion alone robustly improves survival and overall necroptosis-induced cell death in all tissues and ameliorates early immune dysfunction. The genetic and pathological rescue in affected tissues at the endpoint underscores that necroptosis contributes not only to disease initiation but also to its clinical progression and organ-specific pathology. These findings establish necroptosis as an important effector pathway underlying both the onset and manifestation of SAVI and possibly other autoinflammatory diseases arising from pathogenic STING activation. Importantly, the robust rescue observed in *Sting1$^{N153S}$* mice with RIPK3 co-deficiency alone, without the necessity for caspase-8 inactivation, demonstrates that caspase-8 expression in SAVI is functionally irrelevant for controlling necroptosis. This, together with our biochemical data demonstrating the role of STING in (1) the formation of a ZBP1–RIPK1–RIPK3 complex independent of TNFR1 and FADD and (2) the upregulation of Z-NAs required to activate ZBP1, explains the activation of necroptosis in patients despite maintaining functional expression of caspase-8.

In summary, our study identifies a STING-driven necroptotic signalling axis that underlies severe inflammatory disease in vivo. This positions STING as a central node in a feedforward necroptotic inflammatory circuit that underlies sustained tissue pathology. By establishing STAIN as a pathogenic mechanism in SAVI, our findings offer a conceptual advance in understanding interferonopathies and identify the STING–ZBP1–necroptosis axis as a rational and tractable target for therapeutic intervention in this otherwise treatment-refractory disease.

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

# Methods

## Mice

$Casp8^{fl/fl}$ mice were provided by S. M. Hedrick[43] and the $K14$-$Cre$[44] and $Tnfr1^{-/-}$ (ref. 45) mice were obtained as previously described. $Zbp1^{-/-}$ mice were purchased from Cyagen ($C57BL/6NCya$-$Zbp1^{em1}/Cya$, S-KO-16146) and $Sting1^{fl/fl}$ ($B6;SJL$-$Sting1^{tm1.1Camb}/J$, 031670)[46], $Sting1^{-/-}$ ($B6(Cg)$-$Sting1^{tm1.2Camb}/J$, 025805)[46], $Cgas^{-/-}$ ($B6(C)$-$Cgas^{tm1d(EUCOMM)Hmgu}/J$, 026554)[47] and $Sting1^{N153S/WT}$ ($B6J.B6N$-$Sting1^{em1Jmin}/J$, 033543)[40] mice were purchased from The Jackson Laboratory. $Mlkl^{-/-}$ mice were generated by the Walczak laboratory[48] and $Casp8^{+/-}$ and $Ripk3^{-/-}$ mice were provided by R. Hakem, V. Dixit and K. Newton, respectively[49]. $Mavs^{-/-}$ mice were provided by the SFB1403 consortium[19]. All mice were maintained on a C57BL/6N background; C57BL/6J strains were backcrossed for at least ten generations. Male mice carrying the $K14$-$cre$ transgene were crossed to obtain all $Casp8^{E\text{-}KO}$ models. $Casp8^{-/WT}Ripk3^{-/-}$ or $Casp8^{-/WT}Mlkl^{-/-}$ were bred to obtain $Casp8^{-/-}Ripk3^{-/-}$ and $Casp8^{-/-}Mlkl^{-/-}$ double-knockout mice, respectively. Mice heterozygous for $Sting1^{N153S}$ ($Sting1^{N153S/WT}$) were bred with $Ripk3^{-/-}$ mice to obtain $Sting1^{N153S/WT}Ripk3^{-/-}$ mice. Mice were housed under SPF conditions in individually ventilated cages (12 h–12 h light–dark cycle) at the University of Cologne's Medical Faculty and CECAD animal facilities, with ad libitum access to food and water. Temperature and humidity were controlled. Animals were monitored daily and euthanized after reaching predefined humane-endpoint criteria for disease severity. No additional exclusion criteria were applied. Experimental groups were not randomized because group assignment was determined by genotype; within each genotype, female and male mice were enrolled without preference and allocated to experimental conditions at random. Sample size was estimated to reach necessary statistical power. The age of mice at analysis is reported in the corresponding figure legends. All mouse scoring was performed at least by three independent scientists, experiments and the immunohistochemical evaluation of pathology were performed under blinded conditions. All of the procedures complied with German animal welfare regulations and were approved by local authorities (Landesamt für Natur, Umwelt und Verbraucherschutz Nordrhein-Westfalen).

## Cell lines

$Casp8$-knockout cell lines were generated by CRISPR–Cas9-mediated gene editing. WT and $Zbp1$-knockout MEFs were electroporated with $Casp8$-targeting gRNA (Mm.Cas9.CASP8.1.AA, IDT) using the 4D-Nucleofector System and Nucleofector Kit V (Lonza) according to the manufacturer's instructions. $Casp8^{-/-}Sting1^{-/-}$ double-knockout cell lines were generated by isolating and transfecting $Casp8^{fl/fl}Sting1^{-/-}$ MEFs with cre-GFP plasmid (13776, Addgene), followed by FACS-based single-cell cloning and expansion. Knockout efficiency was confirmed by western blotting and genomic sequencing. HT29 and HaCaT cells were obtained from ATCC. HT29 $ZBP1^{-/-}$ cells were generated via CRISPR–Cas9 (sgRNA: CAGCTGGGCAAGTTTCACCG), and the HT29 $Casp8^{-/-}$ cell line was kindly provided by the Pasparakis laboratory. All primary MEFs were isolated from corresponding mouse strains and immortalized as described previously[48].

## Cell death assay

Immortalized MEFs were seeded to 96-well plate ($1 \times 10^4$ cells per well) and incubated with either DMSO or pretreated with 20 µg ml$^{-1}$ DMXAA (D5817, Sigma-Aldrich). Similarly, HaCaT and HT-29 cell lines were pretreated with 10 µM ADU-S100 (HY-12885B, MedChemExpress)[50]. The next day, medium containing DMSO, DMXAA or ADU-S100 was replenished and cells were additionally treated with 100 ng ml$^{-1}$ recombinant TNF (in house), 20 µM Z-VAD-FMK (APE-A1902-25MG, APExBIO), 2 µM BV6 (S7597, Selleck Chemicals) or combinations of them. In experiments using RIPK3, MLKL, TNF or STING inhibitor, 10 µM GSK'872 (S8465, Selleck Chemicals), 1 µM NSA (S8251, Selleck Chemicals),

50 µg ml$^{-1}$ etanercept (Enbrel, PAA115147, Pfizer) or 1 µM C-178 (S6667, Selleck Chemicals)[25] were added as pretreatments, respectively. Cell viability was measured in the Spark Multimode Microplate Reader (TECAN) using the CellTiter-Glo Luminescent Cell Viability Assay (G7573, Promega).

## Protein lysate preparation and western blot analysis

Cell lysates were prepared and analysed by western blotting as previously described[28]. A list of the antibodies used is provided in Supplementary Table 3. Uncropped blot images are provided in Supplementary Fig. 1.

## Immunoprecipitation

G-Sepharose beads (20 µl; P3296, Sigma-Aldrich) were incubated with 1.5 µg of FADD antibody (ab124812, Abcam) or 1 µg of ZBP1 antibody (AG-20B-0010-C100, Adipogen) per mg of protein lysate for 4 h at 4 °C with rotation. Cleared protein lysates were subsequently added to the bead–antibody complexes and incubated overnight at 4 °C with rotation. Beads were washed five times with wash buffer (150 mM NaCl, 150 mM Tris-HCl 1 M, pH 7.4, 5% glycerol, 0.5% Triton X-100), and bound proteins were eluted by boiling in 60 µl of 2× SDS loading dye for 5 min.

## Dermatitis scoring criteria

Skin lesions in $Casp8^{fl/fl}Tnfr1^{-/-}Zbp1^{-/-}K14$-$cre^{tg/WT}$, $Casp8^{fl/fl}Zbp1^{-/-}Tnfr1^{-/WT}K14$-$cre^{tg/WT}$, $Casp8^{fl/fl}Tnfr1^{-/-}Sting1^{fl/fl}K14$-$cre^{tg/WT}$, $Casp8^{fl/fl}Tnfr1^{-/-}Sting1^{fl/WT}K14$-$cre^{tg/WT}$ and $Casp8^{fl/fl}Tnfr1^{-/-}Sting1^{-/-}K14$-$cre^{tg/WT}$ mice were scored using an adjusted version of the scoring system previously described[48]. Two main clinical criteria were assessed to determine a total score: (1) the number of body regions presenting lesions and (2) the characteristics of each lesion. For the first criteria skin from six different body regions were assessed and appearance of lesions in each of these body regions generated a score of 1 (head = 1, neck = 1, back = 1, abdomen = 1, flank = 1 and tail = 1). For the second criteria, the characteristic of each lesion was defined by its severity. Punctuated small crusts received a score of 1, coalescent crusts received a score of 2 and ulceration received a score of 3. The sum of all individual scores according to these two was calculated to determine the total severity. Scoring of skin lesions was performed in a blinded manner by three independent researchers.

## Histological and immunohistochemical analysis of paraffin-embedded tissues

Tissues were collected, fixed and processed as previously described[26,51]. H&E staining, along with detection of p-MLKL(Ser345) and Cl-CASP3, were performed accordingly[26]. For immunodetection of ZBP1, STING, p-STAT1(Tyr701), CD3, Ly6G, pHH3-S10 (Ser10) and γH2AX-S139 (Ser139), the slides were retrieved in Tris-EDTA-based antigen-retrieval buffer, pH 9 (S236784-2, Agilent Technologies), for 10 min at 114−121 °C using the TintoRetriever Pressure Cooker. For F4/80, CD45, K6A, K10 and K14 staining, the slides were retrieved in sodium citrate buffer, pH 6 (C9999, Sigma-Aldrich), under the same conditions. For delicate tissues prone to detachment (skin from adult mice), antigen retrieval was performed at 60 °C in a water bath overnight. Primary antibodies were incubated at 4 °C overnight or at room temperature for 45 minutes (for ZBP1). The blocking steps, antibody detection procedures, counterstaining and mounting were performed as previously described[26]. For p-STAT1, Ly6G and F4/80, HRP-conjugated secondary antibody incubation was followed by TSA-biotin signal amplification (NEL700A001KT, Akoya Biosciences; 1:100) and then detection was performed using the Avidin-Biotin Complex-HRP amplification system (PK-6100, Vector Laboratories). TUNEL staining was performed using the DeadEnd Fluorometric TUNEL System (Promega, G3250) according to the manufacturer's instructions. A list of the antibodies used is provided in Supplementary Table 4.

## Immunocytochemistry and immunofluorescence

For DNA, mitochondrial and TFAM staining, cells were fixed in 4% methanol-free formaldehyde (Cell Signaling Technology, 47746) for 15 min at room temperature, quenched with 50 mM ammonium chloride in PBS and permeabilized with 0.1% Triton X-100 in PBS for 10 min at 37 °C. For ZBP1 and Z-DNA/Z-RNA staining, cells were fixed for 10 min and permeabilized with 0.4% Triton X-100 in PBS for 10 min at 37 °C. After blocking, the following primary antibodies diluted in blocking buffer were incubated at 4 °C overnight: mouse anti-ZBP1 (AG-20B-0010-C100, Adipogen, 1:100), anti-Z-DNA/Z-RNA (Z22) (Ab00783-23.0, Absolute Antibody, 1:200), anti-TFAM (ABE483, EMD Millipore; 1:200) and anti-DNA (CBL186, EMD Millipore, 1:100). After washing, cells were incubated with secondary antibodies (Thermo Fisher Scientific, 1:500), including, goat anti-mouse Alexa Fluor 488 (A11001), goat anti-rabbit Alexa Fluor Plus 594 (A32740), goat anti-rabbit Alexa Fluor Plus 488 (A32731) and goat anti-mouse Alexa Fluor Plus 594 (A32742). Cell plating, blocking, washing steps and mounting were performed as previously described[28]. For mitochondrial staining, cells were incubated with 200 nM MitoTracker Deep Red (M22426, Thermo Fisher Scientific) for 30 min at 37 °C before fixation. For nucleic acid degradation, cells were treated with DNase I or RNase A as previously described[22].

## Image analysis

Image acquisition, analysis and quantification of whole-tissue digital images were performed as previously described[26,51]. For mouse tissue confocal microscopy, three representative images per biological replicate ($n = 5$) were acquired and quantified in Fiji (ImageJ) with BioVoxxel Toolbox for Voronoi segmentation. DAPI-stained nuclei served as seeds to generate Voronoi cells approximating cell boundaries. A custom Fiji script identified positive cells by overlapping cell boundaries and target fluorophore signals. Positive cell counts from three images per replicate were pooled. Immunocytochemical images were manually segmented in Fiji, and the mean fluorescence intensity was quantified by rolling-ball background subtraction using a custom script. Co-localization of Z-NA and ZBP1 was assessed using the BIOP JACoP plugin after background subtraction. At least 10 images and 30 cells per staining condition were analysed. Data are presented as mean ± s.e.m. and plotted in GraphPad Prism. Statistical significance was determined by ordinary one-way or two-way ANOVA with Tukey's multiple-comparison test.

## Blood cell analysis

Whole blood cell count analysis was performed in peripheral blood using VETSCAN HM5 analyser (Zoetis) according to the manufacturer's instructions.

## RNA isolation from mouse tissue

RNA was extracted from skin tissue using the RNeasy Plus Mini kit (74136, Qiagen). Skin was weighted at 30 mg, snap-frozen in liquid nitrogen and stored at −80 °C before being lysed. Skin tissues were lysed in lysis buffer (Buffer RLT + 40 µl DTT 1 M per ml + 0.5% Reagent DX) according to the kit's manufacturer instructions in TissueLyser II (85300, Qiagen) using 7 mm stainless steel beads (69990, Qiagen). Extracted RNA was stored at −80 °C.

## RNA isolation from whole blood in human samples

Whole blood from patients with SAVI and healthy donors was collected either with Tempus Tubes (4342792, Applied Biosystems) or PAXgene Tubes (762165, Qiagen) and total RNA was isolated using the Tempus Spin RNA Isolation kit (4380204) or with PAXgene Blood RNA kit (Qiagen), according to manufacturer's instructions.

## RT−qPCR

cDNA was prepared using the Lunascript RT SuperMix kit (M3010, New England Biolabs). RT−qPCR was performed using Luna Universal qPCR Master Mix (M3003, New England Biolabs) in QuantStudio 5 (A34322, Thermo Fisher Scientific). The housekeeper genes *Gapdh* and *HPRT* were used for transcripts normalization in mice and patients with SAVI, respectively. The relative fold change in gene expression (that is, relative quantification) of the samples was calculated as $2^{-\Delta\Delta C_t}$ to the controls (littermates for mice and healthy individuals for patients with SAVI). The primer for *mZbp1* was obtained from Integrated DNA Technologies (Mm.PT.58.21951435). Sequences for all other primers used are listed in Supplementary Table 5.

## 3′ mRNA-seq libraries and sequencing

A total of 2 µg of RNA was provided to the Cologne Centre for Genomics (CCG) for 3′ mRNA-seq. Library preparation was performed using the Lexogen QuantSeq 3′ mRNA sequencing protocol. The resulting libraries were assessed for quality using a TapeStation system and quantified using qPCR. Sequencing was then carried out on the NovaSeq 6000 platform, using a single-read (SR) 1×100 bp sequencing strategy.

## Differential expression analysis

RNA-seq reads were aligned to the reference genome (hg38 or mm10) using STAR aligner, followed by transcript quantification with RSEM. For mouse data, technical batch effects from two sequencing runs were corrected using ComBat_seq (sva package). One sample per genotype was excluded to maintain consistency after a WT control sample showed abnormal upregulation of *Zbp1*. The removed samples were those that clustered the least with *Zbp1* expression in all other genotypes. An additional round of sequencing was performed to analyse the *Casp8[E-KO]Sting1[−/−]* in comparison to all of the other genotypes. The analysis was performed as independent experiment. For patient data, single-sample gene set enrichment analysis was performed using the gsva method (gsva package) with the hallmark gene sets. The nine samples with highest IFNγ scores were classified as IFN[high] samples and used for differential expression and pathway analyses (patient samples classified as IFN[high]: P1post1, P1post2, P1post3, P2post1, P2post2, P2post3, P2post5, P3post1, P3post3). Differential expression analysis was performed using DESeq2 (v.1.44.0). Volcano plots were generated using ggplot2 (v.3.5.1) and heat maps were created using heatmap.2 (gplots, v.3.2.0) with row-wise scaling on the DEseq2-normalized expression data. For RNA-seq data, statistical significance of differential expression was assessed using two-sided Wald tests implemented in the DESeq2 package, with Benjamini−Hochberg correction for multiple testing (adjusted $P < 0.05$).

## Pathway analysis

Pathway analysis was performed using Metascape[52] or the GSEA GUI tool (v.4.3.3). For Metascape, significantly upregulated genes ($q < 0.05$) were submitted to the web-tool (Express Analysis). Enrichment was calculated using Metascape's default cumulative hypergeometric test with Benjamini−Hochberg correction, and enrichment $q$ values were plotted as $-\log[q]$. Mouse samples were analysed using *Homo sapiens* as a reference species to enable human−mouse pathway comparison. For GSEA, DESeq2-normalized counts were used as the input and analysed using the default parameters with gene set permutation type. The gene set 'inflammatory cell death' was manually curated and contains the following genes: *TNF*, *TNFRSF1A*, *TNFRSF10*, *TNFRSF10B*, *RIPK1*, *FADD*, *CASP8*, *CFLAR*, *RIPK3*, *MLKL*, *CASP1*, *TNFSF10*, *FASLG*, *FAS*, *ZBP1*, *ADAR*, *TRIM25*, *EIF2AK2*, *IL1B*, *TLR4*, *BIRC2*, *RBCK1*, *TREX1*, *NLRC4*, *TLR5* and *NAIP*. The gene sets REACTOME_RIP_MEDIATED_NFKB_ACTIVATION_VIA_ZBP1 and REACTOME_ZBP1_DAI_MEDIATED_INDUCTION_OF_TYPE_I_IFNS were merged to give the customized ZBP1-mediated process geneset.

## Protein extraction from mouse skin tissue and mouse serum and human plasma isolation

Total protein was extracted from mouse skin tissue in cell lysis buffer 2 (895347, R&D Systems) in TissueLyser II (85300, Qiagen) using 7 mm

stainless steel beads (69990, Qiagen). Extracted proteins were stored at −80 °C. For human plasma isolation, whole blood was collected in EDTA blood collection tubes. Subsequently, the samples were centrifuged at 1,500g for 15 min, and the resultant supernatant (plasma) was carefully aspirated, aliquoted and stored at −80 °C for further analysis.

### Luminex assay
Cytokine analysis in skin protein lysates from mice was conducted using the Mouse Premixed Multi-Analyte Kit (R&D Systems) according to the manufacturer's instructions. For cytokine analysis in plasma from patients with SAVI and healthy donors, the corresponding human multi-analyte kit from R&D Systems was used.

### Flow cytometry analysis
Spleen, lung and thymus were dissected from 6–8-week-old mice. Lung digestion and single-cell suspensions were prepared as described previously[53]. Red blood cell lysis (1× RBC lysis buffer, BioLegend) was performed for 4 min on ice for lung and spleen samples; spleens for erythrocyte analysis and thymi were not subjected to RBC lysis. The samples were blocked with anti-mouse CD16/32 antibodies (101320, BioLegend, 1:100 dilution), washed with FACS buffer (PBS supplemented with 1% FBS and 5 mM EDTA), then stained with viability dye (65-0865-18, Thermo Fisher Scientific, 1:1,000) followed by surface marker staining in Brilliant Stain Buffer (566349, BD). All of the incubation steps were performed for 30 min at 4 °C. Data were acquired on the BD FACSymphony A3 system and analysed using FlowJo v.10.10 and GraphPad Prism. Statistical significance was determined using unpaired t-tests. A list of all of the antibodies used is provided in Supplementary Table 6. The gating strategy is included in Supplementary Fig. 9.

### Depletion of mtDNA and measurement of relative mtDNA levels
Immortalized MEFs were treated with 150 μM ddC or 10 μM IMT1B for 96 h, or with medium alone as a control. DNA was isolated using the DNeasy Blood & Tissue Kit (Qiagen) according to the manufacturer's protocol for cells. Relative mtDNA levels were measured using RT–qPCR with the Brilliant III Ultra-Fast SYBR QPCR Master Mix (Agilent Technologies) using different primers. qPCR was performed on the QuantStudio K FlexSystem (Applied Biosystems, Life Technologies). The samples were adjusted for total DNA content by *Actb*. Relative mtDNA levels were determined using the comparative ($2^{-\Delta\Delta C_t}$) quantification method. A list of the sequences for all of the primers used is provided in Supplementary Table 7.

### Extraction of the total RNA from MEFs
Total RNA was isolated from MEFs using TRIzol reagent (Life Technologies). MEFs were pelleted and resuspended in 1 ml of TRIZOL. After incubation of 5 min at room temperature, chloroform was added and subsequent phase separation was performed according to the manufacturer's instructions. The RNA pellet was resuspended in RNase/DNase-free $H_2O$ (Gibco).

### RT–PCR and RT–qPCR
The total RNA that was previously isolated was treated with DNase (New England Biolabs). Digested RNA (100 ng μl$^{-1}$ per sample) was reverse transcribed using the high-capacity reverse transcription kit (Applied Biosystems, Life Technologies). The generated cDNA was amplified using the Brilliant III Ultra-Fast SYBR QPCR Master Mix (Agilent Technologies) using different primers listed in Supplementary Table 8. qPCR was performed on the QuantStudio K FlexSystem (Applied Biosystems, Life Technologies). The samples were adjusted for total RNA content by *HPRT*. The relative expression of mRNAs was determined using the comparative ($2^{-\Delta\Delta C_t}$) quantification method.

### Ethics statement
The study involving the collection of the included patient samples was approved by the local Institutional Ethical Committee of Bambino Gesù Children's Hospital, Rome, Italy (PNRR-MR1-2022-12375873). Written informed consent to participate in this study was provided by the participants' legal guardian/next of kin. Clinical information relative to patient samples is listed in Supplementary Table 9. Patient data are available on request.

### Statistical analysis
Survival curves were compared using two-sided Gehan–Breslow–Wilcoxon tests. Data in graphs are shown as the mean ± s.e.m. unless otherwise indicated, with each dot representing one mouse or individual sample. For comparisons involving more than two groups, statistical significance was determined using one-way or two-way ANOVA followed by Tukey's or Šidák's multiple-comparison test as appropriate. In some cases, two-tailed unpaired t-tests were performed when comparing two groups, as specified in the figure legends. Flow cytometry data were analysed using two-way ANOVA or unpaired t-tests as indicated. P values are provided in the figures or figure legends; $P > 0.05$ was considered to be not significant. For cell viability assays, data represent the mean ± s.e.m. of at least two independent experiments, each performed with three technical replicates. The details on sample size, replicates and statistical tests are provided in each figure legend.

### Reporting summary
Further information on research design is available in the Nature Portfolio Reporting Summary linked to this article.

## Data availability
RNA-seq data for human patients are available on request. RNA-seq data for mouse tissues are available through the Gene Expression Omnibus of the National Institutes of Health under accession number GSE304254 (RNA-seq). Source data are provided with this paper.

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

**Acknowledgements** We thank the members of the Liccardi and Walczak laboratories as well as N. Peltzer and A. Annibaldi for discussions; the members of the CECAD imaging facility, particularly M. Babaki, P. Zentis and C. Jüngst, the staff at the Histology Core Facility of SFB 1403, especially E. Stade, and the staff of the CECAD in vivo Research Facility of the University of Cologne for their support. We thank M. Guschlbauer and the staff of the animal facilities of the medical faculty of the University of Cologne for their constant support and assistance. We thank M. Pasparakis, the CRC1403 and the CECAD animal facility for providing mouse models for initial experiments not included here. G.L. is funded by the Institute of Biochemistry I (IBC1) of the Medical Faculty of the University of Cologne, Köln Fortune, the CANcer TARgeting (CANTAR) network funded by the NRW-funded Research Programme Netzwerke 2021, two

Collaborative Research Centre (CRC) grants (SFB1399 project C06 (413326622); SFB1530 project A03 (455784452)) funded by the Deutsche Forschungsgemeinschaft (DFG) and is associated to the CRC SFB1403 also funded by the DFG; H.W. by the Alexander von Humboldt Foundation, a Wellcome Trust Investigator Award (214342/Z/18/Z), a Medical Research Council Grant (MR/S00811X/1), a Cancer Research UK Programme Grant (A27323), three CRC grants (SFB1399 project C06 (413326622); SFB1530 project A03 (455784452) and SFB1403 project A12 (414786233)) funded by the DFG and the CANcer TARgeting (CANTAR), funded by Netzwerke 2021; and G.P. and F.D.B. by the European Union, Next Generation EU, NRRP M6C2, Investment 2.1 Enhancement and strengthening of biomedical research in the NHS (PNRR-MR1-2022-12375873). Open access funding was provided by Universität zu Köln.

**Author contributions** G.L. conceived and led the study, developed the overarching research strategy and supervised all aspects of the work. G.L. and K.K. designed experiments. G.L., H.W. and K.K. discussed and interpreted the data. K.K. performed all in vivo experiments, all immunohistochemistry and immunofluorescence in mouse tissue and cultured cells and conducted all image acquisition and analysis. G.L., K.K., J.S., C.K. and D.B. performed in vitro experiments. G.L. optimized and performed all immunoprecipitations. J.S., C.K. and H.R.-K. provided support with mouse scoring and dissections. D.B. and K.K. performed cytokine analysis in mice. A.B.V., L.G. and M.P. analysed all RNA-seq data. F.L. performed and analysed all RT–qPCR data. K.K., J.S., D.B., C.K. and H.R.-K. collected organs and cells from N153 mice. J.S. and D.B. performed the flow cytometry experiment. A.M., J.S. and D.B. planned and analysed the flow cytometry experiment. K.K. and M.C. performed the mitochondrial DNA experiments, and discussed and analysed the data with A.T.; E.L., G.P., A.I. and F.D.B. collected data from patients with SAVI and provided mRNA and plasma samples from these patients. J.S. and D.B. processed and analysed samples from patients with SAVI. K.K., J.S. and D.B. prepared mRNA from mice. G.L. and K.K. prepared the figures. K.K., H.W. and G.L. wrote the manuscript.

**Funding** Open access funding provided by Universität zu Köln.

**Competing interests** The authors declare no competing interests.

**Additional information**
**Correspondence and requests for materials** should be addressed to Gianmaria Liccardi.

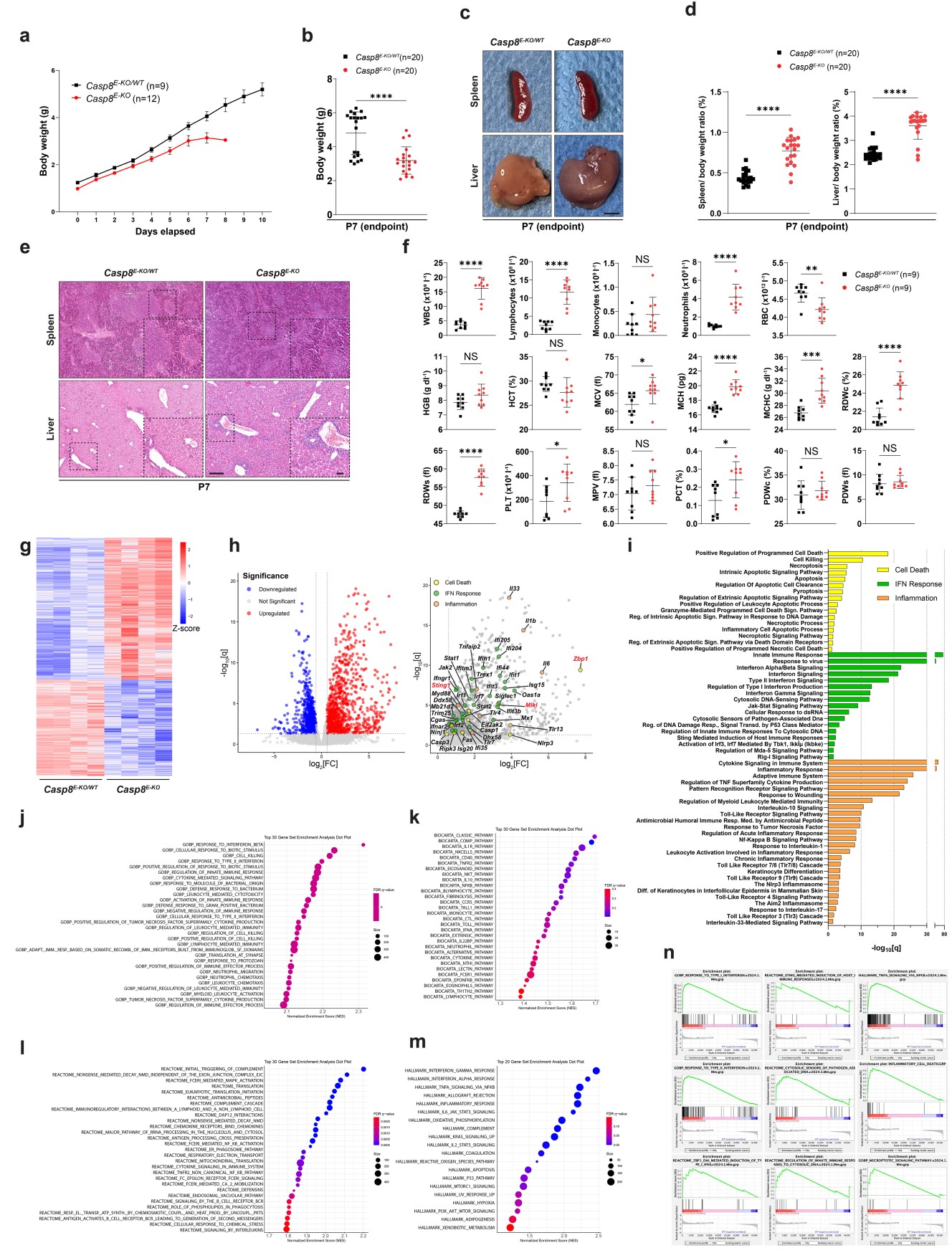

**Extended Data Fig. 1** | See next page for caption.

**Extended Data Fig. 1 | *Caspase-8* Loss Reprograms Keratinocytes to Induce ISGs, Necroptosis and DNA Sensing. a**, Graph of the trajectory of body weight changes in *n* mice of indicated genotypes. **b**, Graph of body weight of *n* mice of indicated genotypes at survival endpoint (P7). **c-e**, Representative images (**c**), graphs of organ-to-body-weight ratio (**d**) and representative images of tissue sections stained with H&E (**e**) of spleen and liver of *n* mice of indicated genotypes. Scale bars: 2 mm (spleen and liver images), 200 μm (main bright-field images) and 50 μm (insets). **f**, Peripheral blood analysis of indicated values in mice of indicated genotypes at survival endpoint. *n* = 9 per group. **g**, Heatmap of row-scaled expression values of differentially expressed genes in *Casp8^E-KO* versus *Casp8^E-KO/WT* skin (*p-adj* < 0.05). **h**, Volcano plots depicting differentially expressed genes in *Casp8^E-KO* versus *Casp8^E-KO/WT* skin at P5. Genes with |FC| > 1.5 and *p-adj* < 0.05 are displayed. **i**, Pathway enrichment analysis of upregulated genes in *Casp8^E-KO* versus *Casp8^E-KO/WT* skin (*p-adj* < 0.05, FC > 1), using Metascape. Bar plot shows −log$_{10}$(*q*) values for enriched pathways. **j-m**, Dot plots of Gene Set Enrichment Analysis (GSEA) for top 20 or 30 upregulated pathways in *Casp8^E-KO* versus *Casp8^E-KO* control skin. Gene Ontology Biological Process (GOBP) (**j**), BioCarta (**k**), Reactome (**l**) and Hallmark (**m**) databases. Dot size indicates gene set size; colour intensity represents −log$_{10}$(*p-adj*). **n**, Enrichment plots for representative pathways identified by GSEA. Black vertical lines show positions of pathway genes in the ranked gene list. In **a**, **b**, **d** and **f**, data are presented as mean ± s.e.m.; each dot represents one mouse; *P* values were calculated via two-tailed unpaired t-test; *$P$ ≤ 0.05, **$P$ ≤ 0.01, ***$P$ ≤ 0.001, ****$P$ ≤ 0.0001; NS, not significant ($P$ > 0.05). In **g**, **h**, differential gene expression analysis was performed using the Wald test as implemented in DESeq2, with *P* values adjusted for multiple testing using the Benjamini-Hochberg method (*p-adj*). MCV, mean corpuscular volume; MCHC, mean corpuscular haemoglobin concentration; RDWc, red cell distribution width (coefficient of variation); RDWs, red cell distribution width (standard deviation); PLT, platelets; MPV, mean platelet volume; PCT, plateletcrit; PDWc, platelet distribution width (coefficient of variation); PDWs, platelet distribution width (standard deviation).

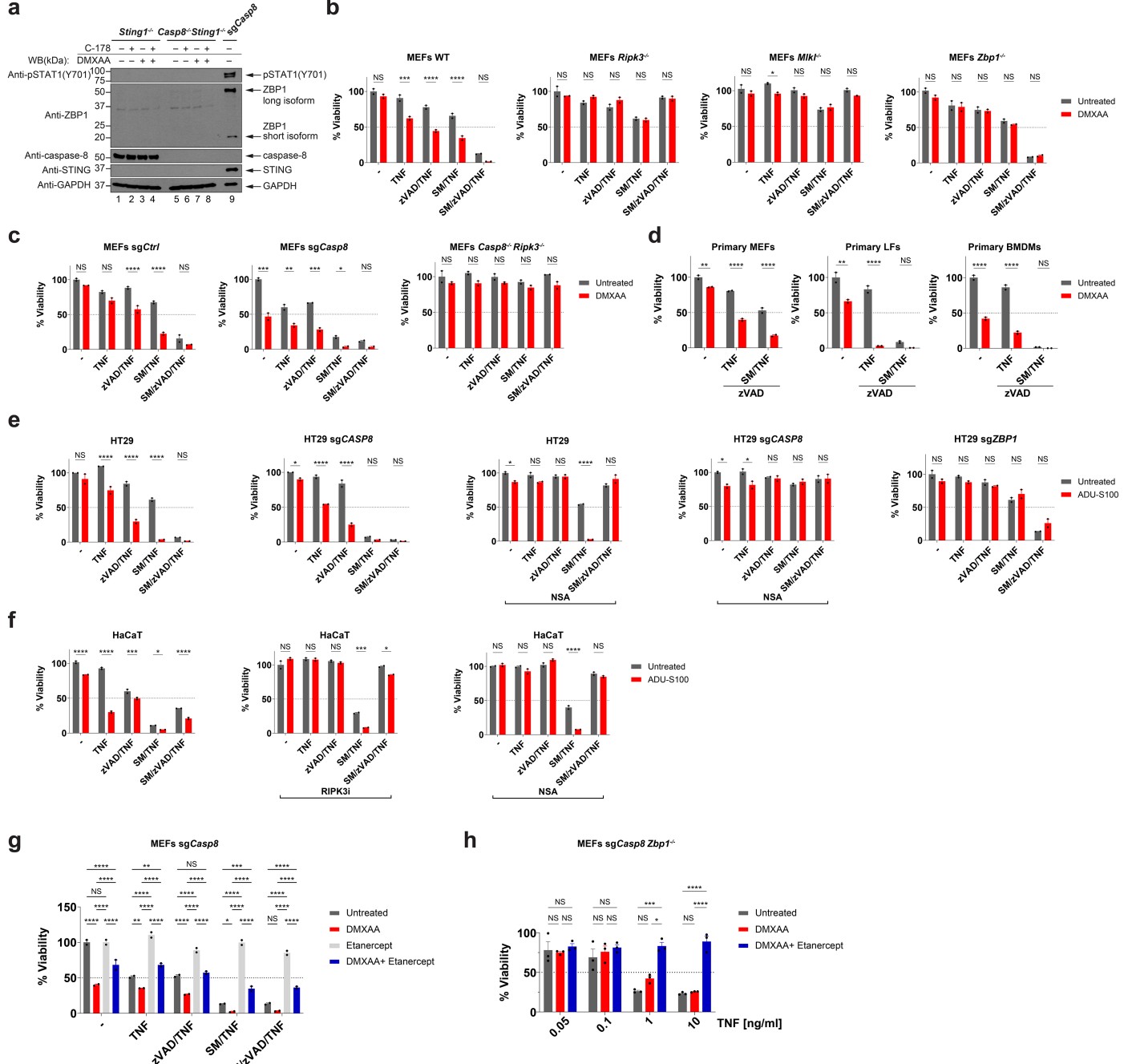

**Extended Data Fig. 2 | STING sensitizes to Necroptosis via ZBP1 Independently of TNFR1. a**, Immunoblot analysis with the indicated antibodies of lysates from immortalized *Sting1⁻/⁻* and *Casp8⁻/⁻ Sting1⁻/⁻* MEFs treated with C-178 (10 μM) and DMXAA (20 μg ml⁻¹) for 48 h; *Casp8⁻/⁻* MEFs were used as control. **b-d**, Relative cell viability of immortalized WT, *Ripk3⁻/⁻*, *Mlkl⁻/⁻* and *Zbp1⁻/⁻* MEFs (**b**), sgCtrl, *Casp8⁻/⁻*, *Casp8⁻/⁻;Ripk3⁻/⁻* MEFs (**c**) and primary WT MEFs, lung fibroblasts (LFs) and bone-marrow-derived macrophages (BMDMs) assessed by CellTiter-Glo assay, pre-treated with DMXAA overnight, followed by treatment with recombinant TNF (100 ng ml⁻¹), Z-VAD-FMK (20 μM) and/or BV6 (2 μM) for 48 h. **e**, **f**, Relative cell viability of HT-29, HT-29 *CASP8⁻/⁻*, HT-29 *ZBP1⁻/⁻* (**e**) and HaCaT (**f**) cells assessed by CellTiter-Glo assay, pre-treated with ADU-S100 (10 μM) alone or in combination with NSA (1 μM) or GSK'872 (RIPK3i, 10 mM) overnight,

followed by treatment with recombinant TNF, Z-VAD-FMK and/or BV6 for 48 h. **g**, Relative cell viability of *Casp8⁻/⁻* MEFs assessed by CellTiter-Glo assay, pre-treated with DMXAA and/or etanercept (50 μg ml⁻¹) overnight, followed by treatment with recombinant TNF, Z-VAD-FMK and/or BV6 for 48 h. **h**, Relative cell viability of immortalized *Casp8⁻/⁻;Zbp1⁻/⁻* MEFs assessed by CellTiter-Glo assay, pre-treated with DMXAA and/or etanercept overnight, followed by titration with recombinant TNF (0.05, 0.1, 1 and 10 ng ml⁻¹) for 48 h. In **b-h**, data are presented as mean ± s.e.m.; each dot represents an independent experiment each containing three technical replicates; *P* values were calculated via two-way Anova followed by Šidák multiple comparisons test; *P ≤ 0.05, **P ≤ 0.01, ***P ≤ 0.001, ****P ≤ 0.0001; NS, not significant (P > 0.05).

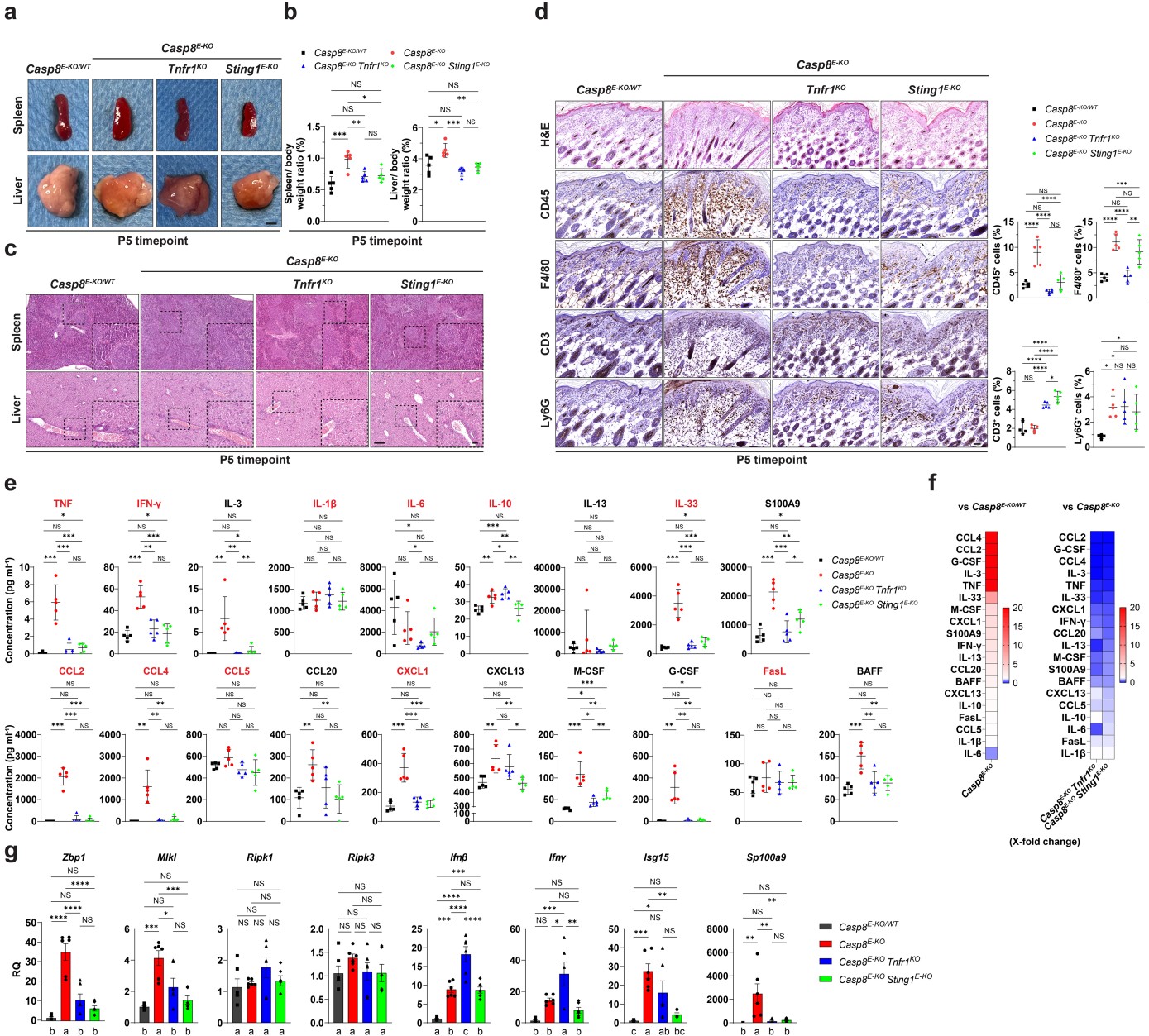

**Extended Data Fig. 3 | TNFR1 or STING Loss Reveals Distinct Immune and Transcriptional Responses. a-c,** Representative images (**a**), graphs of organ-to-body-weight ratio (**b**) and representative images of tissue sections stained with H&E (**c**) of spleen and liver of P5 mice of indicated genotypes. *n* = 5 per group. Scale bars: 2 mm (spleen and liver images), 200 μm (main bright-field images) and 50 μm (insets). **d,** Representative images of consecutive skin sections from P5 mice of indicated genotypes stained with H&E and the indicated antibodies. *n* = 5 per group. Scale bar: 50 μm. Graphs show immunostaining quantification. **e,** Graphs of cytokine levels (pg ml⁻¹) of skin homogenates from P5 mice of indicated genotypes, analysed via Luminex-Multiplex assay for the indicated targets. *n* = 5 per group. Cytokines highlighted in red denote those implicated in necroptosis, as detailed in Supplementary Table 1. Data are presented as mean ± s.e.m.; each dot represents one mouse; *P* values were

calculated via two-tailed t-test. **f,** Heatmaps showing X-fold changes in cytokine levels relative to *Casp8^E-KO/WT* control and *Casp8^E-KO* respectively at P5. **g,** RT–qPCR analysis of mRNA expression of the indicated genes in skin from P5 mice of indicated genotypes. Graphs represent relative quantification (log₂RQ) normalized to *Gapdh*, with expression levels shown relative to *Casp8^E-KO/WT*. Data represent three biological replicates from two independent experiments. Different letters indicate statistically significant differences between groups, determined by one-way ANOVA followed by Tukey's HSD test (FDR = 0.05). In **b** and **d**, data in graphs are presented as mean ± s.e.m.; each dot represents one mouse; *P* values were calculated via two-way Anova followed by Tukey's multiple comparisons test; *P* ≤ 0.05, **P* ≤ 0.01, ***P* ≤ 0.001, ****P* ≤ 0.0001; NS, not significant (*P* > 0.05).

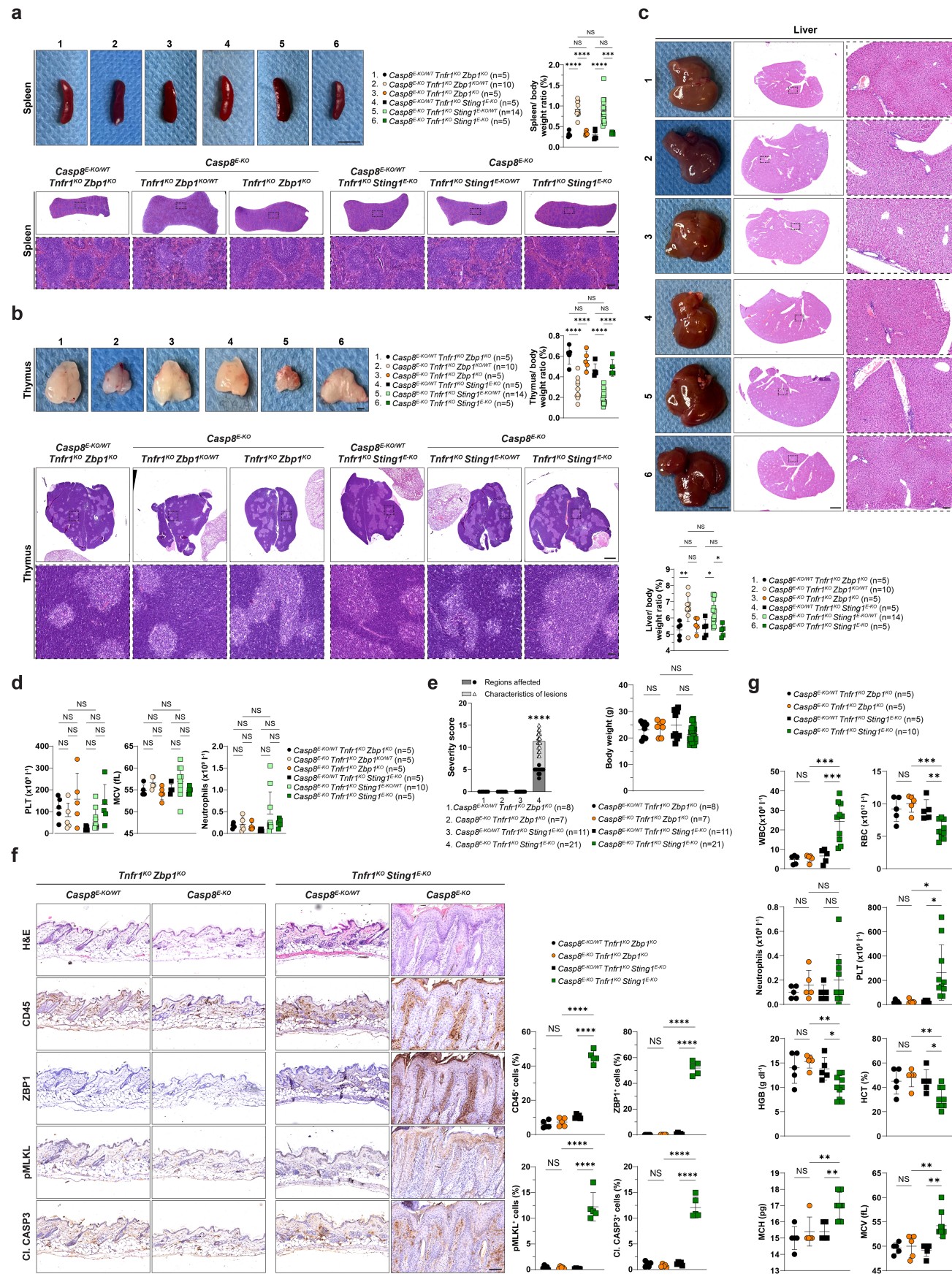

**Extended Data Fig. 4** | See next page for caption.

**Extended Data Fig. 4 | *Sting1^{E-KO}* Delays Lethal Inflammation in *Casp8^{E-KO}* *Tnfr1^{-/-}* Mice. a-c,** Representative images, graphs of organ-to-body-weight ratio and representative images of tissue sections stained with H&E of spleen (**a**), thymus (**b**) and liver (**c**) of 4-week-old mice of indicated genotypes. *n* values are indicated. Scale bars: 5 mm (spleen and liver images), 1 mm (thymus images), 1 mm (main bright-field images) and 100 μm (insets). **d,** Neutrophil, PLT and MCV values in peripheral blood from 4-week-old mice of indicated genotypes. *n* values are indicated. **e,** Graphs of dermatitis severity score and body weight of 12-14-week-old mice of indicated genotypes. *n* values are indicated. Control mice included age-matched *Casp8^{E-KO/WT}; Tnfr1^{KO}; Zbp1^{KO}* and *Casp8^{E-KO/WT};*

*Tnfr1^{KO}; Sting1^{E-KO}* littermates. **f,** Representative images of consecutive skin sections from 12-14-week-old mice of indicated genotypes stained with H&E and the indicated antibodies *n* = 5 per group. Scale bar: 100 μm. Graphs show immunostaining quantification. **g,** Peripheral blood analysis of indicated values in 12-14-week-old mice of indicated genotypes. *n* values are indicated. In **a-g**, data in graphs are presented as mean ± s.e.m.; each dot represents one mouse; *P* values were calculated via two-way Anova followed by Tukey's multiple comparisons test; *$P \leq 0.05$, **$P \leq 0.01$, ***$P \leq 0.001$, ****$P \leq 0.0001$; NS, not significant ($P > 0.05$).

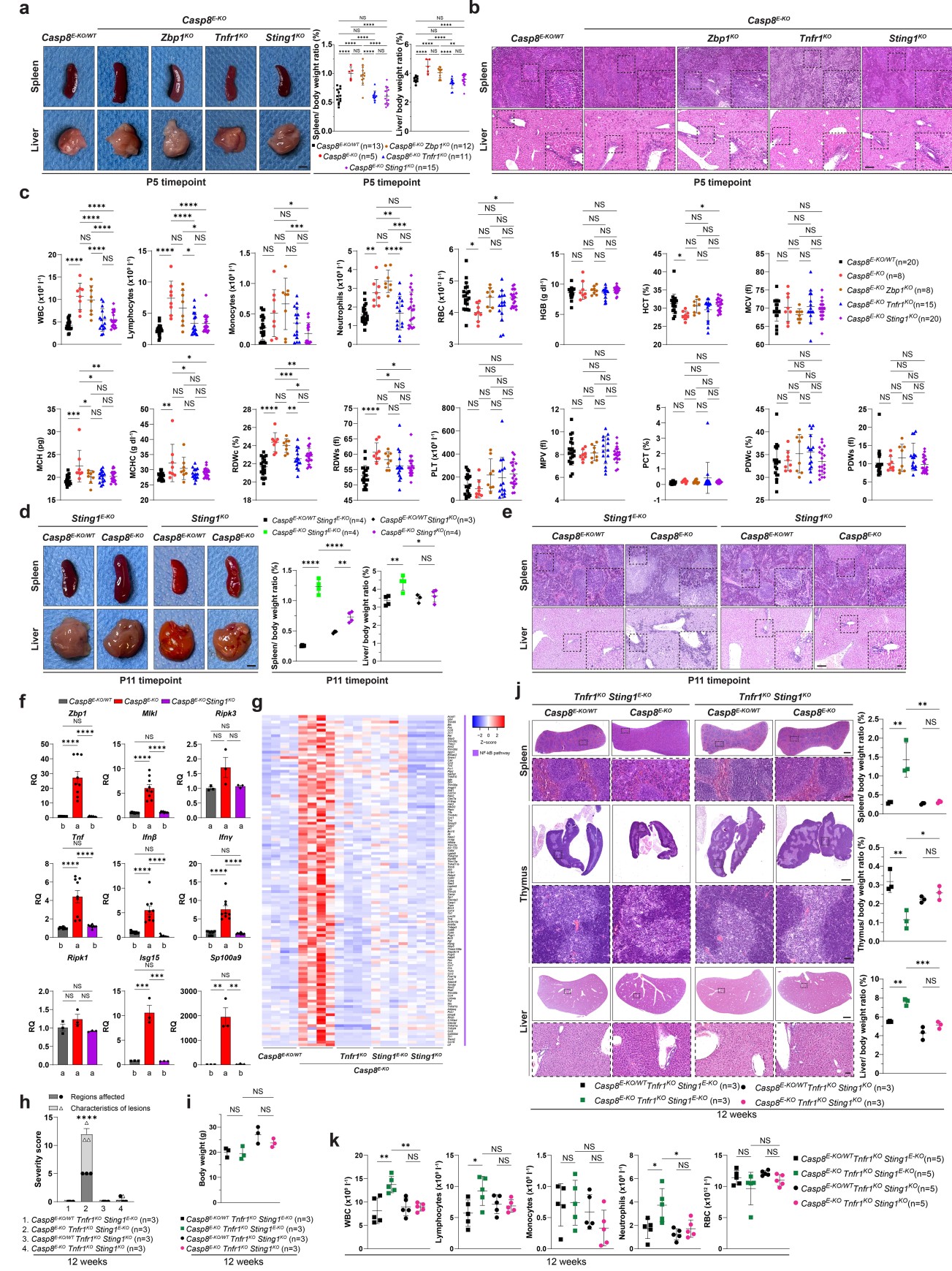

**Extended Data Fig. 5** | See next page for caption.

**Extended Data Fig. 5 | *Sting1⁻ᐟ⁻* inhibits early necroptosis and immune dysregulation in Casp8*ᴱ⁻ᴷᴼ*. a**, **b**, Representative images, graphs of organ-to-body-weight ratio (**a**) and H&E-stained tissue sections (**b**) of spleen and liver from P5 mice of indicated genotypes. *n* values are indicated. Scale bars: 2 mm (spleen and liver images), 200 μm (main bright-field images) and 50 μm (insets). **c**, Peripheral blood analysis of indicated values in P5 mice of indicated genotypes. *n* values are indicated. **d**, **e**, Representative images, graphs of organ-to-body-weight ratio (**d**) and H&E-stained tissue sections (**e**) of spleen and liver from *n* mice of indicated genotypes at the survival endpoint of *Casp8ᴱ⁻ᴷᴼ; Sting1ᴱ⁻ᴷᴼ* mice (P11). Scale bars: 2 mm (spleen and liver images), 200 μm (main bright-field images) and 50 μm (insets). **f**, RT–qPCR analysis of gene expression in skin from P5 mice of indicated genotypes. Graphs represent relative quantification (log₂ RQ) normalized to *Gapdh*, relative to *Casp8ᴱ⁻ᴷᴼ/ᵂᵀ*. Data represent three biological replicates from one or two independent experiments as indicated. Different letters indicate statistically significant differences between groups, determined by one-way ANOVA followed by Tukey's HSD test (FDR = 0.05). **g**, Heatmap of NF-κB-associated genes downregulated in *Casp8ᴱ⁻ᴷᴼ; Tnfr1ᴷᴼ*, *Casp8ᴱ⁻ᴷᴼ/ᵂᵀ; Sting1ᴷᴼ*, or both, compared to *Casp8ᴱ⁻ᴷᴼ* mice. Expression values are DESeq2-normalized, row-scaled, and colour-coded by intensity (red, high; blue, low). Differential expression was determined using the Wald test in DESeq2 with Benjamini–Hochberg adjusted *p*-values (*p-adj* < 0.05). **h**-**k**, Graphs of dermatitis score (**h**) and body weight (**i**), representative images of indicated H&E-stained tissue sections with corresponding graphs of organ-to-body-weight ratio (**j**), and peripheral blood analysis of indicated values (**k**) in 12-week-old mice of indicated genotypes. *n* values are indicated. Scale bars: 1 mm (main images) and 100 μm (insets). In **a**, **c**, **d** and **h**-**k**, data in graphs are presented as mean ± s.e.m.; each dot represents one mouse; *P* values were calculated via two-way Anova followed by Tukey's multiple comparisons test; *$P \leq 0.05$, **$P \leq 0.01$, ***$P \leq 0.001$, ****$P \leq 0.0001$; NS, not significant ($P > 0.05$).

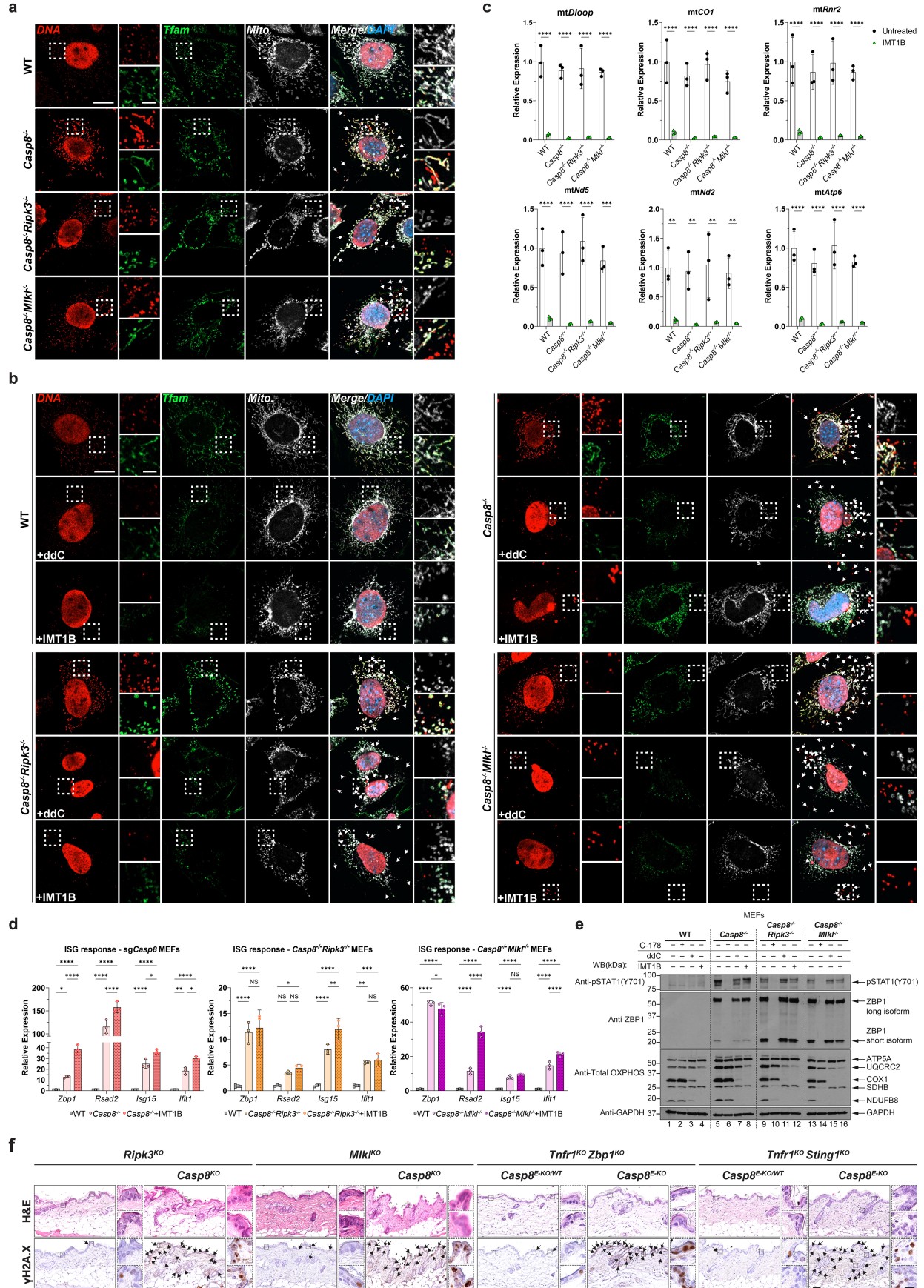

**Extended Data Fig. 6 |** See next page for caption.

**Extended Data Fig. 6 | STING Activation is not mediated by Mitochondrial DNA. a**, **b**, Confocal images of indicated MEFs stained for DNA (Alexa Fluor 594, red), TFAM (Alexa Fluor 488, green), mitochondria (MitoTracker Deep Red) and nuclei (DAPI, blue) untreated (**a**, **b**) or following 96 h treatment with ddC (150 µM) or IMT1B (10 µM) (**b**). Insets highlight magnified regions. Arrows mark DNA foci lacking TFAM and MitoTracker signals. Images represent single optical slices (0.4 µm). n ≥ 20 cells/condition. Scale bars: 10 µm and 1 µm (insets). **c**, **d** RT–qPCR analysis of indicated mitochondrial genes to assess mtDNA depletion (**c**), and of indicated ISGs to assess IFN response (**d**), in the indicated MEFs. Data are shown as relative expression ($\log_2$) to untreated WT control normalized to *beta-Actin*; graphs represent mean ± s.e.m. $n = 3$ technical replicates; $P$ values were calculated via two-way Anova followed by Tukey's multiple comparisons test; $*P \leq 0.05$, $**P \leq 0.01$, $***P \leq 0.001$, $****P \leq 0.0001$; NS, not significant ($P > 0.05$). **e**, Immunoblot analysis of lysates from the indicated MEFs treated with C-178 (10 µM), ddC, or IMT1B as above for 96 h. **f**, Representative images of consecutive skin sections from 12-4-week-old mice of indicated genotypes stained with H&E and the indicated antibodies. $n = 5$ per group. Arrows indicate γH2A.X positive areas/cells. Scale bars: 100 µm (main images) and 10 µm (insets).

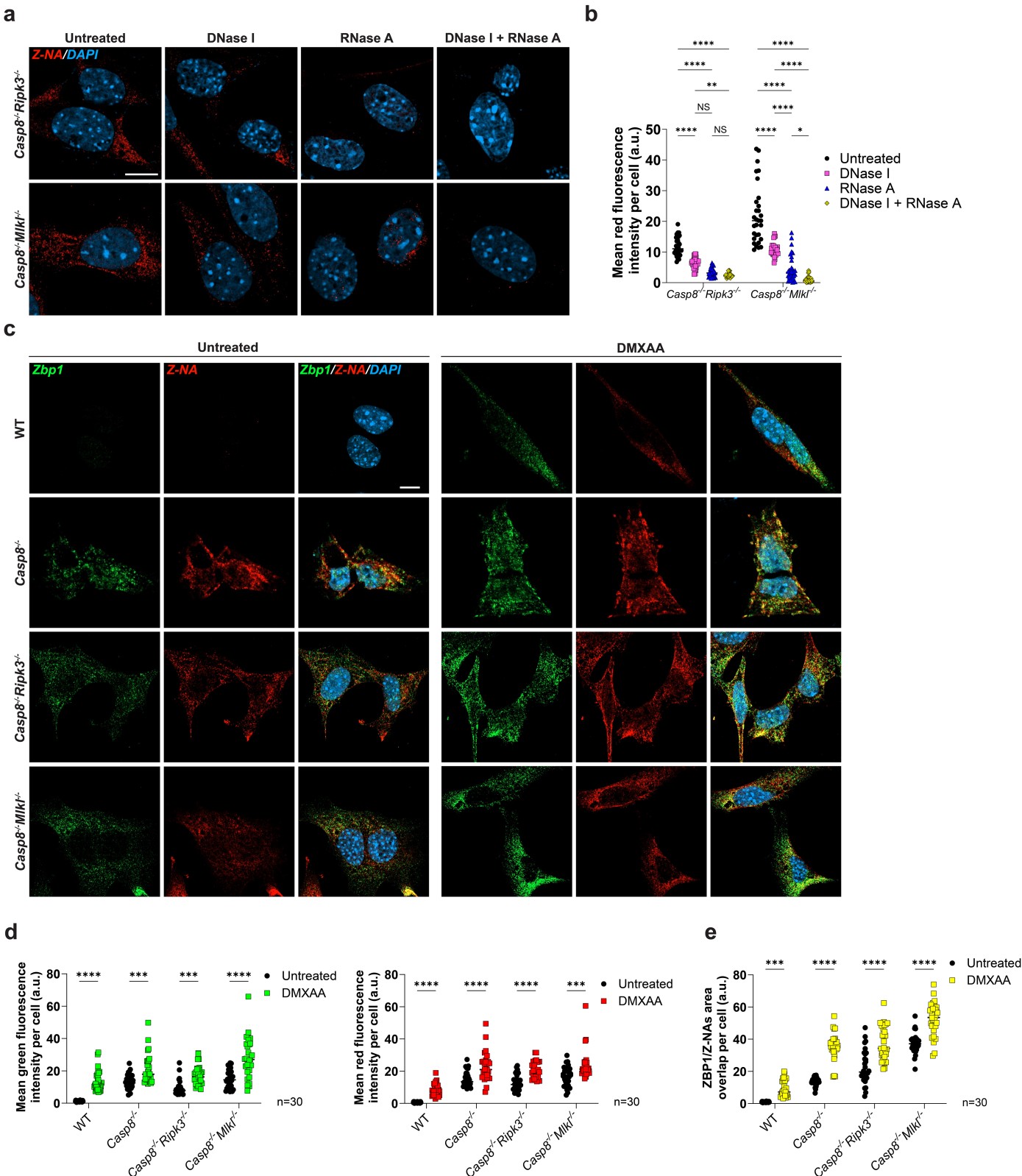

**Extended Data Fig. 7 | STING activation enhances ZBP1-Associated Z-NA Accumulation. a**, **b**, Confocal images of indicated MEFs stained for Z-NA (Alexa Fluor 594, red) and nuclei (DAPI, blue) under basal conditions or following treatment with DNase I (25 U mL⁻¹) and/or RNase A (1 mg mL⁻¹) (**a**) and quantification of mean Z-NA (red) fluorescence intensity (**b**). **c-e**, Confocal images of indicated MEFs stained for Z-NA (Alexa Fluor 594, red), ZBP1 (Alexa Fluor 488, green) and nuclei (DAPI, blue) untreated or following 48 h treatment

with DMXAA (20 µg ml⁻¹) (**c**), quantification of ZBP1 (green) and Z-NA fluorescent intensity (red) (**d**) and their area of overlap (**e**). Images represent single optical slices (0.4 µm). Scale bar: 10 µm. In **b**, **d** and **e**, data are presented as mean ± s.e.m.; each dot represents one cell and 30 cells/condition were analysed via two-way Anova followed by Tukey's multiple comparisons test; *$P \le 0.05$, **$P \le 0.01$, ***$P \le 0.001$, ****$P \le 0.0001$; NS, not significant ($P > 0.05$).

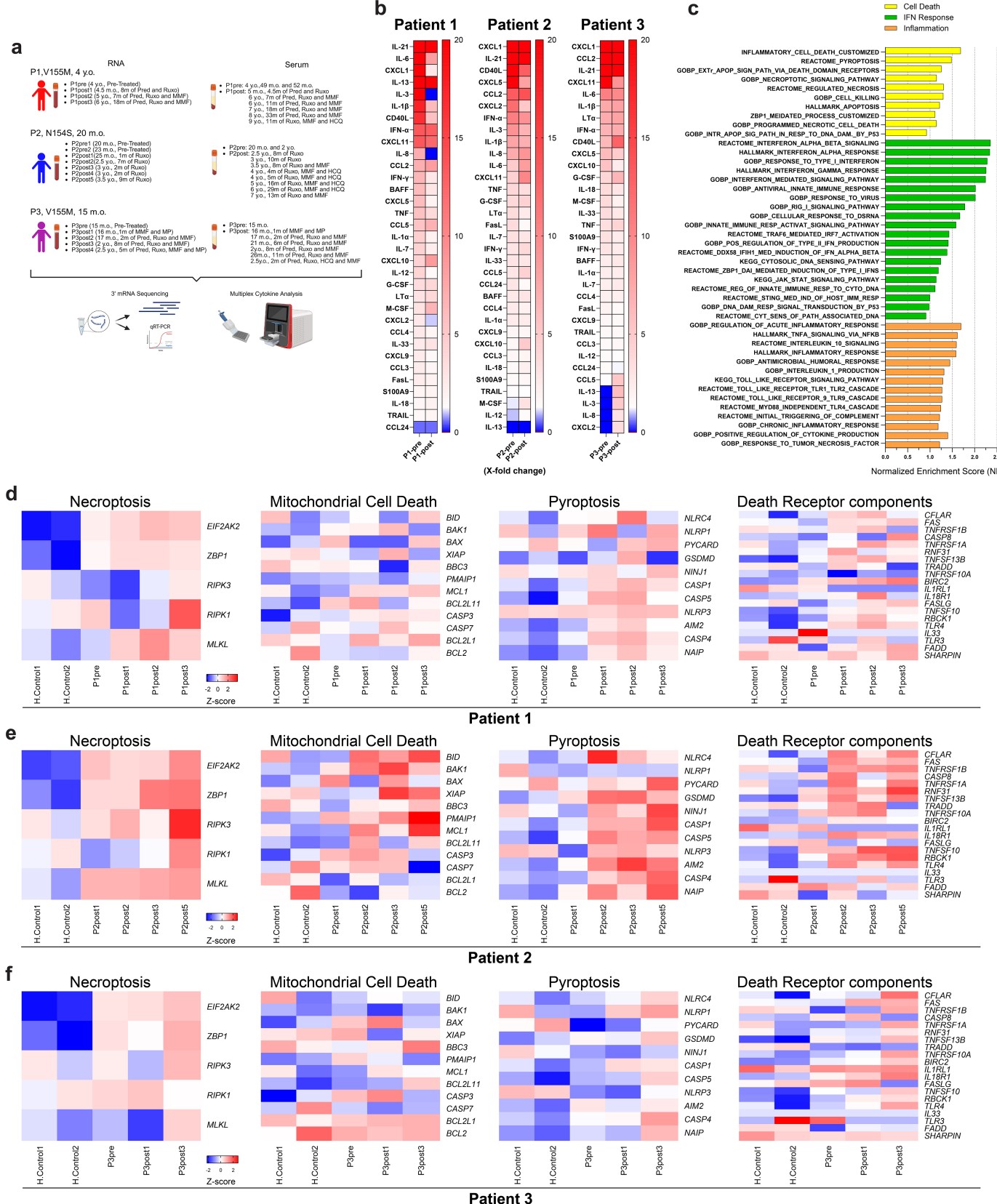

**Extended Data Fig. 8 |** See next page for caption.

**Extended Data Fig. 8 | SAVI longitudinal Samples Show Inflammatory IFN and Necroptosis Signatures. a**, Schematic representation of SAVI patient samples, illustrating patient mutations, sample types, and timepoints of sample collection during the treatment course. Notation: P(x)post/pre(y), where x = patient number, y = sample number, pre = pre-treatment, and post = post-treatment, Pred: Prednisone, MTX: Methotrexate, Ruxo: Ruxolitinib, MMF: Mycophenolate, HCQ: Hydroxychloroquine and MP: Metilprednisolone. This image was created in BioRender. Saggau, J. (2025) https://BioRender. com/9yrgm6e. **b**, Heatmaps showing X-fold changes in cytokine levels relative to healthy controls for each patient. **c**, Pathway enrichment analysis bar plot from GSEA analysis for Gene Ontology Biological Process (GOBP), Hallmark, Reactome, and KEGG databases, as well as manually curated gene sets for inflammatory cell death and ZBP1-mediated processes in high-IFN patient samples. **d-f**, Heatmaps depicting DEseq2-normalized expression levels of selected genes involved in necroptosis, mitochondrial cell death, pyroptosis, and death receptor components in IFN-high samples from Patient 1 (**d**), Patient 2 (**e**), and Patient 3 (**f**) and healthy donors. Expression values are DESeq2-normalized, row-scaled, and colour-coded by intensity (red, high; blue, low). Differential expression was determined using the Wald test in DESeq2 with Benjamini–Hochberg adjusted *P*-values (p-adj < 0.05).

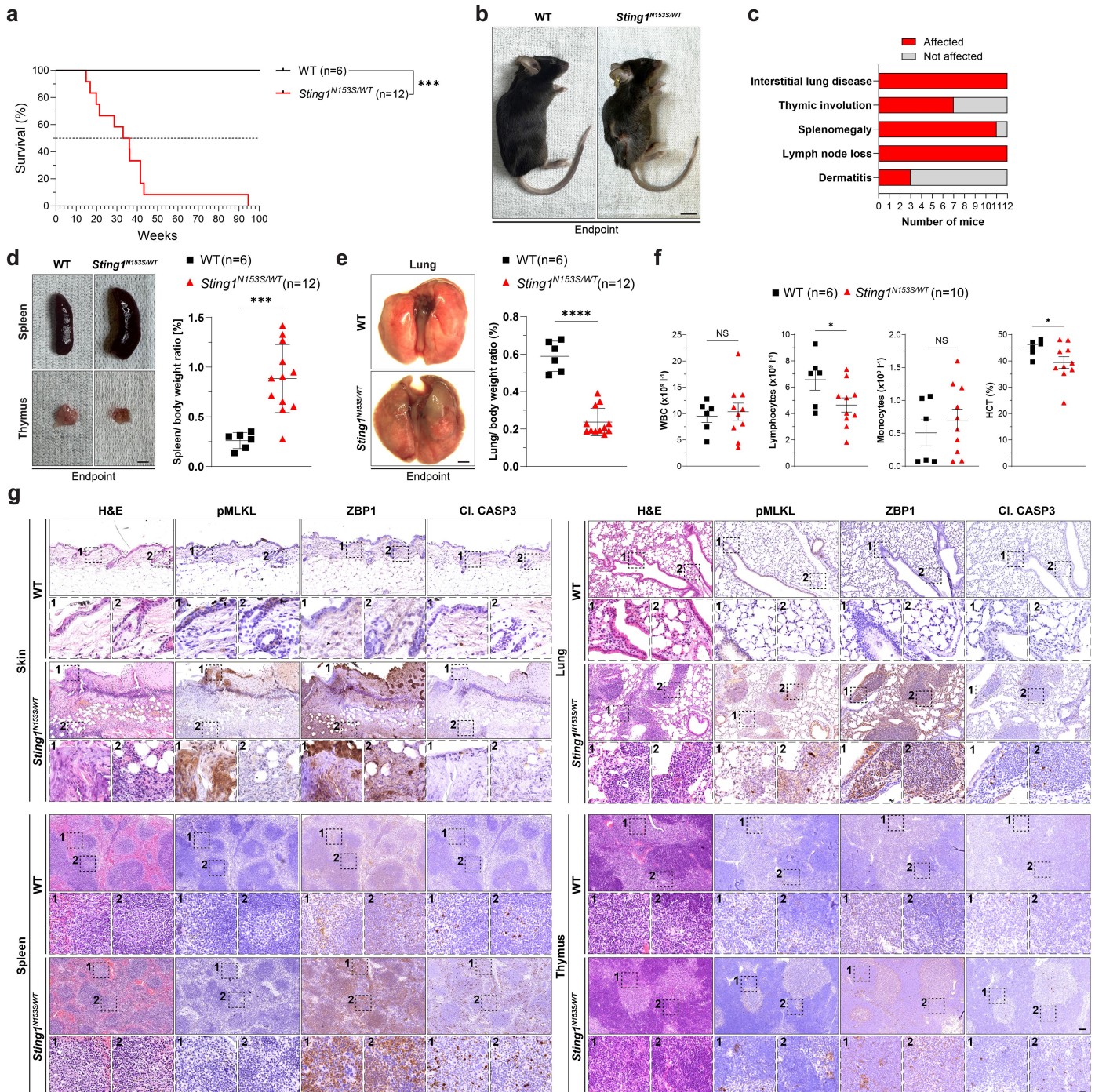

**Extended Data Fig. 9 | STAIN promotes Systemic Inflammation in SAVI mice. a**, Kaplan–Meier survival curves of mice of indicated genotypes. *n* values are indicated. *p* values were calculated by two-sided Gehan–Breslow–Wilcoxon test. **b**-**e**, Representative images (**b**), bar plot showing the distribution of SAVI-associated clinical manifestations (**c**) and representative images and graphs of organ-to-body-weight ratio of spleen, thymus (**d**) and lung (**e**) of *Sting1^{N153S/WT}* mice at survival endpoint (*n* indicated. Examined lymph nodes include axillary, inguinal, and submandibular). Scale bars: 3 mm (spleen and thymus images) and 2 mm (lung images). **f**, Peripheral blood analysis of

indicated values in endpoint *Sting1^{N153S/WT}* mice. *n* values are indicated. **g**, Representative images of consecutive skin, lung, spleen and thymus sections from *Sting1^{N153S/WT}* at survival endpoint stained with H&E and the indicated antibodies. *n* = 5 per group. Scale bars: 100 µm (main images) and 20 µm (insets). In **a**-**g**, age-matched WT littermates were used as controls. In **d**-**f**, data are presented as mean ± s.e.m.; each dot represents one mouse; *P* values were calculated via two-tailed unpaired t-test; *$P \leq 0.05$, ***$P \leq 0.001$, ****$P \leq 0.0001$; NS, not significant ($P > 0.05$).

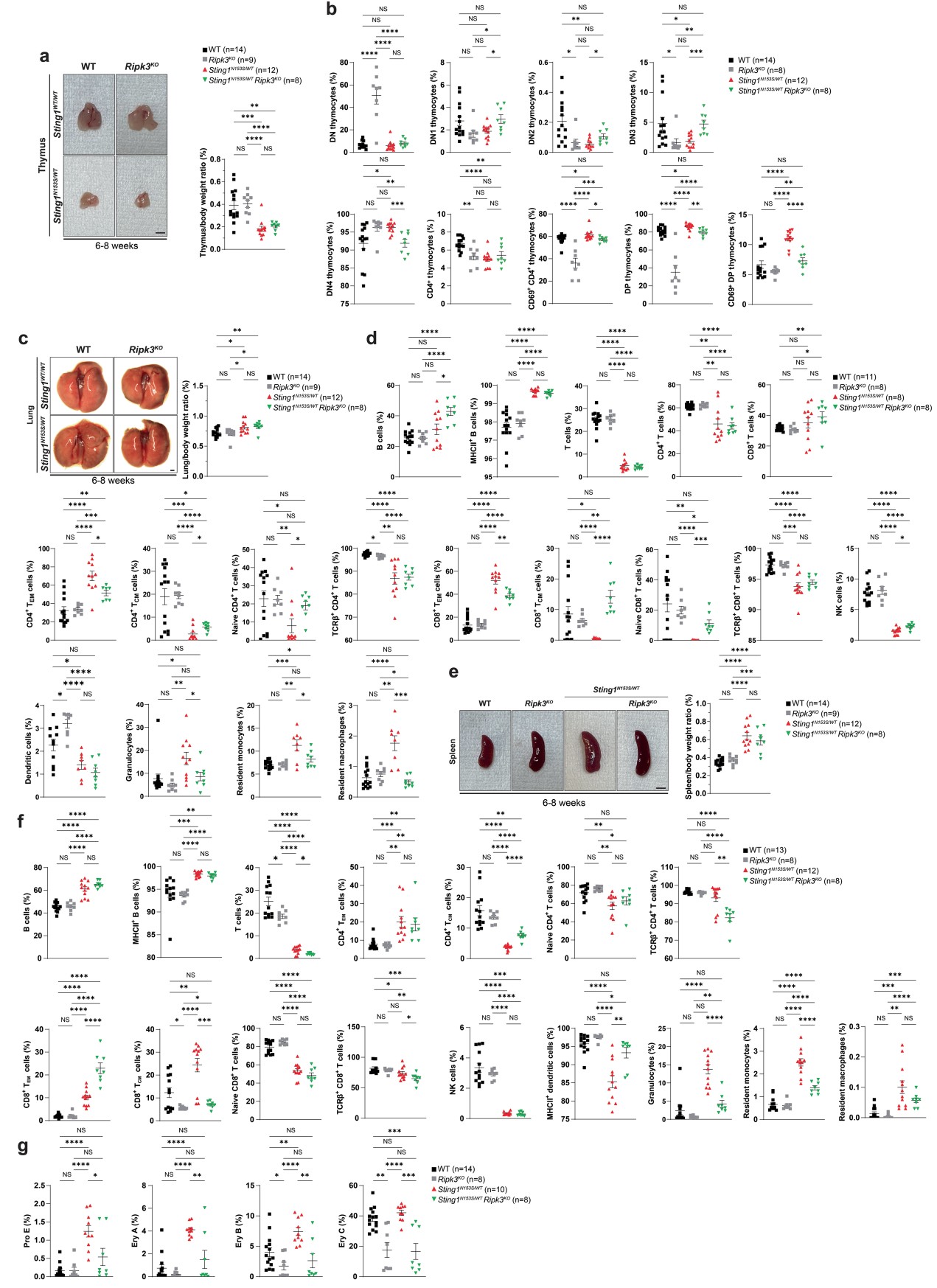

**Extended Data Fig. 10** | See next page for caption.

**Extended Data Fig. 10 | STAIN promotes Immune Depletion in SAVI mice.**
**a**, **b**, Representative images, organ-to-body-weight ratio (**a**) and flow cytometric thymocyte phenotyping (**b**) of thymi from 6-8-week-old mice of indicated genotypes. *n* values are indicated. CD4⁻CD8⁻ (double negative; DN), CD4⁺CD8⁺ (double positive; DP) and CD4⁺ thymocytes are depicted as percentage of viable cells. All other populations were plotted as percentage of their parental populations. **c**, **d**, Representative images, organ-to-body-weight ratio (**c**) and flow cytometric phenotyping (**d**) of lungs from 6-8-week-old mice of indicated genotypes. *n* values are indicated. Immune cells are depicted as percentage of CD45⁺ cells and their respective sub-populations are shown as percentage of parental populations. **e-g**, Representative images, organ-to-body-weight ratio (**e**) and flow cytometric immune cell phenotyping (**f**) and erythrocyte phenotyping (**g**) of spleens of 6-8-week-old mice of the indicated genotypes. *n* values are indicated. Immune cell types are depicted as percentage of CD45⁺ cells, their respective sub-populations are shown as percentage of parental populations and erythrocytes were plotted as percentage of live cells. In **a-g**, age-matched WT and *Ripk3*^KO littermates were used as controls,;data are presented as mean ± s.e.m.; each dot represents one mouse; *P* values were calculated via two-way Anova followed by Tukey's multiple comparisons test (**a**, **c** and **e**) or via two-tailed unpaired t-test (**b**, **d**, **f** and **g**); *$P \leq 0.05$, **$P \leq 0.01$, ***$P \leq 0.001$, ****$P \leq 0.0001$; NS, not significant ($P > 0.05$). The gating strategy for cell populations shown in **b**,**d**,**f** and **g** is provided in Supplementary Fig. 9.

# Reporting Summary

## Statistics

For all statistical analyses, confirm that the following items are present in the figure legend, table legend, main text, or Methods section.

| n/a | Confirmed | |
|---|---|---|
| ☐ | ☒ | The exact sample size (*n*) for each experimental group/condition, given as a discrete number and unit of measurement |
| ☐ | ☒ | A statement on whether measurements were taken from distinct samples or whether the same sample was measured repeatedly |
| ☐ | ☒ | The statistical test(s) used AND whether they are one- or two-sided<br>*Only common tests should be described solely by name; describe more complex techniques in the Methods section.* |
| ☐ | ☒ | A description of all covariates tested |
| ☐ | ☒ | A description of any assumptions or corrections, such as tests of normality and adjustment for multiple comparisons |
| ☐ | ☒ | A full description of the statistical parameters including central tendency (e.g. means) or other basic estimates (e.g. regression coefficient) AND variation (e.g. standard deviation) or associated estimates of uncertainty (e.g. confidence intervals) |
| ☐ | ☒ | For null hypothesis testing, the test statistic (e.g. *F*, *t*, *r*) with confidence intervals, effect sizes, degrees of freedom and *P* value noted<br>*Give P values as exact values whenever suitable.* |
| ☒ | ☐ | For Bayesian analysis, information on the choice of priors and Markov chain Monte Carlo settings |
| ☒ | ☐ | For hierarchical and complex designs, identification of the appropriate level for tests and full reporting of outcomes |
| ☒ | ☐ | Estimates of effect sizes (e.g. Cohen's *d*, Pearson's *r*), indicating how they were calculated |

*Our web collection on statistics for biologists contains articles on many of the points above.*

## Software and code

Policy information about availability of computer code

| Data collection | n/a |
|---|---|
| Data analysis | n/a |

For manuscripts utilizing custom algorithms or software that are central to the research but not yet described in published literature, software must be made available to editors and reviewers. We strongly encourage code deposition in a community repository (e.g. GitHub). See the Nature Portfolio guidelines for submitting code & software for further information.

## Data

Policy information about availability of data

All manuscripts must include a data availability statement. This statement should provide the following information, where applicable:
- Accession codes, unique identifiers, or web links for publicly available datasets
- A description of any restrictions on data availability
- For clinical datasets or third party data, please ensure that the statement adheres to our policy

Sequencing Data from patient samples are available upon request. Sequencing data relative to the mouse model as indicated in the manuscript are deposited. Accession number is available in the Data availability paragraph contained in the Material and Methods section.

# Research involving human participants, their data, or biological material

Policy information about studies with [human participants or human data](). See also policy information about [sex, gender (identity/presentation), and sexual orientation]() and [race, ethnicity and racism]().

| | |
|---|---|
| Reporting on sex and gender | 3 female patiens |
| Reporting on race, ethnicity, or other socially relevant groupings | Caucasian White |
| Population characteristics | N/A no clinical trial performed |
| Recruitment | Patients were recruited on the basis of their positive screening for STING activating mutations |
| Ethics oversight | The study was approved by the Ethics Committee of Bambino Gesù Children's Hospital (protocol number 3086 ) and was conducted in accordance with the Declaration of Helsinki. Written informed consent was obtained from all participants and/or their parents or legal guardians. |

Note that full information on the approval of the study protocol must also be provided in the manuscript.

# Field-specific reporting

Please select the one below that is the best fit for your research. If you are not sure, read the appropriate sections before making your selection.

☒ Life sciences          ☐ Behavioural & social sciences          ☐ Ecological, evolutionary & environmental sciences

For a reference copy of the document with all sections, see [nature.com/documents/nr-reporting-summary-flat.pdf]()

# Life sciences study design

All studies must disclose on these points even when the disclosure is negative.

| | |
|---|---|
| Sample size | All experiments were repeated at least three times with at least 2 biological replicates for in vitro experiments, while in vivo experiments included at least 3 biological replicates. All mouse survival cohorts are based on a minimum of at least 5 mice per genotype. n values represent number of mice and are accurately reported throughout all figure panels. |
| Data exclusions | no data was excluded |
| Replication | everything was replicated at least three times |
| Randomization | No randomisation needed |
| Blinding | stainings were performed blindly when possible |

# Reporting for specific materials, systems and methods

We require information from authors about some types of materials, experimental systems and methods used in many studies. Here, indicate whether each material, system or method listed is relevant to your study. If you are not sure if a list item applies to your research, read the appropriate section before selecting a response.

## Materials & experimental systems

| n/a | Involved in the study |
|---|---|
| ☐ | ☒ Antibodies |
| ☐ | ☒ Eukaryotic cell lines |
| ☒ | ☐ Palaeontology and archaeology |
| ☐ | ☒ Animals and other organisms |
| ☒ | ☐ Clinical data |
| ☒ | ☐ Dual use research of concern |
| ☒ | ☐ Plants |

## Methods

| n/a | Involved in the study |
|---|---|
| ☒ | ☐ ChIP-seq |
| ☐ | ☒ Flow cytometry |
| ☒ | ☐ MRI-based neuroimaging |

## Antibodies

| | |
|---|---|
| Antibodies used | Primary antibodies used: FADD (05-486, Millipore, 1:1000), RIPK1 (3493, Cell Signalling Technology, 1:1000 and 610459, BD |

| Antibodies used | Biosciences, 1:500), pRIPK1-S166 (31122, Cell Signalling Technology, 1:500), RIPK3 (95702, Cell Signalling Technology, 1:1000), pRIPK3 (91702, Cell Signalling Technology, 1:500), MLKL (MABC604, Millipore, 1:1000), pMLKL-S345 (37333, Cell Signalling Technology, 1:1000), ZBP1 (AG-20B-0010-C100, Adipogen, 1:1000), pSTAT1-Y701 (5483, Cell Signalling Technology, 1:500), TBK1 (3013, Cell Signalling Technology, 1:1000), pTBK1-S172 (72971, Cell Signalling Technology, 1:500), Caspase-8 (ALX-804-447, Enzo Life Sciences, 1:1000) GAPDH (G9545, Sigma-Aldrich, 1:10000), Total OXPHOS (ab110413, Abcam, 1:500), CD3 (ab5690, Abcam, 1:1400), Ly6G (87048, Cell Signalling Technology, 1:100), p-HH3 (06-570, Sigma-Aldrich, 1:500), γ-H2AX (9718, Cell Signalling Technology,1:50), c-Casp3 (9664 for skin, 9661 for the rest of the tissues, Cell Signalling Technology, 1:100), F4/80 (MCA497R, BioRad, 1:50), CD45 (550539, BD Biosciences, 1:200), Keratin-6A (905701, BioLegend, 1:400), Keratin-10 (ab9026, Abcam, 1:200) and Keratin-14 (905304, BioLegend, 1:400), Z-DNA/Z-RNA [Z22] (Ab00783-23.0, Absolute Antibody), TFAM (ABE483, EMD Millipore; 1:200) and DNA (CBL186, EMD Millipore; 1:100), B220-FITC (103205, BioLegend, 1:200), B220-AF700 (103232, BioLegend, 1:200), CD11b-BUV661 (612977, BD Biosciences, 1:250), CD11b-FITC (101205, BioLegend, 1:200), CD11c-BV605 (117334, BioLegend, 1:200), CD19-BUV395 (565965, BD Biosciences, 1:100), CD25-BV421 (562606, BD Biosciences, 1:100), CD3-PE-Cy7 (100219, BioLegend, 1:200), CD3-FITC (100203, BioLegend, 1:200), CD4-BUV496 (612952, BD Biosciences, 1:200), CD44-BB700 (566507, BD Biosciences, 1:200), CD45-BUV563 (612924, BD Biosciences, 1:250), CD5-FITC (553020, BD Biosciences, 1:100), CD62L-AF700 (104441, BioLegend, 1:100), CD64-PE-Dazzle (139319, BioLegend, 1:100), CD69-APC (560689, BD Biosciences, 1:100), CD71-PE (113807, BioLegend, 1:100), CD8-BV711 (563046, BD Biosciences, 1:100), GR-1-FITC (108405, BioLegend, 1:200), Ly6C-BV785 (128041, BioLegend, 1:150), Ly6G-FITC (551460, BD Biosciences, 1:100), MHCII-BUV805 (748844, BD Biosciences, 1:100), NK1.1-BUV737 (741715, BD Biosciences, 1:100), TCRβ-PE (553172, BD Biosciences, 1:100) and Ter119-BV421 (116233, BioLegend, 1:200). |
|---|---|
| Validation | All antibodies used have all been published before in our field or tested in presence of knock out cells or functionally validated withing the context of this paper. Antibodies recently published to be used for any staining have been also referenced throughout the manuscript. |

# Eukaryotic cell lines

Policy information about cell lines and Sex and Gender in Research

| Cell line source(s) | ATCC for human cell lines, for mouse all lines were derived as primary MEFs from E13.5 embryos obtained from time mating of mice bred in the study , and genotyped for desired genotype or previously described. The cell lines were subsequently immortalised and validated functionally and biochemically. Validation of all knockout cell lines is included in the figures of the manuscript. |
|---|---|
| Authentication | all cell lines have been authenticated morphologically biochemically and functionally |
| Mycoplasma contamination | all cell are routinely tested every month for mycoplasma |
| Commonly misidentified lines (See ICLAC register) | no misidentified cell lines were utilised |

# Animals and other research organisms

Policy information about studies involving animals; ARRIVE guidelines recommended for reporting animal research, and Sex and Gender in Research

| Laboratory animals | C57BL/6N |
|---|---|
| Wild animals | This study did not involve wild animals. |
| Reporting on sex | both males and females were used for the study. There was no difference in any of our result based on sex differences |
| Field-collected samples | Ad libitum access to water and standard chow was provided, while room temperature and humidity were automatically controlled. Continuous daily monitoring was conducted, and when the animals met predetermined endpoint criteria, they were humanely euthanized. All experimental procedures adhered strictly to German animal protection laws, received approval from local ethics committees, and were sanctioned by local government authorities (Landesamt für Natur, Umwelt und Verbraucherschutz Nordrhein-Westfalen). |
| Ethics oversight | Landesamt für Natur, Umwelt und Verbraucherschutz Nordrhein-Westfalen |

Note that full information on the approval of the study protocol must also be provided in the manuscript.

# Plants

| | |
|---|---|
| Seed stocks | n/a |
| Novel plant genotypes | n/a |
| Authentication | n/a |

# Flow Cytometry

## Plots

Confirm that:

☒ The axis labels state the marker and fluorochrome used (e.g. CD4-FITC).

☒ The axis scales are clearly visible. Include numbers along axes only for bottom left plot of group (a 'group' is an analysis of identical markers).

☒ All plots are contour plots with outliers or pseudocolor plots.

☒ A numerical value for number of cells or percentage (with statistics) is provided.

## Methodology

| | |
|---|---|
| Sample preparation | Spleen, lung and thymus were dissected from 6-8-week-old mice. Lungs were chopped and digested in 1.5 mL Liberase/DNAse digestion buffer for 30 min at 37°C. The spleens, thymi and digested lungs were passed through a 70 μM cell strainer to obtain single-cell suspensions, then centrifuged at 1,500 rpm for 5 min at 4°C. For lung and spleen, red blood cell lysis was performed using 1x RBC lysis buffer for 4 min on ice. The reaction was stopped by addition of PBS followed by centrifugation as above. Spleens for erythrocyte analysis and thymi were not subjected to red blood cell lysis. All samples were blocked with anti-mouse CD16/32 antibody for 30 min at 4°C, then washed with FACS buffer. Cell viability was assessed by incubation with a fixable viability dye for 30 min at 4°C. After washing, cells were incubated with the respective surface antibodies in Brilliant Stain Buffer for 30 min at 4°C. |
| Instrument | BD FACSymphony A3 Cell Analyser |
| Software | Data were collected using the BD FACS Diva software. Raw FCS files were analysed using FlowJo v10.10 and statistical analysis and graphing were performed using GraphPad Prism 10. |
| Cell population abundance | Flow cytometry was used to assess the abundance of distinct cell populations in whole spleen, lung and thymus. Immune cells in spleen and lung were identified by CD45-BUV563 staining and further classified using the markers listed in Supplementary Table 4. Thymocytes were defined by B220-APC-R700-negativity, followed by further refinement using lineage markers. In the spleen, erythroid populations were gated as lineage-negative cells and defined by CD71-PE and TER119-BV421 expression. Cell population abundances are reported as percentages of viable cells or of the parent population, as indicated in the figure legends (Fig. 6 and Extended Data Fig. 9). |
| Gating strategy | For spleen, lung and thymus analysis, the starting population was defined by SSC-A/FSC-A gating and including all acquired cells. Single cells were identified via FSC-H/FSC-A and further refined by FSC-W/FSC-A gating. Viable cells were gated as APC-H7 (fixable viability dye)-negative events against SSC-A. In spleen and lung, immune cells were gated as CD45-BUV563-positive versus SSC-A and further subdivided using lineage markers listed in Supplementary Table 4. Thymocytes were gated as B220-APC-R700-negative cells versus SSC-A and further characterised by expression of CD4-BUV496 versus CD8-BV711 and CD25-BV421 versus CD44-BBB700. Erythrocyte populations in the spleen were gated as lineage-BB515-negative cells versus SSC-A, further defined by TER119-BV421 versus CD71-PE expression and subsequently clustered based on FSC-A versus CD71-PE. The full gating strategy for every organ is further specified in Supplementary Figure 2. |

☒ Tick this box to confirm that a figure exemplifying the gating strategy is provided in the Supplementary Information.

