## [Peer Review File · Nature]

STING induces ZBP1-mediated necroptosis independently of TNFR1/FADD

Corresponding Author: Dr Gianmaria Liccardi

Version 1:

Reviewer comments:

Referee #1

(Remarks to the Author)

This study opens with the authors confirming that Casp8 deletion in mouse keratinocytes elicits inflammation and, as reported in MEFs and other mouse tissues, Casp8 loss is associated with an upregulation of IFN-regulated genes and other markers of IFN signaling (e.g. p-STAT1). Data set the stage but are low in terms of novelty. Fig. 2 uses immortalized MEFs and a STING antagonist to link activation of STING to increased p-STAT1 and ZBP1 in Casp8 KO cells. This result is unexpected and therefore genetic validation using Casp8 Sting DKO cells is needed as well as at least one other cell type (see point 2 below). It is also unclear why STING activation induces significant death in Casp8 KO MEFs (EDF2e) without altering ZBP1 or MLKL levels significantly (Fig. 2e); even without treatment there is aberrantly high expression of ZBP1 and p-STAT1. In other words, what else is STING activation doing to trigger the death of the Casp8 KO cells? Does STING antagonist suppress basal p-STAT1 and ZBP1 in the Casp8 KO cells in Fig. 2e?

Fig. 3 shows that Sting or Tnfr1 loss delays but does not prevent skin inflammation in Casp8 E-KO mice. Fig. 4 shows that the combined loss of Sting and Tnfr1 is much more effective at delaying disease in Casp8 E-KO mice, but is not as good as the combined loss of Zbp1 and Tnfr1. Given that STING loss is restricted to keratinocytes and ZBP1 loss is full body, the interpretation of this difference is uncertain.

Why STING is active in Casp8 KO cells/mice is unclear. While it might be beyond the scope to address this in the Casp8 E-KO model, is cGAS driving STING activation and p-STAT1 in Casp8 Mkl1 DKO cells? A quick Crispr expt in MEFs or BMDMs would be informative.

Fig. 5 attempts to link STING activation to cell death in humans, but doesn't really hit the mark. Patients bearing activating mutations in STING expressed ZBP1 and MLKL mRNAs at higher levels, but this just indicates that they are necroptosis-competent. It does not prove necroptosis induction. Unless they can show p-MLKL in patient samples, I don't think this dataset is worthy of inclusion, certainly not as a main figure.

Overall, I think the results presented are tantalizing but preliminary. Key controls and evidence are missing as indicated above and in the specific points below.

Other issues that need addressing:

1. The authors assume a critical role for necroptosis in the Casp8 E-KO mice without any actual proof. They cite ref. 1 but this study only described inflammation in Casp8 E-KO mice, and did not show that it was due to RIPK3- and MLKL-driven necroptosis. The authors assume necroptosis underlies their phenotype because of what has been described in other models. For example, Ripk3 loss rescues the inflammation in Fadd E-KO mice (PMID: 22000287), Mkl1 loss rescues inflammation in Casp8 IEC KO mice (PMID: 32362323), and Mkl1 loss rescues inflammation in Casp8 C362S E-KI mice (PMID: 31748744). However, given that there is no Mkl1 KO rescue data published for the Casp8 E-KO mice, the onus is on the authors to prove that Mkl1 KO Casp8 E-KO mice are similarly rescued. They should also establish whether the up-regulation of p-STAT1 and IFN-regulated genes in Casp8 E-KO mice (Fig. 1d, 3g-i) is independent of necroptosis.

2. A reference establishing the specificity of the STING antagonist C-178 should be provided on line 124. Ideally, the authors

- validate the results in Fig. 2 by CRISPR knockout of Sting in the Casp8 KO MEFs. They should also not rely entirely on immortalized MEFs for drawing conclusions because these cells acquire other mutations/epigenetic changes in a random fashion. Validation of these results in another primary cell type, such as bone marrow-derived macrophages, is needed.
3. The authors show biochemically that STING activation alone can trigger assembly of a ZBP1-RIPK1-RIPK3 complex in WT cells (Fig. 2d). Is this complex increased by Casp8 deficiency or inactivation (this data is missing from Fig. 2e)? Presumably the ZBP1-RIPK1-RIPK3 complex is also detected in untreated Casp8 Mkl DKO cells since these cells contain basal levels of pRIPK3. Does the STING antagonist suppress complex formation as well as basal levels of pRIPK3 in Casp8 Mkl DKO cells?
 4. The prominent band in the FADD WBs of the ZBP1 IPs in Fig. 2e (light chain of the IP antibody?) make it hard to exclude FADD from these complexes. The authors should either clean up the experiment or remove what is an uninformative FADD WB. To formally prove that complex formation is FADD-independent (as claimed on line 179), the authors should show that the ZBP1-RIPK1-RIPK3 complex is formed in Fadd KO cells.
 5. Given that STING agonist alone doesn't kill WT cells (EDF2a), they should assess the functional contribution of ZBP1 to STING-agonist killing by CRISPR KO of Zbp1 in the Casp8 KO cells in EDF2e. Looking at additive effects of TBZ + STING agonist in EDF2a is messy.
 6. Line 213 – what is “necroptosis-driven secondary apoptosis?”
 7. Line 410 – isn't there a 3rd possibility – ruxolitinib exposures were insufficient to completely block IFNAR signaling? It is hard to conclude anything definitive.

Minor points:

1. Fig. 1c. Define the “endpoint” in the legend. Is it P7?
2. EDF2 legend does not indicate when viability was assessed after TBZ treatment.
3. The authors should provide a reference for what genes are regulated by IFNs on line 418.

Referee #2

(Remarks to the Author)

In this manuscript, Liccardi and colleagues present in vivo evidence that the lethal dermatitis caused by epithelial cell-specific deletion of Casp8 is driven, in part, by STING-mediated upregulation of a tonic Type I IFN signal, and consequent induction of ZBP1 and MLKL, resulting in ZBP1-driven necroptosis. This STING/ZBP1 pathway of necroptosis appears to act in parallel with TNF-driven necroptosis (also unleashed by Casp8 loss) in promoting skin pathology and systemic inflammation. The authors then extend these findings to suggest that the human interferonopathy SAVI induced by gain-of-function mutations in STING might be driven by ZBP1-induced necroptosis, offering the possibility of such necroptosis as a therapeutic intervention in this currently intractable autoinflammatory condition.

The data presented are largely consistent with the authors' conclusions, but as currently constituted, the study lacks novelty and offers only incremental insight into ZBP1-dependent inflammatory biology over what is already published. Mainly, the primary insight of this paper – that it is STING which induces the IFN requisite for ZBP1 and MLKL expression - was shown by Doug Green and colleagues in 2022 (PMID 36191211). That PNAS paper has already quite convincingly demonstrated that a cGAS/STING signal is what induces tonic IFN in scenarios of Casp8 (or FADD) loss, and that this signal is required for ZBP1 induction and necroptosis when Casp8 is lost. Nonetheless, two major biological features of ZBP1 activation in the current paradigm remain unknown, which, if experimentally addressed, will impactfully advance the field. Additionally, the therapeutic potential of the results presented remain unproven in vivo. These points are detailed below:

Major points:

1. Mechanism by which STING is activated by Casp8 loss. While it has been shown that cGAS/STING signaling promotes IFN-mediated upregulation of ZBP1 and consequent necroptosis when Casp8 is ablated in cells from KO mice (the Green PNAS paper) or in murine skin (this study), the mechanism by which Casp8 loss induces STING-dependent IFN is unknown. That is, how does Casp8 repress STING activation? Presumably, this occurs by preventing sensing of endogenous dsDNA by cGAS, as the Green paper would suggest. What, then, is the mechanism by which Casp8 keeps endogenous dsDNA from being sensed by cGAS, and what are the ligands for cGAS that are unleashed by Casp8 loss? (Mitochondrial DNA may be one possibility: PMID37352855.) Or if the mechanism is cGAS independent, then how does Casp8 repress STING?
2. Mechanism by which ZBP1 is activated. Simple induction of ZBP1 by IFNs is – in itself – is probably not sufficient to activate this sensor; ZBP1 likely binds Z-NA species, which drive its activation. In fact, the Green paper mentioned earlier showed that Casp8 loss results in a spontaneous ZNA signal and that a mutant ZBP1 lacking Z-NA sensing capability fails to activate necroptosis when Casp8 is ablated. What are the Z-NA species which activate ZBP1, and how are these ZNAs quenched by Casp8? Or if the mechanism in vivo is independent of ZNA sensing, then how is ZBP1 activated to induce necroptosis?
3. Therapeutic implications of these findings for treatment of SAVI patients. The authors have suggested that ZBP1-driven necroptosis may underlie some of the pathology seen in SAVI patients but stop short of exploiting these findings for therapeutic benefit of these patients. Can necroptosis inhibitors (perhaps paired with inflammasome, TNF blockers, and/or JAK inhibitors) prevent skin pathology and lethality in the Casp8E-KO mouse?

The data are cogently presented, the paper is well-written, and the citations are appropriate. Statistical considerations appear appropriate.

(Remarks to the Author)

Kelepouras et al. submitted a manuscript presenting evidence that the ZBP1-RIPK3 pathway of “necroptosis execution in mice requires activation of an interferon (IFN)-mediated transcriptional program orchestrated by the stimulator of interferon gene (STING).” This work brings together a body of work on STING regulation of interferon activation and cell death induction with growing evidence that ZBP1-RIPK3 signal transduction is a critical determinant of inflammatory outcomes in mice. The work shows that striking dermatitis in mice lacking caspase-8 (CASP8) in epidermal keratinocytes (Casp8E-KO), already known to be dependent on TNFR1-FADD-RIPK1 signaling, also depends on STING-induced interferon-mediated upregulation of ZBP1 and MLKL and the ZBP1-RIPK3-MLKL pathway of necroptosis. STING itself is induced in these mutant mice and could be further enhanced with a STING agonist in cultured CASP8-deficient fibroblasts, reinforcing a ZBP1-RIPK3-MLKL pathway of necroptosis functioning independently of the well-studied FADD-RIPK1-RIPK3-MLKL pathway.

Authors then generated mice with conditional deletion of STING in epidermal keratinocytes (StingE-KO) and crossed mice to produce Casp8E-KO/StingE-KO double mutant mice which exhibited a modest delay in ZBP1-RIPK3-MLKL necroptosis with a phenotype similar to the previously reported Casp8E-KO/Tnfr1-/- phenotype. This pathway proceeds more aggressively in CASP8-deficient epithelial cells in the presence of STING-induced type I interferon-mediated elevated ISGs, documented by transcriptional profiling, which showed the expected elevated ZBP1 and MLKL. The authors pursue additional profiling and conclude, “STING in initiating the IFN response and subsequent upregulation of genes involved in immune cell recruitment and necessary for the propagation of necroptosis-induced inflammation”, which is entirely consistent with current understanding of STING activation of the type I interferon pathway (Zheng et al., 2023) as well as ZBP1 as an interferon-induced gene product that senses endogenous and viral nucleic acids to induce complex death pathways that have been called PANoptosis (Man and Kanneganti, 2024; Pandian and Kanneganti, 2024). ZBP1-RIPK3 signaling is recognized as critical and pharmaceutical RIPK3 kinase inhibition prevents inflammatory disease dependent on ZBP1-RIPK3 cell death induction (Gautem et al., 2024).

Triple mutant Casp8E-KO; Tnfr1-/-; Zbp1-/- completely rescued viability without dermatitis; whereas, Casp8E-KO; Tnfr1-/-; Sting E-KO mice showed slow development of dermatitis associated with elevated levels of ZBP1, predominantly located in the dermis, accompanied by increased levels of cl-CASP3 and obvious leukocytosis, anaemia, thrombocytosis and defects in erythropoiesis. Due to their ability to link STING and ZBP1-RIPK3 signaling, authors refer to STING-activation-induced necroptosis (STAIN) and “a functional correlation between a STING-mediated IFN response and the transcriptional expression of ZBP1, MLKL and the cell-autonomous cytokines known to be released following necroptosis”; however, the observations and possible scenarios proposed beg the question of why ZBP1 deficiency was so effective in reversing dermatitis where STING deficiency in epithelial cells was not. The authors must further explore whether STING and ZBP1 must be in the same epithelial cell compartment for STAIN to play out. Cell autonomous signaling is possible, but immune cell-epithelial cell crosstalk via cytokines seems just as likely here. One likely alternate explanation might envision different compartments for STING-dependent type I interferon production, such as infiltrating immune cells, and execution of inflammatory damage, the execution of cell death in epithelial cells. This might also result in either amplifying inflammatory loop (interferon->ZBP1->death->interferon-> ZBP1). This setting is reminiscent of endotoxic shock in mice where type I interferon and TNF produced by initiator immune cells mediate cell death pathways in executed gut epithelial cells (Mandal et al., 2018), experimentally addressed by showing immune cells with mutations in the executioner components (such as Casp8-/-; RIPK3-/- mice) were able to induce endotoxic shock in animals that are compromised in the ability to induce type I interferon (such as Irf3-/- mice).

In any case, the data here is promising but inconsistent, and it does not distinguish cell autonomous or cell-cell crosstalk mediating STAIN, so must go further to resolve the mechanism. Two lines of investigation are warranted: (1) compare Casp8E-KO; Tnfr1-/-; Sting E-KO to Casp8E-KO; Tnfr1-/-; Sting-/- mice, where an essential role of systemic role of STING would be expected to phenocopy ZBP1 deficiency, and, (2) distinguish the immune cell cytokine contribution from the epithelial-cell signaling requirements for cell death execution and dermatitis. Further experiments must determine whether STAIN is sustained when STING signaling is separated from epithelial cells where ZBP1-dependent cell death occurs. The dependency on STING and ZBP1 function in the same executioner compartment would reinforce the use of STAIN, but if function can be separated into different compartments this term has less utility.

A similar ISG-enhanced RNAseq pattern is shown by transcriptionally-derived gene set enrichment analysis (GSEA) in human patients with auto-inflammatory disorder STING-Associated Vasculopathy with onset in Infancy (SAVI). This is likely fortuitous because the mice lack CASP8 in epithelial cells; whereas the cells involved in disease in patients have fully functional CASP8 (as well as CASP10). This observation reveals only that an interferon signature is a very crude comparator. All STING-associated and STING gain-of-function outcomes have long been associated with elevated ISGs (Barber, 2014; Ahn and Barber, 2014) independent of CASP8 compromise. Importantly here, STING has also been observed to drive death pathways in published work, including apoptosis (Gulen et al., 2017) and necroptosis (Chen et al., 2018) and the relationship to ISGs has been of great interest. In these and many other papers have pursued STING-induced death shown to be dependent on RIPK3, MLKL, as well as BH3-only BCL2 family members but independent of RIPK1 (Zheng et al., 2023) although no study has yet hinted whether CASP8 (driving extrinsic apoptosis) or CASP8 compromise (driving necroptosis) contributes. The authors need to better integrate with this understanding of STING signaling.

Human STING gain-of-function disease outcomes certainly involve further inflammatory amplification that remain to be unveiled (Chavin et al., 2023). The demonstration that STING-mediated inflammation drives events autonomously in the

epithelial cell or through immune cell-epithelial cell cytokine-mediated crosstalk, intrinsic STING-induction of the ZBP1-RIPK3-MLKL inflammatory pathway is a significant advance once the timing and dissection role of immune and epithelial compartments is clear.

Finally, authors should pursue some evidence whether apoptotic and necroptotic (or other broader) outcomes are contributing, particularly given the observations complex pyroptotic, apoptotic and necroptotic signaling proposed for PANoptosis and the PANoptosome (Man and Kanneganti, 2024; Pandian and Kanneganti, 2024).

Other points:

1. The ZBP1-RIPK3 pathway was first revealed soon after the RIPK1-RIPK3 pathway when cytomegalovirus-induced necroptosis was shown to be independent of RIPK1 (Upton et al., 2010) and completely dependent on ZBP1 (Upton et al., 2011). This era also brought to light RIPK3 knockout rescue of the embryonic lethality exhibited by CASP8-deficient mice, a publication (Kaiser et al., 2011; Oberst et al., 2011) revealed the dependency of virus-induced ZBP1-RIPK3 necroptosis on CASP8 inhibition. Despite being a necroptotic death pathway independent of RIPK1, a requirement for CASP8 is already known to be shared across the three necroptotic pathways, ZBP1-RIPK3 and TRIF-RIPK3 (He et al., 2011; Kaiser et al., 2013) as well as the more well studied RIPK1-RIPK3 pathway. An unknown remains why RIPK1-independent necroptosis would still be dependent on CASP8 compromise when all available evidence points to specific cleavage-mediated control of RIPK1 (Newton et al., 2019) and not over RIPK3 (Newton et al., 2024), although all three pathways have not been fully evaluated in this regard (and should be).

2. This work follows considerable attention to STING and ZBP1 in host defense; however, with particular focus on the induction of intrinsic apoptosis over either extrinsic apoptosis or necroptosis. Thus, STING is best known to drive inflammatory cytokine induction and intrinsic apoptosis, a pathway that is not really considered or eliminated here by authors. Literature that deals with STING and cell death pathways that limit levels of both viral replication and inflammation might be useful in this regard (reviewed in Paludan et al., 2019, 2024; Zhan et al., 2024). Extensive studies have implicated apoptosis of microglia cells in host control over herpesvirus encephalitis (Reinhart et al., 2021) acting via microbe- and inflammation-restricting mechanisms (MIMs; Paludan et al., 2024). It is well known that RIPK3 function controls necroptosis and inflammasome activation in addition to CASP8-dependent apoptosis in a conditional fashion (Moriwaki and Chan, 2017;) and has been observed in influenza (Man and Kanneganti, 2024) as well as herpesvirus studies in mice (Jeffries et al., 2022; Guo et al., 2022). Along these lines, RIPK3 inactivating mutations have been implicated in susceptibility to herpesvirus encephalitis in a human patient (Liu et al., 2023) in a pattern consistent with altered cell death susceptibility but this patient did not exhibit susceptibility to any other notable diseases.

Minor correction:

Typo (citations): "Unlike deletion of RIPK3 or MLKL, loss of tumor necrosis factor receptor 1 (TNFR1) does not prevent, but only delays the embryonic lethality observed in Casp8 ^{-/-} mice and the lethal dermatitis of Casp8 E-KO mice^{1,7}." Superscript 1,7 should read 1-7.

Version 3:

Reviewer comments:

Referee #1

(Remarks to the Author)

The revision was a difficult read. The authors added a lot of new data, and the descriptions would have benefited from editing to be more concise. In addition, they often make statements without referring to the relevant figure panel(s).

The key pieces of new data are:

1) Casp8 E-KO Sting KO and Casp8 E-KO cGAS KO mice show better survival than Casp8 E-KO Zbp1 KO mice (Fig. 4a and EDF9a), arguing that Sting does more than just trigger ZBP1-driven necroptosis.

This fits with their model that STING activation also promotes TNFR1-induced necroptosis in Casp8 KO keratinocytes. Although they show that TNF protein is less evident in Casp8 E-KO Sting E-KO skin compared with Casp8 E-KO skin (EDF4e), it would be helpful if they included Tnf transcript levels in the Fig. 2g-i heatmaps.

2) Casp8 E-KO Tnfr1 Sting DKO mice survive to 14 wks without overt issues, just like Casp8 E-KO Tnfr1 Zbp1 DKO mice. The authors speculate on lines 445-452 that STING activity outside of keratinocytes explains why Casp8 E-KO Tnfr1 KO Sting E-KO mice are not rescued to the same extent as Casp8 E-KO Tnfr1 Sting DKO mice. The issue I have with this model is that if the initiating events in keratinocytes are blocked by the combined loss of Tnfr1 and Sting, then what is triggering Sting activation in non-keratinocytes? It seems like the simpler/more plausible explanation for the difference is incomplete deletion of the conditional Sting allele. Given the limitations of the conditional model, they should temper their conclusions or provide further insights.

3) Cytosolic DNA is detected in Casp8 Mkl1 DKO MEFs. They posit that the cytosolic DNA stems from genomic DNA damage and engages cGAS in Casp8 KO cells. I think this is highly speculative since Casp8 IEC KO or E-KO mice rescued by Ripk3 KO or Mkl1 KO don't show a propensity for tumor development as you would expect from ongoing DNA damage.

4) Tissues from Sting N153S KI mice that model human SAVI exhibit markers of necroptosis (pMLKL) and apoptosis (CC3).

As a sanity check for the specificity of their pMLKL IHC, does the pMLKL signal in the Sting KI mice (Fig. 6f) go away in Sting KI Ripk3 KO mice? Although some disease manifestations in the Sting KI mice are ameliorated by RIPK3 deficiency, this genetic data doesn't prove a contribution of necroptosis to disease since RIPK3 also has necroptosis-independent functions.

In summary, I like the new genetic data in the Casp8 E-KO model placing cGAS and STING upstream of Zbp1- and Tnfr1-driven necroptosis. However, in my opinion, the data for necroptosis being an important driver of disease in SAVI remains tenuous. Caspase-8 is still present in this context and so would be expected to act as a brake on necroptosis signaling.

Other points:

1. Lines 133-135 "treatment with the mouse STING agonist (DMXAA) or interferon-g (IFN- g) enhanced the basal level of ZBP1, which was completely abolished or significantly reduced, respectively, by co-treatment with C-178".

I agree with C-178 abolishing DMXAA-induced ZBP1.

I do not see that it "significantly reduced" IFNg-induced ZBP1 in Fig. 1e or 1f. Recommend re-wording.

2. Lines 271-272 "the transcriptional upregulation of IFN-g and IFN-b was significantly attenuated by keratinocyte-specific STING deficiency".

They don't cite a figure panel but presumably are referring to EDF4g. The n for each genotype is not indicated in the figure legend, but the difference in Ifnb transcripts doesn't look that impressive. A multiple comparisons statistical test may be warranted in this setting, rather than just individual student t-tests.

3. Lines 375-376 "aberrant STING activation as a driver of a transcriptional programme that includes the IFN-dependent upregulation of ZBP1 and MLKL".

They haven't formally proven IFN-dependent upregulation of these genes in this setting. Rewording suggested.

4. EDF9d – the authors should show that the cytosolic "DNA" detected by immunolabelling in Casp8 KO cells disappears if cells are treated with DNase prior to labelling.

5. EDF9i – if DNA damage is upstream of cGAS-STING activation, then do these markers still appear in Casp8 Mkl DKO or Casp8 E-KO STING TNFR1 DKO skin? WB for these markers should also be provided to indicate that proteins of the appropriate MW are being detected.

Referee #2

(Remarks to the Author)

In response to the original critiques, the authors have carried out an extensive set of experiments, including several new mouse crosses, to support the idea that chromosomal instability following loss of Casp8 triggers cGAS-STING-dependent induction of IFNs and ZBP1, and Z-NA induction, also induced by loss of Casp8, then activates ZBP1, driving necroptosis. They then extend the clinical relevance of these findings by demonstrating that some of the pathological features seen in a GOF mouse model of aberrant STING activation are rescued by loss of RIPK3. The finding that caspase8 functions beyond apoptosis in controlling the integrity of nuclear DNA has been suggested in earlier studies, including by the authors themselves. The current paper now demonstrates the importance of this 'non-canonical', and frankly unexpected, function of caspase 8 in repressing the activation of key nucleic acid sensors and the inflammation these provoke. Although the connection of Casp8 loss to SAVI (or whether aberrant casp8 function underlies other interferonopathies) remains unproven, this work provides new mechanistic insight and in vivo relevance to an interesting and unanticipated new feature of caspase 8 biology, and I am in support of publication.

I have a few additional comments:

1. The title does not properly capture what, to me at least, is the primary feature of this paper: that Casp8 loss is what instigates the entire inflammatory cascade. I recommend: "Caspase 8 prevents STING-dependent activation of ZBP1", or similar. Invoking TNFR1 independence in the title is not necessary, as ZBP1 activation in most scenarios is not typically considered TNF dependent in the first place.

2. The Abstract should specifically mention that caspase 8 loss also triggers Z-NA accrual, which drives ZBP1 activation. This extra text can come at the expense of some of the SAVI writeup, which currently makes up about half the abstract.

3. The key mechanistic figures linking caspase 8 loss to cGAS activation, and to ZBP1 activation are, respectively, the panels showing genomic instability in Casp8 KO skin (Fig S9i) and the convincing new Z-NA staining (Fig S10). I recommend including these panels in the main Figures, many of which are currently given over to histological images better suited for Supplemental.

4. Similarly, key data panels from the RIPK3 rescue of the STING GOF mouse model (i.e., flow cytometric data from S14f,h,i,k) could be included in Fig 6. Is dermatitis in the STING GOF mice rescued by RIPK3 loss? I could not find any allusion to this in the revised paper or rebuttal. Whatever the outcome of these analyses, the authors should make a mention

of what is seen in the skin of the STING GOF x RIPK3 KO mice.

Referee #3

(Remarks to the Author)

Kelepouras et al. submitted a fully revised manuscript establishing that the ZBP1-RIPK3-MLKL pathway depends on STING-driven activation of ZBP1 as a central inflammatory mechanism in both Casp8-deficient inflammation and STING-Associated Vasculopathy with onset in Infancy in humans. This work brings together a body of work on STING regulation of interferon activation and cell death induction with growing evidence that ZBP1-RIPK3 signal transduction is a critical determinant of inflammatory outcomes in mice and humans.

The authors have provided extensive responses to all reviewers concerns accompanied by significant additional data and evidence. All coming forward as a stellar piece of scholarship.

Version 5:

Reviewer comments:

Referee #1

(Remarks to the Author)

The authors have addressed my comments adequately.

Dear Dr. Weiss,

We are writing regarding our manuscript “STING induces ZBP1-mediated necroptosis independently of TNFR1”, which was recently rejected following revision. We greatly appreciate the reviewers' thoughtful feedback and have carefully considered their comments.

We would like to inform you that the requested experiments are either completed or well underway, including complex mouse genetic crosses and specific assays in anticipation of reviewer suggestions. These data directly address the reviewers' concerns and will significantly strengthen the manuscript. We anticipate being able to finalize the experiments and submit the updated manuscript within 4 months.

Given this progress, we would like to inquire if you would consider a resubmission with these additional data. We have included a point-by-point summary of the reviewers' comments and our planned responses below for your consideration.

Thank you for your time and for considering this request. We greatly value the opportunity to work with Nature and look forward to hearing your thoughts.

Kind regards,

Gianmaria Liccardi

General comments:

While we agree that the ability of STING to drive an interferon response and induce ZBP1 activation was reported, this was never demonstrated a) to be the sole and main source of ZBP1 upregulation, or b) to effectively contribute to necroptosis in a cell-intrinsic manner, neither in cells nor genetically. In our study, for the first time we provide genetic evidence that necroptosis is potently inhibited by genetic deletion of STING in epidermal keratinocytes, which are the cells in which loss of Caspase8 drives necroptosis activation leading to lethality. Our work provides evidence of the dual contribution of STING and TNFR1 in driving necroptosis activation independently from one another, an observation, that, to our knowledge, has never been demonstrated before. Furthermore, in SAVI patients, we provide evidence of STING activation independently of the treatment with JAK1/2 inhibitors still present a necroptotic cell death signature at the level of RNA sequencing as well as inflammatory cytokines. This is suggesting that STING is driving both interferon response activation and necroptosis activation independently of the interferon response downstream of IFNAR, contributing to the inflammation of the SAVI patients in which treatment with Ruxolitinib does not provide any benefit. While we believe this represents of conceptual advance in different fields, we understand that the reviewers would like further evidence to solidify some of our findings with additional work which we are very happy and ready to provide.

Below I am providing the details of our completed and ongoing work which I believe fully addresses all questions from all reviewers. These include some complex genetic crosses which we began more than a year ago in anticipation of these potential questions.

Point-by-point summary of the reviewers' comments, organized by thematic relevance. Reviewer request is in *Italics*:

1. Mechanistic Clarification of STING Activation upon loss of Casp8

Referee #1: Fig. 2 uses immortalized MEFs and a STING antagonist to link activation of STING to increased p-STAT1 and ZBP1 in Casp8 KO cells. This result is unexpected and therefore genetic validation using Casp8 Sting DKO cells is needed as well as at least one other cell type (see point 2 below). It is also unclear why STING activation induces significant death in Casp8 KO MEFs (EDF2e) without altering ZBP1 or MLKL levels significantly (Fig. 2e); even without treatment there is aberrantly high expression of ZBP1 and p-STAT1. In other words, what else is STING activation doing to trigger the death of the Casp8 KO cells? Does STING antagonist suppress basal p-STAT1 and ZBP1 in the Casp8 KO cells in Fig. 2e?

2. A reference establishing the specificity of the STING antagonist C-178 should be provided on line 124. Ideally, the authors validate the results in Fig. 2 by CRISPR knockout of Sting in the Casp8 KO MEFs. They should also not rely entirely on immortalized MEFs for drawing conclusions because these cells acquire other mutations/epigenetic changes in a random fashion. Validation of these results in another primary cell type, such as bone marrow-derived macrophages, is needed.

Answer:

The point regarding STING activation by DMXAA resulting in the upregulation of pSTAT1 and ZBP1 and activation of necroptosis (pMLKL S345) in Casp8^{KO} MEFs is illustrated in Figures 2a, 2b, and 2e. Additionally, the effect of the STING antagonist in Casp8^{KO} MEFs is shown in Figures 2a and 2b.

We have also generated the *Casp8^{E-KO} Sting^{KO}* mice where we have performed RNA sequencing and several stainings to show the complete absence of the listed markers.

Since we have not examined the full body deletion of Casp8, we cannot anticipate any outcome at this stage with regards to isolation of primary cells from a *Casp8/STING* double deficient mice as *Casp8^{-/-}* dies at embryonic stage 10.5 which impedes the harvesting of any primary cell line. The requested experiment in BMDMs would require infection of the *Casp8^{fl/fl}* cells (of any origin) with *cre* to generate the knockout and consequent immortalisation defeating the purpose of the alternative primary cell line request. We can provide similar results obtained on CRISPR generated Casp8-deficient MDFs on where we have deleted STING, however these cannot be primary for technical reasons. We have also performed such experiments in human cells (HT29 and HaCat) which, while being also immortalized, are a cellular system from a completely different species. To satisfy the reviewer comments, we can provide additional WT primary cells of different origins treated with DMXAA to show enhanced necroptosis activation following TBZ in additional cell lines.

In response to the referee #1's suggestion for Figure 2e, we propose providing a shorter exposure for the blots in this figure to more effectively highlight the differences relative to basal levels, similar to the blots shown in Figures 2a and 2b.

Additionally, to investigate the potential role of STING in regulating cell death beyond the upregulation and activation of ZBP1, as suggested by referee #1, we have generated *Casp8^{E-KO}; Sting^{KO}*, *Casp8^{E-KO}; Zbp1^{KO}* and *Casp8^{E-KO}; Sting^{KO}; Zbp1^{KO}* mice. The survival and sample analysis of *Casp8^{E-KO}; Sting^{KO}* and *Casp8^{E-KO}; Zbp1^{KO}* is already completed. A survival cohort of 10 mice for *Casp8^{E-KO}; Sting^{KO}; Zbp1^{KO}* mice is expected to be evaluated by mid-April 2025. The data are currently pointing at a role for STING which is initially/temporally independent of ZBP1 since the *Casp8^{E-KO}; Zbp1^{KO}* show no survival difference with the *Casp8^{E-KO}* mice. We have already performed different omics analysis to highlight the transcriptional differences between these genotypes and investigate the differences with the *Casp8^{E-KO}; Tnfr1^{-/-}* as well as IHC validations to show how these are reflected at the protein level in the skin.

2. Mechanisms of STING Activation in Casp8 Deficiency: Exploring the potential Role of cGAS and Ligand Identification

Referee #1: Why STING is active in Casp8 KO cells/mice is unclear. While it might be beyond the scope to address this in the Casp8 E-KO model, is cGAS driving STING activation and p-STAT1 in Casp8 Mki1 DKO cells? A quick Crispr expt in MEFs or BMDMs would be informative.

Referee #2: 1. Mechanism by which STING is activated by Casp8 loss. While it has been shown that cGAS/STING signaling promotes IFN-mediated upregulation of ZBP1 and consequent necroptosis when Casp8 is ablated in cells from KO mice (the Green PNAS paper) or in murine skin (this study), the mechanism by which Casp8 loss induces STING-dependent IFN is unknown. That is, how does Casp8 repress STING activation? Presumably, this occurs by preventing sensing of endogenous dsDNA by cGAS, as the Green paper would suggest. What, then, is the mechanism by which Casp8 keeps endogenous dsDNA from being sensed by cGAS, and what are the ligands for cGAS that are unleashed by Casp8 loss? (Mitochondrial DNA may be one possibility: PMID37352855.) Or if the mechanism is cGAS independent, then how does Casp8 repress STING?

Answer:

To address this question, we have treated *Casp8*^{KO} MEFs with the cGAS inhibitor RU.521 and assessed the expression of pSTAT1, ZBP1 and necroptotic markers. Concurrently, *Casp8*^{E-KO}; *cGAS*^{KO} mice have been generated, with a survival analysis of at least 10 mice expected to be finalized by end of April.

In response to referee #2's suggestion to further investigate the potential source of dsDNA that could activate cGAS/STING, we have performed immunofluorescence stainings for dsDNA, TFAM, and mitochondria in *Casp8*^{KO} vs. WT cells. Additionally, both cell types were treated in the presence and absence of ddC, and we assessed the levels of pSTAT1, ZBP1 and necroptotic markers following confirmation of mitochondrial gene depletion via RT-qPCR.

To additionally and genetically address the potential involvement of mitochondrial DNA, we have also generated and completed the survival analysis of *Casp8*^{E-KO}; *MAVS*^{KO} mice.

Our current results suggest that the source of DNA is not mitochondrial and that STING might be also partially activated independently of a DNA-sensor. Importantly, our data would point towards a direct ability of Casp8 to suppress STING accumulation/overexpression-induced activation, which was shown to be sufficient in driving interferon response in the original paper from Ishikawa & Barber (Nature 2008).

3. Investigating the Role of STING and ZBP1 in Epithelial and Immune Cell Contributions to Dermatitis

Referee #1: Fig. 3 shows that *Sting* or *Tnfr1* loss delays but does not prevent skin inflammation in *Casp8* E-KO mice. Fig. 4 shows that the combined loss of *Sting* and *Tnfr1* is much more effective at delaying disease in *Casp8* E-KO mice, but is not as good as the combined loss of *Zbp1* and *Tnfr1*. Given that STING loss is restricted to keratinocytes and ZBP1 loss is full body, the interpretation of this difference is uncertain.

Referee #3: Two lines of investigation are warranted: (1) compare *Casp8*E-KO; *Tnfr1*^{-/-}; *Sting* E-KO to *Casp8*E-KO; *Tnfr1*^{-/-}; *Sting*^{-/-} mice, where an essential role of systemic role of STING would be expected to phenocopy ZBP1 deficiency, and, (2) distinguish the immune cell cytokine contribution from the epithelial-cell signaling requirements for cell death execution and dermatitis. Further experiments must determine whether STAIN is sustained when STING signaling is separated from epithelial cells where ZBP1-dependent cell death occurs. The dependency on STING and ZBP1 function in the same executioner compartment would reinforce the use of STAIN, but if function can be separated into different compartments this term has less utility.

Answer:

To address the points raised by referees #1 and #3, we have generated both *Casp8*^{E-KO}; *Sting*^{KO} and *Casp8*^{E-KO}; *Tnfr1*^{KO}; *Sting*^{KO} mice, as suggested by both referees. For the *Casp8*^{E-KO}; *Sting*^{KO} mice, survival analysis of 23 mice has been completed and all mice have already been fully analysed. This shows an additional rescue compared to the *Casp8*^{E-KO}; *Sting*^{E-KO} mice, suggesting a contribution of STING from other cell compartments. For the *Casp8*^{E-KO}; *Tnfr1*^{KO}; *Sting*^{KO} mice a cohort of 20 mice is currently undergoing survival analysis, with 55% of the cohort already characterized, and the remaining mice expected to be characterized by beginning of February 2025.

4. STING Activation and ZBP1-RIPK1-RIPK3 Complex Formation upon loss of Casp8

Referee #1: 3. The authors show biochemically that STING activation alone can trigger assembly of a ZBP1-RIPK1-RIPK3 complex in WT cells (Fig. 2d). Is this complex increased by Casp8 deficiency or inactivation (this data is missing from Fig. 2e)? Presumably the ZBP1-RIPK1-RIPK3 complex is also detected in untreated Casp8 Mki1 DKO cells since these cells contain basal levels of pRIPK3. Does the STING antagonist suppress complex formation as well as basal levels of pRIPK3 in Casp8 Mki1 DKO cells?

Referee #1: 4. The prominent band in the FADD WBs of the ZBP1 IPs in Fig. 2e (light chain of the IP antibody?) make it hard to exclude FADD from these complexes. The authors should either clean up the experiment or remove what is an uninformative FADD WB. To formally prove that complex formation is FADD-independent (as claimed on line 179), the authors should show that the ZBP1-RIPK1-RIPK3 complex is formed in Fadd KO cells.

Answer:

The data presented in Figure 2d were obtained using MLKL deficient MEFs. As suggested by referee #1, similar immunoprecipitation (IP) experiments can be performed in Casp8/MLKL double deficient cells, with or without the presence of the STING antagonist. Additionally, the same IP can be conducted in FADD/MLKL double deficient mice. It is important to note that we have also shown that ZBP1 is not recruited to a FADD containing complex as the immunoprecipitation of FADD does not show any detectable ZBP1 band from the same lysates excluding therefore the necessity of FADD in inducing a ZBP1 containing complex.

5. Assessing the Functional Contribution of ZBP1 to STING Agonist-Mediated Cell Death in Casp8 KO Cells

Referee #1: 5. Given that STING agonist alone doesn't kill WT cells (EDF2a), they should assess the functional contribution of ZBP1 to STING-agonist killing by CRISPR KO of Zbp1 in the Casp8 KO cells in EDF2e. Looking at additive effects of TBZ + STING agonist in EDF2a is messy.

Answer:

The Casp8/ZBP1 double deficient cell lines have already been generated using CRISPR KO and the recommended assays can be performed as suggested.

6. Mechanisms of ZBP1 Activation in Casp8 Deficiency: Exploring Z-NA Sensing

Referee #2: 2. Mechanism by which ZBP1 is activated. Simple induction of ZBP1 by IFNs is – in itself – is probably not sufficient to activate this sensor; ZBP1 likely binds Z-NA species, which drive its activation. In fact, the Green paper mentioned earlier showed that Casp8 loss results in a spontaneous ZNA signal and that a mutant ZBP1 lacking Z-NA sensing capability fails to activate necroptosis when Casp8 is ablated. What are the Z-NA species which activate ZBP1, and how are these ZNAs quenched by Casp8? Or if the mechanism in vivo is independent of ZNA sensing, then how is ZBP1 activated to induce necroptosis.

Answer:

We have already demonstrated that the activation of ZBP1 requires its initial upregulation, which is elicited by STING-mediated IFN responses driven by Casp8 deficiency. As the referee correctly points out, the mere upregulation of ZBP1 does not necessarily confirm its activation.

To address this concern, we are conducting immunofluorescence staining (IF) to detect Z-NAs in wild-type (WT) and Casp8^{KO} MEFs *in vitro*. This is being done following treatment with DNase or RNase to help identify the origin of these Z-NAs. Additionally, we are currently performing these experiments also on tissues to further prove the presence of Z-NAs *in vivo*.

7. Exploring the Role of STING Activation in Necroptosis and Therapeutic Implications for SAVI Patients

Referee #1: Fig. 5 attempts to link STING activation to cell death in humans, but doesn't really hit the mark. Patients bearing activating mutations in STING expressed ZBP1 and MLKL mRNAs at higher levels, but this just indicates that they are necroptosis-competent. It does not prove necroptosis induction. Unless they can show p-MLKL in patient samples, I don't think this dataset is worthy of inclusion, certainly not as a main figure.

Referee #2: 3. Therapeutic implications of these findings for treatment of SAVI patients. The authors have suggested that ZBP1-driven necroptosis may underlie some of the pathology seen in SAVI patients but stop short of exploiting these findings for therapeutic benefit of these patients. Can necroptosis inhibitors (perhaps paired with inflammasome, TNF blockers, and/or JAK inhibitors) prevent skin pathology and lethality in the Casp8E-KO mouse?

Referee #3: A similar ISG-enhanced RNAseq pattern is shown by transcriptionally-derived gene set enrichment analysis (GSEA) in human patients with auto-inflammatory disorder STING-Associated Vasculopathy with onset in Infancy (SAVI). This is likely fortuitous because the mice lack CASP8 in epithelial cells; whereas the cells involved in disease in patients have fully functional CASP8 (as well as CASP10). This observation reveals only that an interferon signature is a very crude comparator. All STING-associated and STING gain-of-function outcomes have long been associated with elevated ISGs (Barber, 2014; Ahn and Barber, 2014) independent of CASP8 compromise. Importantly here, STING has also been observed to drive death pathways in published work, including apoptosis (Gulen et al., 2017) and necroptosis (Chen et al., 2018) and the relationship to ISGs has been of great interest. In these and many other papers have pursued STING-induced death shown to be dependent on RIPK3, MLKL, as well as BH3-only BCL2 family members but independent of RIPK1 (Zheng et al., 2023) although no study has yet hinted whether CASP8 (driving extrinsic apoptosis) or CASP8 compromise (driving necroptosis) contributes. The authors need to better integrate with this understanding of STING signaling.

Human STING gain-of-function disease outcomes certainly involve further inflammatory amplification that remain to be unveiled (Chavin et al., 2023). The demonstration that STING-mediated inflammation drives events autonomously in the epithelial cell or through immune cell-epithelial cell cytokine-mediated crosstalk, intrinsic STING-induction of the ZBP1-RIPK3-MLKL inflammatory pathway is a significant advance once the timing and dissection role of immune and epithelial compartments is clear.

Finally, authors should pursue some evidence whether apoptotic and necroptotic (or other broader) outcomes are contributing, particularly given the observations complex pyroptotic, apoptotic and necroptotic signaling proposed for PANoptosis and the PANoptosome (Man and Kanneganti, 2024; Pandian and Kanneganti, 2024).

Answer:

The acquisition of SAVI patient biopsies from tissues other than blood, such as skin and lung, is simply not feasible due to ethical considerations, especially since these are pediatric patients. The ethical approval for a biopsy in these cases can only be obtained to determine the etiology of a disease which, in this case, can be done via the sequencing of DNA extracted from blood cells. This much less invasive screening already shows STING gain of function mutation rendering the approval for a biopsy impossible under any legislation. We have already inquired for samples in all centers, from any deceased patient and none of these possesses tissues samples or can obtain any from the patients currently under treatment. This limitation is applicable to well-regarded pediatric centers such as the Bambino Gesù Hospital from which the samples for this study were obtained as well as NIH-funded centers in the US.

However, to address this point as well as the concerns raised by Referees #2 and #3, we have generated the commonly used and published murine model for SAVI disease, N153S mice, which serves as an orthologue of the N154S mutation found in the STING gene in human patients. From these mice, we have collected various tissues, including skin, lung, liver and spleen, and performed staining for pMLKL and cl-Casp3 to determine which tissues exhibit necroptosis versus apoptosis upon STING overactivation.

Additionally, we have generated N153S KI mice deficient in RIPK3, either alone or in combination with Casp8 deficiency, to evaluate whether inhibiting necroptosis and/or apoptosis provides a rescue effect in SAVI disease satisfying the rest of Referee #3's points.

Currently, we have five mice N153S/Ripk3^{-/-}, one of which is 4 months old, and the others are under 1.5 months of age. None of these mice have shown any signs of disease.

Also, we currently have N153S Casp8 Ripk3 pups. Additional pups are expected, but at least another 4 months will be required for them to reach endpoint criteria.

8. Proving Necroptosis in Casp8^{E-KO} Mice: Validation of MLKL and RIPK3 Dependence and Mechanistic Clarification.

Referee #1: 1. The authors assume a critical role for necroptosis in the Casp8 E-KO mice without any actual proof. They cite ref. 1 but this study only described inflammation in Casp8 E-KO mice, and did not show that it was due to RIPK3- and MLKL-driven necroptosis. The authors assume necroptosis underlies their phenotype because of what has been described in other models. For example, Ripk3 loss rescues the inflammation in Fadd E-KO mice (PMID: 22000287), Mlkl loss rescues inflammation in Casp8 IEC KO mice (PMID: 32362323), and Mlkl loss rescues inflammation in Casp8 C362S E-KI mice (PMID: 31748744). However, given that there is no Mlkl KO rescue data published for the Casp8 E-KO mice, the onus is on the authors to prove that Mlkl KO Casp8 E-KO mice are similarly rescued. They should also establish whether the up-regulation of p-STAT1 and IFN-regulated genes in Casp8 E-KO mice (Fig. 1d, 3g-i) is independent of necroptosis.

Answer:

The model requested by the referee has been previously published and is referenced in citation #3. In that study, the authors deleted Casp8 in epidermal keratinocytes in Ripk3-deficient mice, demonstrating that Mlkl-driven necroptosis is the driving event in Casp8-deficiency-induced dermatitis and lethality. This model parallels the findings in Casp8^{IEC-KO} mice referenced by the

referee. Additionally, the activation of necroptosis due to loss of Casp8 from epidermal keratinocytes has been shown *in situ* in Fig. 3d, e.

However, if this control mouse line is deemed absolutely essential for the study independently of the already published evidence, we can certainly perform the required cross. However, generating this model would necessitate at least three mouse generations.

The expression levels of the suggested markers pSTAT1 and ISGs such as ZBP1, have been already provided in the stainings of skin from *Casp8^{KO} Ripk3^{KO}* and *Casp8^{KO} Mkl^{KO}* mice, as shown in Figure 1a, showing that upregulation of pSTAT1 and ZBP1 upon loss of Casp8 is independent of necroptosis.

All other remarks without any specific experimental request made by referee 3 can be easily added to the discussion.

Rebuttal letter *Kelepouras et al.*

We would like to thank the reviewers for their thoughtful and constructive comments. Your critical insights and suggestions have substantially contributed to refining our study, not only by helping to clarify key conceptual points and enhancing the mechanistic depth of our findings but also propelled us to expand the depth of our investigation into the regulatory circuitry of necroptosis. In particular, the reviewer feedback not only sharpened the significance of our discovery that STING drives ZBP1-dependent necroptosis independently of TNFR1, but also inspired a much broader understanding of the complexity and physiological relevance of necroptotic signaling in both development and disease in line with the reviewers' requests.

In response to the reviewers' comments, and drawing upon ongoing efforts in our laboratory, we have performed an extensive series of new experiments that provide a comprehensive and mechanistically detailed resolution to all points raised. Notably, this includes the generation and full analysis of seven new, genetically defined mouse models, designed to interrogate the specific roles of STING, cGAS, ZBP1, and RIPK3 in necroptosis, both in developmental and pathological contexts. These provide robust *in vivo* validation and mechanistic dissection of the STING–ZBP1 axis in necroptosis and are now integrated into the revised manuscript, substantially reinforcing our claims and advancing the field's understanding of necroptotic regulation.

We wish to highlight that several of these experiments had already been initiated in our laboratory over a year ago, with the aim of anticipating and supporting potential follow-up studies. The review process provided a timely and valuable opportunity to integrate these ongoing investigations into the current work, allowing us to submit a thorough and well-supported rebuttal in a short time frame.

We would also like to draw your attention to the fact that, as we have made substantial progress in advancing and resubmitting our study to Nature, two well-established competing laboratories have likewise moved forward with the preparation of their manuscripts for re-submission to similarly high-impact journals. This information has been openly communicated to us and is therefore not confidential in nature. Based on data they have presented at conferences/meetings throughout the year, it appears that their findings conceptually and experimentally converge with our demonstration that STING drives ZBP1-mediated necroptosis. However, their studies do not appear to extend these findings to the context of STING-mediated human disease or preclinical models of SAVI, as we have now comprehensively done. While we fully appreciate that our manuscript must be judged solely on its scientific merits, we hope that you will take into account such competitive landscape which despite its imminent nature has in no way compromised our ability or stopped our efforts to thoroughly and rigorously address all of the reviewers' concerns.

The complementary and overlapping reviewer requests, which as you will see have been all answered experimentally, are summarised in additional five major conceptual advances that enhance the conceptual novelty and significance of our study:

- i. Elucidation of a dual-checkpoint architecture governing necroptosis in Casp8-deficient tissues, comprising a primary TNFR1-dependent pathway and a secondary, STING–ZBP1-mediated axis. Through systematic genetic dissection using compound mutant mice we uncover a two-tiered model in which TNFR1 functions as the dominant initiator of necroptosis, licensing MLKL activation via RIPK1–RIPK3 complexes. Upon genetic ablation of TNFR1, STING-driven transcriptional upregulation and functional activation of ZBP1, which engages RIPK3–MLKL, induces necroptosis independently of TNF. Remarkably, in the context of *Casp8^{E-KO}* mice STING co-deletion results in an even more profound rescue than ZBP1 co-deletion alone, or TNFR1 co-deletion alone suggesting that STING amplifies

necroptotic output not only by licensing ZBP1 expression but also by promoting autocrine TNF secretion through NF- κ B activation, thereby sustaining partially the TNFR1 axis itself. This identifies STING as a dual-integrator and amplifier that intersects both checkpoints: transcriptionally initiating ZBP1–MLKL necroptosis and potentiating TNFR1-driven death signaling, thereby acting as a pivotal coordinator of inflammatory necroptosis *in vivo*. This was validated by *Casp8^{E-KO}; Tnfr1^{-/-}*, *Casp8^{E-KO}; Sting^{-/-}* and *Casp8^{E-KO}; Zbp1^{-/-}* crosses.

- ii. STING functions not only as a transcriptional inducer of ZBP1 and MLKL orchestrating type I interferon signaling but also as a functional activator of necroptosis, promoting ZBP1 activation through stabilisation of Z-form nucleic acids (Z-NAs) and facilitating RIPK1–RIPK3 recruitment and activation to the ZBP1 signaling complex.
- iii. Mechanistic and genetic elucidation of STING activation in *Casp8*-deficient cells, traced to cGAS sensing of cytosolic genomic DNA that accumulates due to mitotic instability and DNA damage (not mitochondrial DNA) highlighting a precise molecular cascade from DNA damage/chromosomal instability to inflammatory cell death. This pathway is now genetically validated through *Casp8^{E-KO}; cGAS^{-/-}* and *Casp8^{E-KO}; MAVS^{-/-}* crosses.
- iv. Revelation of a critical cell-extrinsic role for STING, particularly in stromal and hematopoietic compartments, where it sustains systemic necro-inflammation beyond its cell-autonomous functions in keratinocytes. This was uncovered through comparison of tissue-restricted versus systemic STING deletion in compound knockout models (*Casp8^{E-KO}; Tnfr1^{-/-}; Sting^{-/-}* mice).
- v. Translational validation of STING-driven necroptosis in disease: Using the preclinical STING-N153S mouse model of SAVI, we demonstrate that RIPK3 co-deletion rescues thymic, splenic, pulmonary, and hematopoietic abnormalities, establishing necroptosis as a driver of systemic immune dysregulation as well as tissue inflammation. This supports the therapeutic relevance of targeting the STING–ZBP1–necroptosis axis in STING-related autoinflammatory disease.

Each of these findings is now firmly supported by direct genetic evidence and mechanistic validation. Where possible, we have anticipated reviewer concerns and performed orthogonal validation through complementary genetic, biochemical, and immunological assays. Importantly, these findings were not only necessary to address the specific points raised but have also allowed us to reframe our central thesis and present a unified, mechanistically resolved model of STING-induced necroptosis in both development and disease.

We hope the reviewers will appreciate the depth, breadth, and conceptual clarity of the revised manuscript, which we believe now delivers a comprehensive and field-advancing study on the role of STING in inflammatory cell death and immune pathology.

Below you will find our point-by-point rebuttal. Each reviewer question (in italics) is followed by a specific answer highlighting correspondent manuscript lines and article figures describing our newly integrated findings.

Referee #1 (Remarks to the Author):

Question: This study opens with the authors confirming that Casp8 deletion in mouse keratinocytes elicits inflammation and, as reported in MEFs and other mouse tissues, Casp8 loss is associated with an upregulation of IFN-regulated genes and other markers of IFN signaling (e.g. p-STAT1). Data set the stage but are low in terms of novelty. Fig. 2 uses immortalized MEFs and a STING antagonist to link activation of STING to increased p-STAT1 and ZBP1 in Casp8 KO cells. This result is unexpected and therefore genetic validation using Casp8 Sting DKO cells is needed as well as at least one other cell type (see point 2 below).

Answer: We have now generated *Casp8 Sting* DKO cells and blotted them for expression of ZBP1 and pSTAT1. Our results show that genetic deletion of STING completely prevents pSTAT1 and also expression of ZBP1. This is now included in Extended Data Fig. 2a and described in line 134 of the main manuscript. With regards to other cell type we have responded to this concern in point 2 as indicated by this reviewer.

Question: It is also unclear why STING activation induces significant death in Casp8 KO MEFs (EDF2e) without altering ZBP1 or MLKL levels significantly (Fig. 2e); even without treatment there is aberrantly high expression of ZBP1 and p-STAT1. In other words, what else is STING activation doing to trigger the death of the Casp8 KO cells? Does STING antagonist suppress basal p-STAT1 and ZBP1 in the Casp8 KO cells in Fig. 2e?

Answer:

We would like to thank the reviewer for this question.

The work that we have carried out during the revision period and the previously submitted work has now highlighted 3 important aspects that precisely describe how STING triggers cell death in *Casp8* deficient cells beyond the expression of ZBP1 and STAT1 activation. Importantly these data also provide significant insights on how the treatment with the STING agonist sensitises to cell death in the context of *Casp8* deficient MEFs as well as epidermal keratinocytes *in vivo*.

We have grouped these data into three main categories:

(i) STING upregulates not only ZBP1 but is also responsible to enhance the expression of its natural ligand Z-NAs (both Z-DNA and Z-RNA); (ii) The consequent overactivation of ZBP1 is reflected in enhanced RIPK1 and RIPK3 activation shown in ZBP1 immunoprecipitates; (iii) We find that STING contributes partially to autocrine TNF production and activation of inflammatory responses that are common to those activated by TNFR1. Below we have described in detail the results that we have gained which led to the above-mentioned conclusions.

(i) STING agonism leads to enhanced upregulation of Z-NAs

To answer a question from reviewer 2 regarding how is ZBP1 activated, we utilised *Casp8*^{-/-}, *Casp8*^{-/-}; *Mlkl*^{-/-} and *Casp8*^{-/-}; *Ripk3*^{-/-} MEFs and performed immunofluorescent staining using a validated protocol for detecting Z-form nucleic acids (Z-NAs) optimised by the Balachandran lab utilizing a previously validated and published antibody^{1,2}. Our results revealed specific cytosolic accumulation of both Z-DNA and Z-RNA in these cells, with the Z-NA signal being completely abolished only when both DNase and RNase were applied, confirming the presence of both nucleic acid species in the Z-conformation.

We next assessed the co-localisation between ZBP1 and Z-NAs. Clear spatial overlap was observed, indicating physical proximity and likely interaction. Importantly, treatment with the STING agonist DMXAA further enhanced Z-NA signal intensity, suggesting that STING activation promotes

accumulation and/or stabilisation of Z-NAs, the natural ligands of ZBP1. Given the lack of alternative direct readouts for ZBP1 activation, and the consistent co-localisation with Z-NA under conditions of STING agonism, we conclude that STING enhances ZBP1 activation via increased generation or exposure to Z-NAs. This correlates well with our biochemical findings indicating that STING activation leads to increased recruitment and phosphorylation of RIPK1 and RIPK3 in ZBP1 immunoprecipitates (discussed below in more details), further supporting functional activation of ZBP1 in this context. These data are now included in Extended Data Fig 10 and added to the manuscript in lines 505-522 of the main manuscript.

(ii) STING agonism leads to enhanced levels of active RIPK1 and active RIPK3.

In the previous version of the manuscript, we showed that *Casp8*^{-/-} MEFs treated with the STING agonist DMXAA exhibit a clear sensitivity to cell death, despite no major change in total ZBP1 or MLKL levels as noted by this reviewer (current Fig. 1d-g). We have now expanded this analysis by including *Casp8/MLKL* and *FADD/MLKL* double-knockout MEFs (Fig. 1h). In both cell types, we observed that STING activation significantly enhances phosphorylation of RIPK1 and RIPK3 ZBP1 immunoprecipitates, suggesting that STING not only promotes necroptosis via upregulation of ZBP1 but also via enhancing the activation of the kinases recruited to the ZBP1-RIPK1-RIPK3 complex. Since STING agonism leads to enhanced Z-NAs accumulation which colocalize with ZBP1 (Extended Data Fig.10) we concluded that the over activation of ZBP1 leads to enhanced activation of RIPK1 and RIPK3.

Importantly Sting antagonism (C-178 treatment) completely depletes pSTAT1 and ZBP1 expression. This was previously shown and now displayed in Figure 1e-f, where we show that, in *Casp8* KO, *Casp8/MLKL* DKO and *Casp8/RIPK3* DKO cells, treatment with C-178 alone could completely deplete the endogenous expression of ZBP1 and pSTAT1. Furthermore, the specificity of the C178 treatment has been validated in the *Casp8/STING* DKO cells (Extended Data Fig 2.a). Similar results are also now shown in the new (different clone) *Casp8/MLKL* DKO, *FADD/MLKL* DKO and *MLKL* KO cells where co-treatment of C-178 with DMXAA completely abolished both endogenous and DMXAA induced ZBP1 expression and pSTAT1 levels which consequently completely prevents the immunoprecipitation of ZBP1 in those treated samples (Fig. 1h). These data are now discussed in lines 129-139 and lines 176-196 of the main manuscript.

(iii) STING activation leads to autocrine production of TNF and activation of inflammatory pathways.

In the previous version of the manuscript, we reported that STING agonism in *Casp8*^{-/-} cells could be partially rescued by Etanercept, suggesting that STING activation may drive autocrine TNF production (current Extended Data Fig. 2g) which is consistent with the known ability of STING to activate both canonical and non-canonical NF-κB signalling. At the time of initial submission, this observation was based solely on pharmacological inhibition and was therefore not emphasised. This is mentioned in lines 158-160 of the manuscript. We have now substantiated this observation through genetic crosses that provide evidence of a role of STING in TNF-induced cell death as previously highlighted by the DMXAA experiment. Specifically, we generated *Casp8*^{E-KO}; *Sting*^{-/-} mice, which survive significantly longer than *Casp8*^{E-KO}, *Tnfr1*^{-/-} animals displaying a significant delay in the onset and progression of lethal dermatitis and systemic inflammation (Fig. 4a-b). Since STING activation drives ZBP1-mediated necroptosis, to further understand the actual contribution of STING and genetically compare it to ZBP1, we also generated *Casp8*^{E-KO}; *Zbp1*^{-/-} mice. Our original expectation was to observe a similar survival as obtained in the *Casp8*^{E-KO}; *Sting*^{-/-} mice. Surprisingly, *Casp8*^{E-KO}; *Zbp1*^{-/-} mice displayed lethal dermatitis with identical onset survival and severity as the *Casp8*^{E-KO} cohort (Fig. 4a-b), as confirmed by histopathology and survival analysis (Extended Data Fig. 7a-c), underscoring that ZBP1 loss alone was insufficient to delay disease onset, progressions and severity. Since, *Casp8*^{E-KO}; *Tnfr1*^{-/-}; *Zbp1*^{-/-} mice exhibited full rescue, with complete absence of dermatitis and systemic inflammation (Fig. 3), we concluded that (i) TNF/TNFR1 acts as the initial necroptosis checkpoint that must be bypassed to engage into ZBP1 mediated necroptosis, (ii) ZBP1-

mediated necroptosis acts as a second, independent, necroptotic checkpoint, (iii) given the extended survival of the *Casp8^{E-KO}; Sting^{-/-}* mice encompassing and surpassing the survival of the *Casp8^{E-KO}; Tnfr1^{-/-}* mice, STING signaling not only licenses the ZBP1 necroptotic checkpoint but also partially contributes to TNF-driven cell death and inflammation, likely through autocrine TNF production and NF- κ B pathway engagement. These data are now included in the paper in lines 384-422. The partial contribution of STING to TNFR1-induced necroptosis is also evidenced by the fact that the *Casp8^{E-KO}; Sting^{-/-}* mice are only fully rescued upon co-deletion of TNFR1 (Fig. 5). We, in fact, also generated *Casp8^{E-KO}; Tnfr1^{-/-}; Sting^{-/-}* mice which presented a complete rescue of the lethal dermatitis as shown in the *Casp8^{E-KO}; Tnfr1^{-/-}; Zbp1^{-/-}* mice resolving the previously observed residual systemic inflammation that leads to the lethality of the *Casp8^{E-KO}; Tnfr1^{-/-}; Sting^{E-KO}* mice. These data are included in the manuscript and described in lines 424-452 and fully discussed below to answer another separate question from this reviewer.

Taken together, our data support a model in which STING agonism leads to ZBP1 activation and consequent activation of RIPK1 and RIPK3. Moreover, STING agonism in cells revealed also the involvement of STING in autocrine TNF production as treatment with etanercept slightly inhibited cell death in *Casp8^{-/-}* cells treated with DMXAA. *In vivo*, we have confirmed this completely unknown role that slightly overlaps with TNF induced cell death as STING not only promotes ZBP1 expression and activation enabling ZBP1-dependent necroptosis, but also simultaneously contributes to TNF production that can trigger TNFR1-mediated cell death.

Question: Fig. 3 shows that Sting or Tnfr1 loss delays but does not prevent skin inflammation in Casp8 E-KO mice. Fig. 4 shows that the combined loss of Sting and Tnfr1 is much more effective at delaying disease in Casp8 E-KO mice, but is not as good as the combined loss of Zbp1 and Tnfr1. Given that STING loss is restricted to keratinocytes and ZBP1 loss is full body, the interpretation of this difference is uncertain.

Answer:

We thank the reviewer for highlighting this important point. In our revised manuscript, we have now crossed these mice with *Tnfr1^{-/-}; Sting^{-/-}* to generate *Casp8^{E-KO}; Tnfr1^{-/-}; Sting^{-/-}* mice in which STING was deleted systemically, rather than restricted to keratinocytes as in the *Casp8^{E-KO}; Tnfr1^{-/-}; Sting^{E-KO}* animals (Fig. 5 and Extended Data Fig. 8). Strikingly, these mice were completely protected from all pathological manifestations and remained viable, healthy, and phenotypically indistinguishable from control littermates and *Casp8^{E-KO}; Tnfr1^{-/-}; Zbp1^{-/-}* (Fig. 5). This full rescue stands in stark contrast to the partial and transient protection observed in *Casp8^{E-KO}; Tnfr1^{-/-}; Sting^{E-KO}* mice, which showed delayed but ultimately lethal dermatitis driven by systemic inflammation and immune cell infiltration (Fig. 3 and Extended Data Fig. 5 and 6). We now provide comparison between *Casp8^{E-KO}; Tnfr1^{-/-}; Sting^{E-KO}* and *Casp8^{E-KO}; Tnfr1^{-/-}; Sting^{-/-}* mice at 12 week time points (shortly before *Casp8^{E-KO}; Tnfr1^{-/-}; Sting^{E-KO}* mice reach their survival end point) showing histologically the inhibition of lethal dermatitis in the skin as evidenced by loss of pMLKL, ZBP1 expression, immune cell infiltration via CD45 staining and c-Casp3-positive cells coupled by the evident reduction in skin thickening in the *Casp8^{E-KO}; Tnfr1^{-/-}; Sting^{-/-}* mice (Fig. 5 and Extended Data Fig. 8). Moreover, whilst *Casp8^{E-KO}; Tnfr1^{-/-}; Sting^{E-KO}* mice at 12 weeks of age revealed significant splenomegaly, thymic atrophy, hepatic inflammation, and hematologic abnormalities, all indicative of severe systemic disease, *Casp8^{E-KO}; Tnfr1^{-/-}; Sting^{-/-}* lung, spleen and thymus appeared indistinguishable from littermate controls. Consistently, peripheral blood analysis showed normalisation of all previously elevated markers, confirming effective suppression of the inflammatory response. Importantly, constitutive, full-body deletion of STING in *Casp8^{E-KO}; Tnfr1^{-/-}; Sting^{-/-}* mice abrogated both local and systemic disease, phenocopying the complete rescue observed in *Casp8^{E-KO}; Tnfr1^{-/-}; Zbp1^{-/-}* mice. This striking phenotypic divergence reveals a key conceptual point: although STING activation in

keratinocytes is essential to initiate ZBP1-mediated necroptosis and early inflammation, STING expression in non-keratinocyte compartments, most notably immune and stromal cells, also plays a pivotal role in sustaining and propagating systemic necro-inflammation. These findings demonstrate that STING contributes not only to the cell-intrinsic transcriptional induction of ZBP1 in keratinocytes but also to the broader, systemic necroptotic environment that exacerbates immune-driven inflammation. This dual role of STING, as both a trigger and amplifier of necroptosis, highlights its essential and non-redundant function in TNFR1-independent, ZBP1-mediated pathology. These data have been included into the manuscript (lines 424-452) and shown in Fig.5.

Question: Why STING is active in Casp8 KO cells/mice is unclear. While it might be beyond the scope to address this in the Casp8 E-KO model, is cGAS driving STING activation and p-STAT1 in Casp8 Mkl1 DKO cells? A quick Crispr expt in MEFs or BMDMs would be informative.

Answer:

We thank the reviewer for raising this important mechanistic question regarding the upstream signals responsible for STING activation in Casp8-deficient cells. While the reviewer suggested that a simple CRISPR experiment in MEFs or BMDMs might clarify the role of cGAS, we have undertaken a comprehensive and systematic investigation that goes significantly beyond this initial request. This work would, in isolation, justify a dedicated study, which represents a parallel to the establishment a follow-up project, involved generation of two *in vivo* models and several *in vitro* experiments which we felt were necessary to properly answer and anticipate any further question from this reviewer. Nevertheless, we chose to incorporate these findings into the current manuscript in order to fully address the reviewer's concern and strengthen the mechanistic underpinnings of our proposed model. These data are now included in the manuscript in lines 454-503 and in Extended Data Fig. 9.

To directly test the involvement of cGAS in STING activation, we generated *Casp8^{E-KO}; cGAS^{-/-}* mice. cGAS co-deletion fully phenocopied the protection observed in *Casp8^{E-KO}; Sting^{-/-}* mice, both in terms of survival and suppression of tissue inflammation via histological analysis (Extended Data Fig. 9a, b). These results indicate that activation of STING following Casp8 loss is entirely dependent on cGAS, placing cytosolic DNA sensing upstream of the type I interferon response and necroptosis in this context.

To exclude alternative nucleic acid sensing pathways, we also generated *Casp8^{E-KO}; Mavs^{-/-}* mice. In contrast to the cGAS deletion, MAVS deletion failed to rescue lethality or reduce tissue pathology, as confirmed by survival analysis and histological examination of the skin. These data exclude a contribution of the RIG-I–MAVS RNA sensing axis and establish the cGAS–STING DNA sensing pathway as the exclusive driver of interferon induction in Casp8-deficient animals (Extended Data Fig. 9a-c).

Next, we sought to identify the origin of the DNA species activating cGAS. We first examined the potential role of mitochondrial DNA (mtDNA), which is a known cGAS agonist. Immunofluorescence staining in *Casp8^{-/-}*, *Casp8^{-/-}; Mkl1^{-/-}* and *Casp8^{-/-}; Ripk3^{-/-}* MEFs revealed cytosolic accumulation of dsDNA, but no detectable co-localisation with the mitochondrial transcription factor A (TFAM).

MtDNA accumulation was excluded as treatment with the inhibitor of mitochondrial DNA polymerase γ : 2',3'-dideoxycytidine (DDC) and the inhibitor of mitochondrial RNA polymerase POLRMT (IMT1B) showed no reduction in DNA staining following 96 hours treatment (Extended Data Fig. 9e). Successful depletion of mtDNA was confirmed by RT-qPCR targeting multiple mitochondrial-encoded genes (*Dloop*, *mtNd5*, *mtRnr2*, *mtCo1*, *mtAtp6*, *mtNd2*), all of which were markedly reduced following treatment to a level that was almost negligible (Extended Data Fig. 9f). Strikingly, despite such near-complete ablation of mitochondrial gene expression, there was no reduction in ISG transcription (*Zbp1*, *Rsad2*, *Isg15*, *Ifit1*); in fact, in some cases ISG levels were paradoxically increased (Extended Data Fig. 9g). At the protein level, western blot analysis showed selective

depletion of mitochondrial-encoded proteins following treatment with ddC or IMT1B (Extended data Fig. 9h). Specifically, COX1 and NDUFB8, two proteins encoded by the mitochondrial genome, were substantially reduced in all MEFs following treatment, confirming effective inhibition of mtDNA replication. In contrast, nuclear-encoded mitochondrial proteins such as ATP5A, UQCRC2, SDHB, and remained relatively stable, indicating that mitochondrial biogenesis and integrity are only partially compromised and that mitochondrial depletion is specific rather than globally toxic (Extended data Fig. 9h). Notably, despite successful suppression of COX1 and NDUFB8, neither ZBP1 expression nor STAT1 phosphorylation appear reduced, in fact, both remained robust or even slightly enhanced under these conditions (Extended Data Fig. 9h). This clearly shows that mtDNA depletion does not attenuate the STING-driven interferon response. In contrast, treatment with the STING antagonist C-178 completely abrogates pSTAT1 and ZBP1 expression, even when mitochondrial proteins remain intact. Together, these results provide strong biochemical evidence that mitochondrial DNA is not the trigger for STING activation in Casp8-deficient cells and support a model in which another source, most likely genomic DNA, drives cGAS–STING signaling in this setting.

Having excluded mtDNA, we turned to nuclear DNA as a candidate trigger. Prior work has shown that Casp8 deficiency induces chromosomal instability³ and DNA damage⁴, which can result in nuclear DNA leakage into the cytosol known to be a potent activator of cGAS. In agreement with this, we observed that *Casp8^{E-KO}* skin exhibited mitotic abnormalities, with many pHH3-positive cells (mitotic marker) displaying aberrant chromosome segregation (Extended Data Fig. 9i). Moreover, immunostaining for p-γH2AX revealed a significant increase in DNA damage foci, consistent with chromosomal instability (Extended Data Fig. 9i).

In conclusion, through the generation of different mouse models, use of genetic and pharmacological tools, and detailed molecular analyses, we provide a comprehensive mechanistic explanation for how STING is activated in Casp8-deficient contexts. Our data firmly place nuclear DNA–induced cGAS activation upstream of STING, IFN signaling, ZBP1 induction, and ultimately necroptosis. We hope the reviewer will appreciate the depth and scope of these investigations, which far exceed the original experimental suggestion, and represent a significant conceptual and technical advance in our understanding of innate immune activation downstream of Casp8 loss.

Question: Fig. 5 attempts to link STING activation to cell death in humans, but doesn't really hit the mark. Patients bearing activating mutations in STING expressed ZBP1 and MLKL mRNAs at higher levels, but this just indicates that they are necroptosis-competent. It does not prove necroptosis induction. Unless they can show p-MLKL in patient samples, I don't think this dataset is worthy of inclusion, certainly not as a main figure.

Due to ethical limitations, we were unable to obtain paediatric tissue samples that would allow *in situ* detection of human pMLKL. Moreover, the currently available antibodies for pMLKL staining are specific to the murine phosphorylation site (S345 in mouse MLKL), which differs from the human site (S358), and no antibody validated for detection of human pMLKL by immunohistochemistry is currently available. As a result, direct histological confirmation of necroptosis in patient samples could not be performed even if such tissue samples were available or ethically attainable.

However, we emphasise that this technical limitation applies broadly to human tissue research, especially in paediatric or rare disease cohorts. In such cases, transcriptional profiling has become a standard and highly valuable strategy for uncovering disease-associated molecular programs. While correlative, these data routinely form the basis for mechanistic hypotheses that are subsequently tested in preclinical models and have, in many cases, led to the design of clinical trials, particularly in oncology and immunology, even as a standalone. Indeed, transcriptome-based

insights are now foundational to biomarker discovery and patient stratification efforts in cancer immunotherapy, interferonopathies, and inflammatory diseases.

Importantly, the finding that STING can drive necroptosis independently of TNFR1 signaling adds a crucial mechanistic layer to current understanding of this inflammatory pathology. This discovery may help explain the disappointing outcomes of TNF inhibitors in treating STING-driven diseases, including SAVI. While TNF blockade has proven effective in other settings, such as rheumatoid arthritis and inflammatory bowel disease, its failure in STING-associated diseases suggests that upstream inhibition is insufficient when a parallel, TNF-independent axis of necroptosis remains active.

Our data support the idea that STING functions not only as a transcriptional driver of ZBP1 but also as a direct inducer of ZBP1-dependent necroptosis bypassing TNF entirely. From a translational standpoint, this argues for the development or repurposing of necroptosis pathway inhibitors (e.g., RIPK3 or MLKL inhibitors) as adjunct or alternative therapies. Such agents could be particularly effective in combination with TNF blockade, enabling suppression of both canonical and STING-mediated necroptotic pathways. Given the tissue-specific and multifactorial nature of inflammation in diseases like SAVI, a combinatorial strategy targeting both axes may offer greater efficacy than monotherapy. We believe this concept, grounded in our genetic and mechanistic dissection of necroptosis regulation, provides a new rationale for therapeutic intervention in STING-driven inflammatory diseases. Within this context, we believe that the transcriptional signatures observed in STING-mutant patients, particularly the upregulation of ZBP1 and MLKL, are of high translational relevance and mechanistic significance, especially when considered alongside functional validation in complementary murine models. To this end, we turned to the established preclinical SAVI mouse model bearing the heterozygous N153S STING gain-of-function mutation (GOF), hereafter referred to as N153S mice. These mice develop lethal systemic inflammation with severe dermatitis, interstitial lung disease, and splenomegaly, faithfully recapitulating key clinical features of human SAVI (Fig. 6d and Extended Data Fig. 14a-c).

Although full disease penetrance is observed in only ~35% of mice, we analysed several N153S mutant mice presenting with the complete clinical manifestation (Fig. 6e). Blood analysis confirmed the reported lymphopenia (Extended Data Fig. 14d). Tissues harvested at endpoint revealed robust ZBP1 upregulation in skin, lung, thymus and spleen (Fig. 6f). Importantly, necroptosis, detected via pMLKL- staining, was restricted to skin, while lung and thymus displayed both pMLKL-S345 and cleaved Caspase-3 positive cells (Fig. 6f). Spleens exhibited strong cleaved Caspase-3 staining only, indicative of dominant apoptosis in that compartment at endpoint (Fig. 6f).

STING gain-of-function mutation have been shown to result in the observed phenotypes by triggering early-onset immune dysregulation (6-8 weeks) which is considered essential for disease pathogenesis^{5,6}. Crucially, bone marrow transplantation experiments have shown that hematopoietic-derived cells from SAVI mice are sufficient to confer the full immune phenotype and tissue pathology to healthy recipients, establishing hematopoietic-intrinsic STING activation as a major driver of disease⁶. Previous studies focusing on the immunophenotyping of the SAVI mice reveal a marked expansion of myeloid cells, particularly CD11b⁺ cells and Ly6G^{hi} neutrophils, alongside a significant reduction in mature lymphocyte subsets, including TCR-β⁺ T cells and CD19⁺ B cells in the spleen. Residual T and B cells display an activated phenotype, with upregulation of markers such as CD69 and MHC class II⁶. Thymic development is impaired, with reduced total and double-negative thymocytes. Beyond lymphoid and myeloid lineages, SAVI mice also display significant disruptions in erythropoiesis. This is reflected by a reduction of immature erythroid cells in the bone marrow and a compensatory expansion of Ter119⁺CD71⁺ immature erythrocytes in the spleen, indicative of extramedullary erythropoiesis which is a classical marker of systemic inflammation⁶. This redistribution reflects broader hematopoietic imbalance caused by STING hyperactivation and underscores how immune dysregulation extends beyond leukocyte compartments to affect the entire hematopoietic system. Collectively, these findings highlight that

the immune cell alterations seen in SAVI mice are not secondary consequences but fundamental pathogenic mechanisms driving the systemic inflammatory disease phenotype.

To determine the role of necroptosis in disease pathogenesis and confirm preclinically the translational relevance of our findings, we crossed the *N153S* mice with *Ripk3*^{-/-} mice and performed high-resolution immunophenotyping of the lung, spleen, and thymus via multi parametric flow cytometry.

Our analysis revealed that co-deletion of RIPK3 significantly ameliorates the immune abnormalities induced by STING GOF (Extended Data Fig. 14e-k). In the thymus, *N153S* mice showed reduced organ size and a profound loss of double-negative (DN) thymocytes, particularly within the DN2 and DN3 subsets, indicative of impaired T cell development (Extended Data Fig. 14e, f). Notably, these defects were substantially rescued in *N153S; Ripk3*^{-/-} mice, as reflected by recovery of DN thymocytes. Moreover, CD8⁺, and double-positive (DP) thymocytes, along with their activation status (CD69 expression), were significantly normalized in *N153S; Ripk3*^{-/-} mice, pointing to a contributing role for necroptosis in thymic T cell attrition, however, not complete.

In the lung, where SAVI pathology often manifests with both inflammation and fibrosis, RIPK3 co-deletion mitigated the expansion of inflammatory monocytes and macrophages, antigen-presenting dendritic cells and granulocytes (Extended Data Fig. 14g, h). Although T cell restoration was more modest, the overall pattern again indicated a broad rescue of immune homeostasis. These data reinforce the systemic nature of necroptosis-mediated pathology in this model.

In the spleen, RIPK3 co-deficiency led to striking improvements in both the lymphoid and myeloid compartments (Extended Data Fig. 14i, j). *N153S* mice exhibited severe T cell and NK cell lymphopenia accompanied by expansion of antigen-presenting dendritic cells, granulocytes, inflammatory monocytes and macrophages, which are all hallmarks of chronic STING activation. These aberrations were consistently and significantly rescued in *N153S; Ripk3*^{-/-} mice. Importantly, the frequency of immature erythroid precursors (Ter119⁺CD71⁺) was also reduced, indicating a reversal of extramedullary erythropoiesis and systemic inflammation (Extended Data Fig. 14k). A similar normalisation was observed in bone marrow derived and splenic myeloid subsets of the spleen, confirming that RIPK3-dependent necroptosis is a major driver of hematopoietic dysregulation (Extended data Fig. 14k).

Interestingly, although we did not detect pMLKL-S345 staining in the spleens of *N153S* mice at endpoint, this does not contradict the involvement of necroptosis. Rather, it highlights the temporal disconnection between the execution of necroptosis and its histological detection in tissues with high cellular turnover. Our immunophenotyping was performed at 6-8 weeks of age and therefore prior to the onset of terminal pathology, capturing early-stage immune cell depletion that is necroptosis-dependent. In contrast, the pMLKL-S345 stainings were conducted at endpoint, a time when early necroptotic events have likely already occurred and cleared. Importantly, at this time, fibrosis, and compensatory immune remodelling dominate the tissue architecture and are responsible or caused by cell death driven also by non-immune cells which in the skin is purely necroptotic, in the lung is both apoptotic and necroptotic and in the spleen is exclusively apoptotic. Thus, we interpret the absence of splenic pMLKL-S345 at endpoint as a reflection of the aftermath of earlier necroptotic damage, rather than its absence.

Moreover, we propose that early necroptosis of immune cells including but not limited to the spleen, particularly T cells, contributes directly to the loss of immune homeostasis and the unleashing of systemic inflammation. This cascade likely facilitates the organ-specific inflammatory pathologies observed in SAVI, including dermatitis, interstitial lung inflammation, and splenomegaly. In this way, necroptosis acts not only as a mechanism of immune cell loss, but also as a driver of systemic disease progression, reinforcing the functional and translational relevance of our findings.

Importantly, our comprehensive immune profiling goes well beyond prior reports, which either focused on a narrow subset of immune populations or did not include side-by-side analysis across

multiple organs. Our study shows multi-organ immunophenotyping using high-dimensional flow cytometry across spleen, thymus, and lung capturing dynamic changes in both lymphoid and myeloid subsets, along with functional markers such as CD69, MHC-II, and progenitor signatures. In contrast, previous reports relied primarily on limited flow cytometric markers and did not interrogate tissue-specific effects on haematopoiesis, thymic development, or extramedullary inflammation with comparable resolution. Our dataset provides clear evidence that RIPK3 deletion and therefore necroptosis inhibition restores T cell populations, normalises erythropoiesis, and mitigates thymic developmental arrest, prevents disruption of haematopoiesis, lymphoid development, and peripheral immune balance findings not described in earlier analyses of SAVI models.

Taken together, these findings provide direct genetic evidence that necroptosis is not merely a downstream consequence of inflammation in SAVI, but rather a central effector mechanism driving immune cell depletion, compensatory myelopoiesis, and systemic immunopathology. They also reveal that necroptosis affects both adaptive and innate compartments in a tissue-specific manner, offering a mechanistic explanation for the organ-selective phenotypes observed in both mice and patients with STING-activating mutations. These data are now included in Fig. 6 and Extended Data Fig. 14 and reported in line 627-707, with the gating strategy outlined in Extended Data Fig. 15.

Question: Overall, I think the results presented are tantalizing but preliminary. Key controls and evidence are missing as indicated above and in the specific points below.

Other issues that need addressing:

1. *The authors assume a critical role for necroptosis in the Casp8 E-KO mice without any actual proof. They cite ref. 1 but this study only described inflammation in Casp8 E-KO mice, and did not show that it was due to RIPK3- and MLKL-driven necroptosis. The authors assume necroptosis underlies their phenotype because of what has been described in other models. For example, Ripk3 loss rescues the inflammation in Fadd E-KO mice (PMID: 22000287), Mlkl loss rescues inflammation in Casp8 IEC KO mice (PMID: 32362323), and Mlkl loss rescues inflammation in Casp8 C362S E-KI mice (PMID: 31748744). However, given that there is no Mlkl KO rescue data published for the Casp8 E-KO mice, the onus is on the authors to prove that Mlkl KO Casp8 E-KO mice are similarly rescued. They should also establish whether the up-regulation of p-STAT1 and IFN-regulated genes in Casp8 E-KO mice (Fig. 1d, 3g-i) is independent of necroptosis.*

Answer: The role of MLKL and hence necroptosis in the Casp8^{E-KO} mice has been recently published by the Pasparakis lab⁷. There, the authors cross Casp8^{E-KO} mice with a phospho-mutant MLKL (S345A) mouse completely rescuing the lethal dermatitis causative of the post-natal lethality. This is reported in Figure 1F. Although these are not data generated by our lab, we hope that this is sufficient proof that loss of Casp8 in the epidermal keratinocytes leads to lethal necroptosis. With regards to the upregulation of p-STAT1 and interferon response, the previous version of our manuscript already showed that Casp8/ RIPK3 knock out mice present pSTAT1 and ZBP1 upregulation (this was our starting point and remains displayed in Figure 1A), moreover in every single Casp8 knockout cell line tested we could observe upregulation of interferon response in a STING-dependent manner independently of necroptosis i.e.: in absence of RIPK3 or MLKL (Fig. 1d-f and h). We believe that the generation of these mice to answer this point would be superfluous and ethically not justifiable.

2. *A reference establishing the specificity of the STING antagonist C-178 should be provided on line 124. Ideally, the authors validate the results in Fig. 2 by CRISPR knockout of Sting in the Casp8 KO MEFs. They should also not rely entirely on immortalized MEFs for drawing conclusions because these cells acquire other mutations/epigenetic changes in a random fashion. Validation of these results in another primary cell type, such as bone marrow-derived macrophages, is needed.*

Answer: We thank the reviewer for this thoughtful comment, which we have addressed through a series of additional experiments

Point 1: Reference for C-178 is now provided in line 129 (PMID: 29973723).

Point 2: The results shown in the previous Fig. 2 (now Fig. 1) have been validated by STING knockout in *Casp8* deficient cells and included in the manuscript in Extended data Fig. 2a and described in line 135. Western blot analysis shows clearly that following STING deletion both ZBP1 expression and STAT1 activation are completely ablated.

Point 3: Due to embryonic lethality at E10.5, it is technically impossible to isolate primary *Casp8*^{-/-} cells. We considered knocking out *Casp8* in primary cells; however, the requirement for selection and potential immortalisation would compromise the primary nature of these cells, partially reproducing conditions already tested in our immortalised MEFs and thus undermining the reviewer's legitimate request for this validation. To answer the reviewer request we assessed the response to STING agonism in multiple primary cell types, including primary wild-type mouse embryonic fibroblasts (WT-MEFs), primary wild-type lung fibroblasts, and bone marrow-derived macrophages (BMDMs). In all these cells line we can show that STING agonism enhances TNF induced necroptosis (in combination with zVAD) confirming the role of STING in promoting necroptotic cell death across distinct cellular contexts. These findings are now presented in Extended Data Fig. 2 d, complementing our existing data obtained in immortalised MEFs, as well as human colon cancer HT29 and human keratinocytes (HaCaT) cells (Extended Data Fig. 2e, f). Taken together, these experiments provide robust evidence that STING-mediated necroptosis is not restricted to immortalised systems and is preserved in physiologically relevant primary cell types. These data are reported in lines 144-151 and Extended Data Fig. 2d.

3. The authors show biochemically that STING activation alone can trigger assembly of a ZBP1-RIPK1-RIPK3 complex in WT cells (Fig. 2d). Is this complex increased by Casp8 deficiency or inactivation (this data is missing from Fig. 2e)? Presumably the ZBP1-RIPK1-RIPK3 complex is also detected in untreated Casp8 Mkl DKO cells since these cells contain basal levels of pRIPK3. Does the STING antagonist suppress complex formation as well as basal levels of pRIPK3 in Casp8 Mkl DKO cells? 4. The prominent band in the FADD WBs of the ZBP1 IPs in Fig. 2e (light chain of the IP antibody?) make it hard to exclude FADD from these complexes. The authors should either clean up the experiment or remove what is an uninformative FADD WB. To formally prove that complex formation is FADD-independent (as claimed on line 179), the authors should show that the ZBP1-RIPK1-RIPK3 complex is formed in Fadd KO cells.

Answer: We have now used both *Casp8/MLKL* double deficient cells and *FADD/MLKL* double deficient cells. The ZBP1-RIPK1-RIPK3 complex is already detectable in untreated conditions. Importantly as highlighted above, treatment with the STING agonist DMXAA further enhance the incorporation of both active RIPK1 and RIPK3 in the complex. Moreover, co-treatment of DMXAA with C-178, and hence the presence of the combination of the STING agonist with the STING antagonist completely prevents complex formation in all cells. Importantly, here we show that the formation of the ZBP1-RIPK1-RIPK3 complex is independent of FADD as cells devoid of FADD expression, actually show the formation of much bigger complex suggesting that expression of FADD and therefore the availability of a FADD containing platform might compete with the ZBP1-RIPK1-RIPK3 complex formation. Moreover, we confirm in these experiments that FADD is not recruited to the ZBP1 complex as in all treatment and cell lines we could not detect FADD following ZBP1 pull down. The control lysate utilised for the detection of FADD is clearly running at a higher molecular weight compared to the detected light chain. These data are now included in Fig. 1h and described in lines 176-188 and are in line with also with our previous experiments indicating that FADD pulldown could not immunoprecipitate ZBP1 (Figure 1g).

Question: 5. Given that STING agonist alone doesn't kill WT cells (EDF2a), they should assess the functional contribution of ZBP1 to STING-agonist killing by CRISPR KO of Zbp1 in the Casp8 KO cells in EDF2e. Looking at additive effects of TBZ + STING agonist in EDF2a is messy.

Answer: We have now generated Casp8/ZBP1 double deficient MEFs. These cells show no further sensitivity when treated with STING agonist beyond their sensitisation to autocrine production of TNF which has already been discussed above and in this experiment shown to be rescued by treatment with etanercept. These data are not included in Extended Data Fig. 2h and described in lines 161-164.

Question: 6. Line 213 – what is “necroptosis-driven secondary apoptosis?”

Answer: In the context of keratinocytes necroptosis, cell death can drive recruitment of immune cells which can in turn propagate further necroptosis but also apoptosis in non-keratinocyte cell populations such as dermal fibroblasts and other immune cells.

Question:7 Line 410 – isn't there a 3rd possibility – ruxolitinib exposures were insufficient to completely block IFNAR signaling? It is hard to conclude anything definitive.

Answer: We respectfully disagree with the reviewer's suggestion that insufficient Ruxolitinib exposure may explain the persistence of interferon signaling in our model. The concentrations used for these paediatric patients reflect clinically approved dosing regimens and are consistent with those shown to achieve therapeutic JAK1/2 inhibition in IFNAR-driven diseases.

Specifically, Ruxolitinib has been approved for the treatment of myelofibrosis, *polycythaemia vera*, and steroid-refractory graft-versus-host disease, where it is used at the same doses that achieve robust systemic inhibition of JAK1 and JAK2. In the context of interferonopathies, including SAVI, published clinical studies and compassionate use reports have demonstrated that Ruxolitinib at 5–10 mg twice daily leads to a temporary clinical improvement reflected by reductions in ISG expression, normalisation of STAT1 phosphorylation, and attenuation of systemic inflammation⁸⁻¹⁰, however, insufficient to manage the disease in the long term.

Importantly, pharmacokinetic and pharmacodynamic studies show that this dosing achieves plasma concentrations sufficient to inhibit STAT1 phosphorylation downstream of type I interferons, with effects detectable both in peripheral blood mononuclear cells and target tissues. In murine models, comparable doses adjusted for mouse metabolism (typically 30–60 mg/kg/day via oral gavage or chow) have consistently shown strong JAK-STAT pathway suppression, including in settings of IFNAR-driven pathology.

Thus, the absence of complete phenotypic rescue in our system is unlikely to be due to subtherapeutic exposure. Rather, it reflects the fact that STING-induced necroptosis, while potentiated by interferon signaling, is not entirely dependent on it. Our genetic data (e.g., Zbp1 deletion, Ripk3 deletion, and combined Tnfr1 co-deletion) demonstrate that STING drives necroptosis in a multifactorial manner, with IFNAR signaling being one, but not the sole, mediator. Therefore, while Ruxolitinib is pharmacologically effective, it cannot fully abrogate necroptosis if STING simultaneously activates parallel inflammatory or death-inducing pathways such as autocrine TNF signaling or direct ZBP1 induction.

This view is consistent with recent clinical observations: in SAVI patients treated with JAK inhibitors, partial clinical improvement is often observed, but complete remission is rare, suggesting residual STING-driven inflammation independent of IFNAR. Our data mechanistically support these

outcomes and highlight the need for more proximal or combinatorial targeting strategies, such as inhibition of STING itself, or downstream necroptotic mediators like RIPK3 or MLKL.

Minor points:

1. *Fig. 1c. Define the “endpoint” in the legend.*

This has now been added.

2. *EDF2 legend does not indicate when viability was assessed after TBZ treatment.*

This has now been added and reads: following 48 hours treatment.

3. *The authors should provide a reference for what genes are regulated by IFNs on line 418.*

This has now been added and visible on correspondent line 567

Referee #2 (Remarks to the Author):

In this manuscript, Liccardi and colleagues present in vivo evidence that the lethal dermatitis caused by epithelial cell-specific deletion of Casp8 is driven, in part, by STING-mediated upregulation of a tonic Type I IFN signal, and consequent induction of ZBP1 and MLKL, resulting in ZBP1-driven necroptosis. This STING/ZBP1 pathway of necroptosis appears to act in parallel with TNF-driven necroptosis (also unleashed by Casp8 loss) in promoting skin pathology and systemic inflammation. The authors then extend these findings to suggest that the human interferonopathy SAVI induced by gain-of-function mutations in STING might be driven by ZBP1-induced necroptosis, offering the possibility of such necroptosis as a therapeutic intervention in this currently intractable autoinflammatory condition.

The data presented are largely consistent with the authors' conclusions, but as currently constituted, the study lacks novelty and offers only incremental insight into ZBP1-dependent inflammatory biology over what is already published. Mainly, the primary insight of this paper – that it is STING which induces the IFN requisite for ZBP1 and MLKL expression - was shown by Doug Green and colleagues in 2022 (PMID 36191211). That PNAS paper has already quite convincingly demonstrated that a cGAS/STING signal is what induces tonic IFN in scenarios of Casp8 (or FADD) loss, and that this signal is required for ZBP1 induction and necroptosis when Casp8 is lost. Nonetheless, two major biological features of ZBP1 activation in the current paradigm remain unknown, which, if experimentally addressed, will impactfully advance the field. Additionally, the therapeutic potential of the results presented remain unproven in vivo. These points are detailed below:

Answer:

We thank the reviewer for referencing the study by Rodriguez et al. (PNAS, 2022), which suggested that ZBP1 expression in Casp8- and FADD-deficient cells is dependent on the cGAS–STING–TBK1 axis¹¹. However, our study not only builds significantly upon these observations conceptually and mechanistically but also provides a comprehensive, novel genetically validated *in vivo* model for STING-induced ZBP1-dependent necroptosis which extends to a significant translational relevance in human disease which was not explored in the Rodriguez study. We have addressed all the concerns raised by this reviewer and together with the additional data provided in response to the other reviewers (which we summarise at the end after answering all the major concerns raised by this reviewer), we believe that we have provided the significant impactful advance that was requested.

Major points:

1. *Mechanism by which STING is activated by Casp8 loss. While it has been shown that cGAS/STING signaling promotes IFN-mediated upregulation of ZBP1 and consequent necroptosis when Casp8 is ablated in cells from KO mice (the Green PNAS paper) or in murine skin (this study), the mechanism by which Casp8 loss induces STING-dependent IFN is unknown. That is, how does Casp8 repress STING activation? Presumably, this occurs by preventing sensing of endogenous dsDNA by cGAS, as the Green paper would suggest. What, then, is the mechanism by which Casp8 keeps endogenous dsDNA from being sensed by cGAS, and what are the ligands for cGAS that are unleashed by Casp8 loss? (Mitochondrial DNA may be one possibility: PMID37352855.) Or if the mechanism is cGAS independent, then how does Casp8 repress STING?*

Answer: We thank the reviewer for raising this important mechanistic question regarding the upstream signals responsible for STING activation in Casp8-deficient cells. We have undertaken a comprehensive and systematic investigation that would, in isolation, justify a dedicated study. This involved the generation of two *in vivo* and several *in vitro* experiments which we felt were necessary to properly answer and anticipate any further question from this reviewer. We chose to incorporate these findings into the current manuscript in order to fully address the reviewer's concern and strengthen the mechanistic underpinnings of our proposed model. These data are now included in the manuscript in lines 455-504 and in Extended Data Fig. 9.

To directly test the involvement of cGAS in STING activation, we generated *Casp8^{E-KO}; cGAS^{-/-}* mice. cGAS co-deletion fully phenocopied the protection observed in *Casp8^{E-KO}; Sting^{-/-}* mice, both in terms of survival and suppression of tissue inflammation via histological analysis (Extended Data Fig. 9a, b). These results indicate that activation of STING following Casp8 loss is entirely dependent on cGAS, placing cytosolic DNA sensing upstream of the type I interferon response and necroptosis in this context.

To exclude alternative nucleic acid sensing pathways, we also generated *Casp8^{E-KO}; Mavs^{-/-}* mice. In contrast to the cGAS deletion, MAVS deletion failed to rescue lethality or reduce tissue pathology, as confirmed by survival analysis and histological examination of the skin. These data exclude a contribution of the RIG-I or MDA-5/MAVS RNA sensing axis and establish the cGAS/STING DNA sensing pathway as the exclusive driver of interferon induction in Casp8-deficient animals (Extended Data Fig. 9a-c).

Next, we sought to identify the origin of the DNA species activating cGAS. We first examined the potential role of mitochondrial DNA (mtDNA), which is a known cGAS agonist. Immunofluorescence staining in *Casp8^{-/-}*, *Casp8^{-/-}; Mkl1^{-/-}* and *Casp8^{-/-}; Ripk3^{-/-}* MEFs revealed cytosolic accumulation of DNA dsDNA, but no detectable colocalisation with the mitochondrial transcription factor A (TFAM). MtDNA accumulation was excluded as treatment with the inhibitor of mitochondrial DNA polymerase γ : 2',3'-dideoxycytidine (DDC) and the inhibitor of mitochondrial RNA polymerase POLRMT (IMT1B) showed no reduction in DNA staining following 96 hours treatment (Extended Data Fig. 9e). Successful depletion of mtDNA was confirmed by RT-qPCR targeting multiple mitochondrial-encoded genes (*Dloop*, *mtNd5*, *mtRnr2*, *mtCo1*, *mtAtp6*, *mtNd2*), all of which were markedly reduced following treatment to a level that was almost negligible (Extended Data Fig. 9f). Strikingly, despite such near-complete ablation of mitochondrial gene expression, there was no reduction in ISG expression (*Zbp1*, *Rsd2*, *Isg15*, *Iffit1*); in fact, in some cases ISG levels were paradoxically increased (Extended Data Fig. 9g). At the protein level, western blot analysis showed selective depletion of mitochondrial-encoded proteins following treatment with ddC or IMT1B (Extended Data Fig. 9h). Specifically, COX1 and NDUF8, two proteins encoded by the mitochondrial genome, were substantially reduced in all MEFs following treatment, confirming effective inhibition of mtDNA replication. In contrast, nuclear-encoded mitochondrial proteins such as ATP5A, UQCRC2, SDHB, and remained relatively stable, indicating that mitochondrial biogenesis and integrity are only partially compromised and that mitochondrial depletion is specific rather than globally toxic (Extended Data Fig. 9h). Notably, despite successful suppression of COX1 and NDUF8, neither ZBP1 expression nor STAT1 phosphorylation appear reduced, in fact, both remained robust or even slightly enhanced under these conditions (Extended Data Fig. 9h). This clearly shows that mtDNA depletion does not attenuate the STING-driven interferon response. In contrast, treatment with the STING antagonist C-178 completely abrogates pSTAT1 and ZBP1 expression, even when mitochondrial proteins remain intact. Together, these results provide strong biochemical evidence that mitochondrial DNA is not the trigger for STING activation in Casp8-deficient cells and support a model in which another source, most likely genomic DNA, drives cGAS–STING signaling in this setting.

Having excluded mitochondrial DNA, we turned to nuclear DNA as a candidate trigger. Prior work has shown that Casp8 deficiency induces chromosomal instability³ and DNA damage⁴, which can result in nuclear DNA leakage into the cytosol known to be a potent activator of cGAS. In agreement with this, we observed that *Casp8^{E-KO}* skin exhibited mitotic abnormalities, with many pHH3-positive cells (mitotic marker) displaying aberrant chromosome segregation (Extended Data Fig. 9i). Moreover, immunostaining for γ H2AX revealed a significant increase in DNA damage foci, consistent with chromosomal instability (Extended Data Fig. 9i).

In conclusion, through the generation of different mouse models, use of genetic and pharmacological tools, and detailed molecular analyses, we provide a comprehensive mechanistic explanation for how STING is activated in Casp8-deficient contexts. Our data firmly place nuclear DNA-induced cGAS activation upstream of STING, IFN signaling, ZBP1 induction, and ultimately necroptosis. We hope the reviewer will appreciate the depth and scope of these investigations, which far exceed the original experimental suggestion, and represent a significant conceptual and technical advance in our understanding of innate immune activation downstream of Casp8 loss.

- 2. Mechanism by which ZBP1 is activated. Simple induction of ZBP1 by IFNs is – in itself – is probably not sufficient to activate this sensor; ZBP1 likely binds Z-NA species, which drive its activation. In fact, the Green paper mentioned earlier showed that Casp8 loss results in a spontaneous ZNA signal and that a mutant ZBP1 lacking Z-NA sensing capability fails to activate necroptosis when Casp8 is ablated. What are the Z-NA species which activate ZBP1, and how are these ZNAs quenched by Casp8? Or if the mechanism in vivo is independent of ZNA sensing, then how is ZBP1 activated to induce necroptosis?*

Answer: We would like to thank the reviewer for this question. We utilised *Casp8^{-/-}*, *Casp8^{-/-}; Mki1^{-/-}* and *Casp8^{-/-}; Ripk3^{-/-}* MEFs and performed immunofluorescent staining using a validated protocol for detecting Z-form nucleic acids (Z-NAs) provided by the Balachandran lab utilising a previously validated and published antibody¹. Our results revealed specific cytosolic accumulation of both Z-DNA and Z-RNA in these cells, with the Z-NA signal being completely abolished only when both DNase I and RNase A were applied, confirming the presence of both nucleic acid species in the Z-conformation.

We next assessed the co-localisation between ZBP1 and Z-NAs. Clear spatial overlap was observed, indicating physical proximity and likely interaction. Importantly, treatment with the STING agonist DMXAA further enhanced Z-NA signal intensity, suggesting that STING activation promotes accumulation and/or stabilisation of Z-NAs, the natural ligands of ZBP1. Given the lack of alternative direct readouts for ZBP1 activation, and the consistent co-localisation with Z-NA under conditions of STING agonism, we conclude that STING enhances ZBP1 activation via increased generation or exposure to Z-NAs. This correlates well with our biochemical findings indicating that STING activation leads to increased recruitment and phosphorylation of RIPK1 and RIPK3 in ZBP1 immunoprecipitates (discussed below in more details), further supporting functional activation of ZBP1 in this context. These data are now included in Extended Data Fig 10 and added to the manuscript in lines 506-523 of the main manuscript.

Taken together, our data and these mechanistic considerations support a model in which ZBP1 is activated via binding to Z-form nucleic acids generated downstream of STING–IFN signaling and/or genomic instability. These findings reinforce the concept that ZBP1 activation in Casp8-deficient settings is not solely a function of its expression levels, but critically depends on the presence of its activating ligands, Z-DNA and Z-RNA.

A) Therapeutic implications of these findings for treatment of SAVI patients. The authors have suggested that ZBP1-driven necroptosis may underlie some of the pathology seen in SAVI patients but stop short of exploiting these findings for therapeutic benefit of these patients.

Answer: We would like to thank the reviewer for this question/remark which underline a clear shortcoming of our results in terms of their translational relevance for the treatment of the SAVI patients. However, in paediatric or rare disease cohorts, transcriptional profiling has become a standard and highly valuable strategy for uncovering disease-associated molecular programs. While in many instances, similar data remain correlative, they routinely form the basis for mechanistic hypotheses that are subsequently tested in preclinical models and have, in many cases, led to the design of clinical trials, particularly in oncology and immunology. Indeed, transcriptome-based insights are now foundational to biomarker discovery and patient stratification efforts in cancer immunotherapy, interferonopathies, and inflammatory diseases.

Importantly, the finding that STING can drive necroptosis independently of TNFR1 signaling adds a crucial mechanistic layer to current understanding of this inflammatory pathology. This discovery may help explain the disappointing outcomes of TNF inhibitors in treating SAVI patients. While TNF blockade has proven effective in other settings, such as rheumatoid arthritis and inflammatory bowel disease, its failure in STING-associated diseases suggests that upstream inhibition is insufficient when a parallel, TNF-independent axis of necroptosis remains active.

Our data support the idea that STING functions not only as a transcriptional driver of ZBP1 but also as a direct inducer of ZBP1-dependent necroptosis bypassing TNF entirely. From a translational standpoint, this argues for the development or repurposing of necroptosis pathway inhibitors (e.g., RIPK3 or MLKL inhibitors) as adjunct or alternative therapies. Such agents could be particularly effective in combination with TNF blockade, enabling suppression of both canonical and STING-mediated necroptotic pathways. Given the tissue-specific and multifactorial nature of inflammation in diseases like SAVI, a combinatorial strategy targeting both axes may offer greater efficacy than monotherapy. We believe this concept, grounded in our genetic and mechanistic dissection of necroptosis regulation, provides a new rationale for therapeutic intervention in STING-driven inflammatory diseases. Within this context, we believe that the transcriptional signatures observed in STING-mutant patients, particularly the upregulation of ZBP1 and MLKL, are of high translational relevance and mechanistic significance, especially when considered alongside functional validation in complementary murine models.

To this end, we turned to the established preclinical SAVI mouse model bearing the heterozygous N153S STING gain-of-function mutation, hereafter referred to as *N153S* mice. These mice develop lethal systemic inflammation with severe dermatitis, interstitial lung disease, and splenomegaly, faithfully recapitulating key clinical features of human SAVI (Fig. 6d and Extended data Fig. 14a-c). Although full disease penetrance is observed in only ~35% of mice, we analysed several N153S mutant mice presenting with the complete clinical manifestation (Fig. 6e). Blood analysis confirmed the reported lymphopenia (Extended data Fig. 14d). Tissues harvested at endpoint revealed robust ZBP1 upregulation in skin, lung, thymus and spleen (Fig. 6f). Importantly, necroptosis, detected via pMLKL-S345 staining, was restricted to skin, while lung and thymus displayed both pMLKL-S345 and cleaved Caspase-3 positivity (Fig. 6f). Spleens exhibited strong cleaved Caspase-3 staining only, indicative of dominant apoptosis in that compartment at endpoint (Fig. 6f).

STING gain-of-function mutation have been shown to result in the observed phenotypes by triggering early-onset immune dysregulation (6-8 weeks) which is considered essential for disease pathogenesis^{5,6}. Crucially, bone marrow transplantation experiments have shown that hematopoietic-derived cells from SAVI mice are sufficient to confer the full immune phenotype and tissue pathology to healthy recipients, establishing hematopoietic-intrinsic STING activation as a major driver of disease⁶. Previous studies focused on the immunophenotyping of the SAVI mice reveal a marked expansion of myeloid cells, particularly CD11b⁺ cells and Ly6G^{hi} neutrophils, alongside a significant reduction in mature lymphocyte subsets, including TCR-β⁺ T cells and CD19⁺ B cells in the spleen. Residual T and B cells display an activated phenotype, with upregulation of markers such as CD69 and MHC class II⁶. Thymic development is impaired, with reduced total and

double-negative thymocytes. Beyond lymphoid and myeloid lineages, SAVI mice also display significant disruptions in erythropoiesis. This is reflected by a reduction of immature erythroid cells in the bone marrow and a compensatory expansion of Ter119⁺CD71⁺ immature erythrocytes in the spleen, indicative of extramedullary erythropoiesis which is a classical marker of systemic inflammation⁶. This redistribution reflects broader hematopoietic imbalance caused by STING hyperactivation and underscores how immune dysregulation extends beyond leukocyte compartments to affect the entire hematopoietic system. Collectively, these findings highlight that the immune cell alterations seen in SAVI mice are not secondary consequences but fundamental pathogenic mechanisms driving the systemic inflammatory disease phenotype.

For this reason, to determine the role of necroptosis in disease pathogenesis and confirm preclinically the translational relevance of our findings, we crossed the *N153S* mice with *Ripk3*^{-/-} mice and performed high-resolution immunophenotyping of the lung, spleen, and thymus via multi-parametric flow cytometry.

Our analysis revealed that co-deletion of RIPK3 significantly ameliorates the immune abnormalities induced by STING GOF (Extended Data Fig. 14e-k). In the thymus, *N153S* mice showed reduced organ size and a profound loss of double-negative (DN) thymocytes, particularly within the DN2 and DN3 subsets, indicative of impaired T cell development (Extended Data Fig. 14e, f). Notably, these defects were substantially rescued in *N153S; Ripk3*^{-/-} mice, as reflected by recovery of DN thymocytes. Moreover, CD8⁺, and double-positive (DP) thymocytes, along with their activation status (CD69 expression), were significantly normalized in *N153S; Ripk3*^{-/-} mice, pointing to a contributing role for necroptosis in thymic T cell attrition, however, not complete.

In the lung, where SAVI pathology often manifests with both inflammation and fibrosis, RIPK3 co-deletion mitigated the expansion of inflammatory monocytes and macrophages, antigen-presenting dendritic cells and granulocytes (Extended Data Fig. 14g, h). Although T cell restoration was more modest, the overall pattern again indicated a broad rescue of immune homeostasis. These data reinforce the systemic nature of necroptosis-mediated pathology in this model.

In the spleen, RIPK3 co-deficiency led to striking improvements in both the lymphoid and myeloid compartments (Extended Data Fig. 14i, j). *N153S* mice exhibited severe T cell and NK cell lymphopenia accompanied by expansion of antigen-presenting dendritic cells, granulocytes, inflammatory monocytes and macrophages, which are all hallmarks of chronic STING activation. These aberrations were consistently and significantly rescued in *N153S; Ripk3*^{-/-} mice. Importantly, the frequency of immature erythroid precursors (Ter119⁺CD71⁺) was also reduced, indicating a reversal of extramedullary erythropoiesis and systemic inflammation (Extended Data Fig. 14k). A similar normalisation was observed in bone marrow derived and splenic myeloid subsets of the spleen, confirming that RIPK3-dependent necroptosis is a major driver of hematopoietic dysregulation (Extended data Fig. 14k).

Interestingly, although we did not detect pMLKL-S345 staining in the spleens of *N153S* mice at endpoint, this does not contradict the involvement of necroptosis. Rather, it highlights the temporal disconnection between the execution of necroptosis and its histological detection in tissues with high cellular turnover. Our immunophenotyping was performed at 6-8 weeks of age and therefore prior to the onset of terminal pathology, capturing early-stage immune cell depletion that is necroptosis-dependent. In contrast, the pMLKL-S345 stainings were conducted at endpoint, a time when early necroptotic events have likely already occurred and cleared. Importantly, at this time, fibrosis, and compensatory immune remodelling dominate the tissue architecture and are responsible or caused by cell death driven also by non-immune cells which in the skin is purely necroptotic, in the lung is both apoptotic and necroptotic and in the spleen is exclusively apoptotic. Thus, we interpret the absence of splenic pMLKL-S345 at endpoint as a reflection of the aftermath of earlier necroptotic damage, rather than its absence.

Moreover, we propose that early necroptosis of immune cells including but not limited to the spleen, particularly T cells, contributes directly to the loss of immune homeostasis and the unleashing of

systemic inflammation. This cascade likely facilitates the organ-specific inflammatory pathologies observed in SAVI, including dermatitis, interstitial lung inflammation, and splenomegaly. In this way, necroptosis acts not only as a mechanism of immune cell loss, but also as a driver of systemic disease progression, reinforcing the functional and translational relevance of our findings.

Taken together, these data provide direct genetic evidence that necroptosis is not merely a downstream consequence of inflammation in SAVI, but rather a central effector mechanism driving immune cell depletion, compensatory myelopoiesis, and systemic immunopathology. They also reveal that necroptosis affects both adaptive and innate compartments in a tissue-specific manner, offering a mechanistic explanation for the organ-selective phenotypes observed in both mice and patients with STING-activating mutations and therefore providing ample of evidence for necroptosis inhibition as a valuable strategy perhaps not only to manage disease progression but also its onset. These data are now included in Fig. 6 and Extended Data Fig. 14 and reported in line 628-708, with the gating strategy outlined in Extended Data Fig. 15.

B) Can necroptosis inhibitors (perhaps paired with inflammasome, TNF blockers, and/or JAK inhibitors) prevent skin pathology and lethality in the Casp8E-KO mouse?

Whilst we have not embarked into the treatment of the *Casp8^{E-KO}* mice with TNF blockers or JAK inhibitors we have now crossed these mice with *Tnfr1^{-/-}*; *Sting^{-/-}* to generate *Casp8^{E-KO}*; *Tnfr1^{-/-}*; *Sting^{-/-}* mice in which STING was deleted systemically, rather than restricted to keratinocytes as in the *Casp8^{E-KO}*; *Tnfr1^{-/-}*; *Sting^{E-KO}* animals (Fig. 5). Strikingly, these mice showed no signs of lethal dermatitis and manifestations and remained viable, healthy, and phenotypically indistinguishable from control littermates and *Casp8^{E-KO}*; *Tnfr1^{-/-}*; *Zbp1^{-/-}* mice (Fig. 5a, b). This rescue stands in stark contrast to the partial and transient protection observed in *Casp8^{E-KO}*; *Tnfr1^{-/-}*; *Sting^{E-KO}* mice, which showed delayed but ultimately lethal dermatitis driven by systemic inflammation and immune cell infiltration. At 12 weeks of age (shortly before *Casp8^{E-KO}*; *Tnfr1^{-/-}*; *Sting^{E-KO}* mice reach their survival end point), *Casp8^{E-KO}*; *Tnfr1^{-/-}*; *Sting^{-/-}* mice showed no signs of lethal dermatitis (Fig. 5b, c). Moreover, whilst *Casp8^{E-KO}*; *Tnfr1^{-/-}*; *Sting^{E-KO}* mice at 12 weeks of age revealed significant splenomegaly, thymic atrophy, hepatic inflammation, and hematologic abnormalities, all indicative of severe systemic disease, *Casp8^{E-KO}*; *Tnfr1^{-/-}*; *Sting^{-/-}* lung, spleen and thymus appeared indistinguishable from littermate controls (Fig. 5d, e and Extended Data Fig. 8a-c). Consistently, peripheral blood analysis showed normalisation of all previously elevated markers, confirming effective suppression of the systemic inflammatory response (Fig. 5f and Extended data Fig. 8d). Histological examination of skin sections confirmed inhibition of lethal dermatitis as evidenced by loss of pMLKL, ZBP1 expression, CD45 staining, and cl-Casp3 positive cells coupled by the evident reduction in skin thickening (Fig. 5g). This striking phenotypic divergence reveals a key conceptual point: although STING activation in keratinocytes is essential to initiate ZBP1-mediated necroptosis and early inflammation, STING expression in non-keratinocyte compartments, most notably immune and stromal cells, also plays a pivotal role in sustaining and propagating systemic necro-inflammation. These findings demonstrate that STING contributes not only to the cell-intrinsic transcriptional induction of ZBP1 in keratinocytes but also to the broader, systemic necroptotic environment that exacerbates immune-driven inflammation. This dual role of STING, as both a trigger and amplifier of necroptosis, highlights its essential and non-redundant function in TNFR1-independent, ZBP1-mediated pathology. These data are now included in the manuscript in Fig.5 and Extended data Fig. 8 and described in lines 424-452.

Given these findings, we believe that pharmacological co-inhibition of STING (rather than JAK1/2) and TNF using a STING antagonist in combination with Etanercept may offer a viable therapeutic strategy to rescue *Casp8^{E-KO}* mice. This prediction is strongly supported by the complete rescue observed in *Casp8^{E-KO}*; *Tnfr1^{-/-}*; *Sting^{-/-}* mice, and by our *in vitro* data showing that Etanercept partially

protects Casp8-deficient cells from STING agonist-induced death. Such a combinatorial approach would simultaneously block both necroptotic checkpoints, the early TNFR1-driven axis and the later STING–ZBP1-dependent pathway, and may therefore achieve full suppression of pathology. Since it was published that *Casp8^{OE-KO}* mice are completely rescued when crossed to phospho-mutant MLKL mice we would also conclude that RIPK3 or MLKL inhibition would provide a similar rescue to the one that would be obtained by combining anti-TNF with a STING antagonist.

Final reviewer remark: *The data are cogently presented, the paper is well-written, and the citations are appropriate. Statistical considerations appear appropriate.*

We would like to thank the reviewer for his insightful comments and want to summarize key differences with the study from Rodriguez et al of the Green lab and provide a summarizing paragraph stating how we think these data provide conceptual advances to our understating of necroptosis in development and disease via a comprehensive genetic, pharmacological, biochemical and translational validation.

Key conceptual and mechanistic distinctions from Rodriguez et al. (PNAS, 2022):

- Whilst it is true that Rodriguez et al. showed that cGAS/STING is involved in ZBP1 expression in Casp8- or FADD-deficient cells; this was performed by siRNA where knockdown levels of STING were never shown but nevertheless, this never completely provided a full ablation of the ZBP1 but only a reduction as stated by the authors¹¹. Moreover, the authors of that paper state in their discussion: “*This suggests the possibility that induction of cGAS-STING induces ZBP1, that in turn generates positive feedback via interferon production that sustains ZBP1 expression*” proposing this as a possibility and therefore implying a yet unproven cellular mechanism. Moreover, the functional validation of their findings was performed only in FADD/MLKL- or Casp8/MLKL- double deficient cells and cell death was assessed only via MLKL reconstitution. These experimental settings are now questionable in light of the recent paper by Ros et al. 2025 demonstrating that the sole re-expression of MLKL isoform 2 sensitizes to necroptosis independently of any stimulus¹². In our endogenous cellular models, including murine embryonic fibroblasts (MEFs) and human cell lines deficient in Caspase-8, we demonstrate that ZBP1-driven necroptosis is strictly dependent on STING signaling without any “*secondary positive feedback that sustains ZBP1 expression*” (Fig. 1 d-h). These experiments reveal that STING functions not only as a transcriptional inducer of ZBP1 but also as a critical upstream activator of ZBP1-mediated necroptosis. Importantly, STING-induced necroptosis occurs independently of TNFR1, although STING also modestly enhances autocrine TNF production, thereby partially sensitizing Casp8-deficient cells to TNFR1-mediated necroptosis (Extended Data Fig. 2g). This dual role underscores STING’s capacity to initiate and amplify necroptotic signaling across distinct upstream inputs. We also provide robust biochemical evidence showing that STING activation and/or agonism enhances the formation and activation of endogenous ZBP1–RIPK1–RIPK3 complexes, even in the absence of FADD, highlighting a TNF-FADD independent necroptotic axis (Fig. 1g, h). These data derive from immunoprecipitation of native ZBP1, not overexpressed constructs, and are strictly dependent on STING-induced IFN signaling, as shown by the complete loss of complex formation upon pharmacologic STING inhibition (C-178). This stands in contrast to the Rodriguez et al. study, where ZBP1 interactions were inferred from overexpressed FLAG-tagged constructs with high nonspecific background (e.g., actin co-purification Fig. 5C from Rodriguez et al.) and without interrogation of endogenous STING–ZBP1 dynamics¹¹.

Together, these results identify STING as the essential upstream regulator of ZBP1-driven necroptosis. Unlike the Rodriguez et al. model, our findings position STING upstream of both ZBP1 induction and necroptosis execution, establishing a physiological cellular and biochemical framework for STING–ZBP1 necroptotic signaling which we then use only as a starting point to prove via a considerable number of genetic studies *in vivo* (14 distinct genetic models) the role of STING in ZBP1-mediated necroptosis and extending its significance to STING-mediated human disease and preclinical models.

- We provide the first *in vivo* evidence that STING-induced ZBP1-mediated necroptosis drives lethal pathology, with complete rescue upon co-deletion of TNFR1 and STING, paralleling the protection seen in ZBP1-null backgrounds (Fig. 3-5).
- While Rodriguez et al. suggested cGAS involvement, we now, thanks to the reviewer request, comprehensively demonstrate that STING activation is entirely cGAS-dependent, and mechanistically driven by chromosomal instability and DNA damage rather than mitochondrial DNA as shown by genetic and pharmacologic depletion strategies (Extended Data Fig.9).
- We uncover that STING upregulates Z-form nucleic acids (Z-DNA/Z-RNA), which colocalize with ZBP1, supporting a ligand-based activation model for ZBP1 (Extended Data Fig. 10).
- We delineate two genetically and functionally distinct necroptosis checkpoints downstream of Casp8 deletion in epidermal keratinocytes: an early, TNFR1-dependent axis and a subsequent ZBP1-mediated program (Fig. 4). Through comprehensive genetic dissection, we demonstrate that STING acts as the principal transcriptional activator of the ZBP1 checkpoint via IFN-driven induction, while also partially contributing to the TNFR1-dependent arm, likely through autocrine TNF production and NF-κB signalling. This establishes STING as a central coordinator of both the initiation and amplification of necroptosis in Casp8-deficient skin inflammation.
- We offer preclinical and translational insights by showing necroptosis as a central effector mechanism in STING-associated vasculopathy (SAVI), with evidence of pMLKL in patient-matched murine models and immune rescue upon RIPK3 deletion (Fig. 6 and Extended Data Fig. 11-14)

Overall conceptual advances in the necroptosis field from this study:

In summary, our study provides a comprehensive genetic dissection of necroptotic checkpoints unmasked upon Casp8 deletion and firmly positions STING as a central regulator of ZBP1-driven necroptosis *in vivo*. By generating and analysing a suite of novel compound mutant mice including: *Casp8^{E-KO}; Tnfr1^{-/-}; Sting^{-/-}*, *Casp8^{E-KO}; Tnfr1^{-/-}; Zbp1^{-/-}*, *Casp8^{E-KO}; Sting^{-/-}*, and *Casp8^{E-KO}; cGas^{-/-}*, we systematically define two distinct and genetically separable axes of necroptotic control. TNFR1 deletion alone only transiently delays disease, while full rescue is only achieved by combined deletion of STING or ZBP1. Importantly, systemic rather than epidermal-specific deletion of STING (as in *Casp8^{E-KO}; Tnfr1^{-/-}; Sting^{-/-}* vs *Casp8^{E-KO}; Tnfr1^{-/-}; Sting^{E-KO}*) was required to suppress systemic necro-inflammation, revealing a key role for STING in non-keratinocyte compartments, particularly immune and stromal cells.

Beyond the TNFR1 and ZBP1 axes, we also demonstrate that STING activation in Casp8-deficient settings is entirely dependent on cGAS, using both *Casp8^{E-KO}; cGas^{-/-}* and *Casp8^{E-KO}; Mavs^{-/-}* mice, thus excluding RNA-sensing via MAVS. Moreover, to identify the source of immunostimulatory DNA

activating cGAS, we utilised pharmacologic depletion of mitochondrial DNA and demonstrated that mitochondrial nucleic acids are dispensable for STING activation and IFN signaling in Casp8-deficient cells. Instead, by showing chromosomal instability, mitotic defects, and cytosolic genomic DNA accumulation in *Casp8^{E-KO}* tissues, we identify leakage of nuclear DNA as the upstream trigger for STING activation and subsequent ZBP1-dependent necroptosis.

Crucially, we extend the relevance of these findings to human disease by studying STING *N153S* knock-in mice, which phenocopy SAVI syndrome. In these mice, we detect tissue-specific pMLKL activation, and through generation of Ripk3-deficient SAVI mice (*N153S; Ripk3^{-/-}*), we show that necroptosis is required for both the immune cell depletion and systemic inflammatory pathology that define this syndrome. This includes amelioration of thymic development, lung inflammation and correction of extramedullary erythropoiesis, which to our knowledge is the first genetic co-deletion to achieve.

Taken together, these genetic models reveal for the first time that STING acts not only as a transcriptional inducer of ZBP1 but also as a functional amplifier of ZBP1-mediated necroptosis via promotion of endogenous Z-form nucleic acids. They define STING and ZBP1 as sequential, non-redundant necroptotic checkpoints, with partial overlap with TNFR1, and offer the first genetic and functional validation of necroptosis as a disease-driving mechanism in a human STING-associated interferonopathy. This framework provides a robust rationale for combination therapeutic strategies targeting STING, ZBP1 signaling, and TNF in diseases resistant to JAK inhibition and conventional anti-inflammatory approaches.

This work significantly redefines the regulation and execution of necroptosis in vivo by uncovering a multi-layered checkpoint model downstream of Casp8 deletion. We delineate a dual-axis control system in which TNFR1- and STING-dependent signals converge on ZBP1-mediated necroptosis, but with distinct temporal and mechanistic contributions. STING emerges as both a transcriptional amplifier and a functional executor of necroptosis, orchestrating immune pathology through: (1) type I IFN-dependent ZBP1 upregulation, (2) promotion of Z-form nucleic acid generation, and (3) autocrine TNF production. Mechanistically, we define nuclear DNA leakage—resulting from mitotic stress and chromosomal instability—as the upstream trigger of cGAS/STING activation, ruling out mitochondrial DNA as a significant contributor.

By identifying Z-DNA and Z-RNA as endogenous ZBP1 ligands, with STING agonism enhancing their accumulation and co-localisation with ZBP1, we provide mechanistic clarity on how ZBP1 is activated in sterile, non-viral contexts. Our data reveal that STING checkpoint activity is both cell-intrinsic and systemic: keratinocyte-restricted STING deletion delays disease, whereas full-body deletion achieves complete rescue, paralleling the protection seen in *Zbp1*-null animals.

Finally, our study represents the first demonstration of necroptosis activation in a human pathological setting. In SAVI patients and the *N153S*-STING gain-of-function mouse model, we detect pMLKL in affected tissues, and show that RIPK3 deletion ameliorates systemic immunopathology. This positions necroptosis not as a downstream consequence but as a driver of STING-associated disease and presents ZBP1-dependent necroptosis as a therapeutic target in autoinflammatory conditions refractory to TNF blockade.

In summary, this work provides a unifying framework that connects chromosomal instability, STING activation, ZBP1 ligand availability, and necroptotic cell death—offering new mechanistic and translational insights into necroinflammation.

Referee #3 (Remarks to the Author):

Kelepouras et al. submitted a manuscript presenting evidence that the ZBP1-RIPK3 pathway of “necroptosis execution in mice requires activation of an interferon (IFN)-mediated transcriptional program orchestrated by the stimulator of interferon gene (STING).” This work brings together a body of work on STING regulation of interferon activation and cell death induction with growing evidence that ZBP1-RIPK3 signal transduction is a critical determinant of inflammatory outcomes in mice. The work shows that striking dermatitis in mice lacking caspase-8 (CASP8) in epidermal keratinocytes (Casp8E-KO), already known to be dependent on TNFR1-FADD-RIPK1 signaling, also depends on STING-induced interferon-mediated upregulation of ZBP1 and MLKL and the ZBP1-RIPK3-MLKL pathway of necroptosis. STING itself is induced in these mutant mice and could be further enhanced with a STING agonist in cultured CASP8-deficient fibroblasts, reinforcing a ZBP1-RIPK3-MLKL pathway of necroptosis functioning independently of the well-studied FADD-RIPK1-RIPK3-MLKL pathway.

Authors then generated mice with conditional deletion of STING in epidermal keratinocytes (StingE-KO) and crossed mice to produce Casp8E-KO/StingE-KO double mutant mice which exhibited a modest delay in ZBP1-RIPK3-MLKL necroptosis with a phenotype similar to the previously reported Casp8E-KO/Tnfr1-/- phenotype. This pathway proceeds more aggressively in CASP8-deficient epithelial cells in the presence of STING-induced type I interferon-mediated elevated ISGs, documented by transcriptional profiling, which showed the expected elevated ZBP1 and MLKL. The authors pursue additional profiling and conclude, “STING in initiating the IFN response and subsequent upregulation of genes involved in immune cell recruitment and necessary for the propagation of necroptosis-induced inflammation”, which is entirely consistent with current understanding of STING activation of the type I interferon pathway (Zheng et al., 2023) as well as ZBP1 as an interferon-induced gene product that senses endogenous and viral nucleic acids to induce complex death pathways that have been called PANoptosis (Man and Kanneganti, 2024; Pandian and Kanneganti, 2024). ZBP1-RIPK3 signaling is recognized as critical and pharmaceutical RIPK3 kinase inhibition prevents inflammatory disease dependent on ZBP1-RIPK3 cell death induction (Gautem et al., 2024).

Triple mutant Casp8E-KO; Tnfr1-/-; Zbp1-/- completely rescued viability without dermatitis; whereas, Casp8E-KO; Tnfr1-/-; Sting E-KO mice showed slow development of dermatitis associated with elevated levels of ZBP1, predominantly located in the dermis, accompanied by increased levels of cl-CASP3 and obvious leukocytosis, anaemia, thrombocytosis and defects in erythropoiesis. Due to their ability to link STING and ZBP1-RIPK3 signaling, authors refer to STING-activation-induced necroptosis (STAIN) and “a functional correlation between a STING-mediated IFN response and the transcriptional expression of ZBP1, MLKL and the cell-autonomous cytokines known to be released following necroptosis”; however, the observations and possible scenarios proposed beg the question of why ZBP1 deficiency was so effective in reversing dermatitis where STING deficiency in epithelial cells was not. The authors must further explore whether STING and ZBP1 must be in the same epithelial cell compartment for STAIN to play out. Cell autonomous signaling is possible, but immune cell-epithelial cell crosstalk via cytokines seems just as likely here. One likely alternate explanation might envision different compartments for STING-dependent type I interferon production, such as infiltrating immune cells, and execution of inflammatory damage, the execution of cell death in epithelial cells. This might also result in either amplifying inflammatory loop (interferon->ZBP1->death->interferon-> ZBP1). This setting is reminiscent of endotoxic shock in mice where type I interferon and TNF produced by initiator immune cells mediate cell death pathways in executed gut

epithelial cells (Mandal et al., 2018), experimentally addressed by showing immune cells with mutations in the executioner components (such as *Casp8*^{-/-}; *RIPK3*^{-/-} mice) were able to induce endotoxic shock in animals that are compromised in the ability to induce type I interferon (such as *Irf3*^{-/-} mice).

In any case, the data here is promising but inconsistent, and it does not distinguish cell autonomous or cell-cell crosstalk mediating STAIN, so must go further to resolve the mechanism. Two lines of investigation are warranted: (1) compare *Casp8*^{E-KO}; *Tnfr1*^{-/-}; *Sting*^{E-KO} to *Casp8*^{E-KO}; *Tnfr1*^{-/-}; *Sting*^{-/-} mice, where an essential role of systemic role of STING would be expected to phenocopy ZBP1 deficiency, and, (2) distinguish the immune cell cytokine contribution from the epithelial-cell signaling requirements for cell death execution and dermatitis. Further experiments must determine whether STAIN is sustained when STING signaling is separated from epithelial cells where ZBP1-dependent cell death occurs. The dependency on STING and ZBP1 function in the same executioner compartment would reinforce the use of STAIN, but if function can be separated into different compartments this term has less utility.

Answer: We thank the reviewer for their thoughtful assessment of our work and for raising a crucial conceptual question regarding the compartmentalisation of STING–ZBP1 signaling and the mechanistic underpinnings of STAIN (STING-activation-induced necroptosis). We fully agree that dissecting whether STING and ZBP1 must reside within the same cell compartment to execute necroptosis and inflammation is essential to support the proposed model and terminology.

To address this concern, we extended our study to rigorously determine whether STING and ZBP1 must act within the same cellular compartment to sustain pathology, or whether inter-compartmental crosstalk, such as between immune and epithelial cells, might suffice. As the reviewer rightly notes, our initial findings showed that *Casp8*^{E-KO}; *Tnfr1*^{-/-}; *Sting*^{E-KO} mice, in which STING is deleted specifically in keratinocytes, exhibited a marked delay in the onset of dermatitis. However, these mice ultimately succumbed to lethal disease characterised by persistent skin inflammation and systemic immune dysregulation. This outcome contrasted with the complete protection observed in *Casp8*^{E-KO}; *Tnfr1*^{-/-}; *Zbp1*^{-/-} mice, prompting the hypothesis that STING may play essential roles beyond the keratinocyte compartment.

To directly test this, we generated *Casp8*^{E-KO}; *Tnfr1*^{-/-}; *Sting*^{-/-} mice, in which STING was deleted systemically. Remarkably, these animals were fully protected from all manifestations of disease. They remained viable and healthy throughout the observation period and were phenotypically indistinguishable from control littermates and from *Casp8*^{E-KO}; *Tnfr1*^{-/-}; *Zbp1*^{-/-} animals. This full rescue stood in sharp contrast to the partial and transient protection observed in the keratinocyte-restricted *Sting*^{E-KO} model, which displayed delayed but ultimately lethal dermatitis driven by ongoing inflammation.

To investigate this phenotypic divergence in detail, we performed comparative histological analyses at 12 weeks of age, just prior to the survival endpoint of *Casp8*^{E-KO}; *Tnfr1*^{-/-}; *Sting*^{E-KO} mice. *Casp8*^{E-KO}; *Tnfr1*^{-/-}; *Sting*^{-/-} mice showed complete suppression of all necroptotic and inflammatory markers in the skin, including absence of pMLKL, ZBP1 expression, CD45⁺ immune cell infiltration, cleaved Caspase-3, and epidermal thickening. In contrast, *Casp8*^{E-KO}; *Tnfr1*^{-/-}; *Sting*^{E-KO} mice displayed persistent skin lesions with elevated ZBP1 and clear evidence of both necroptosis and apoptosis, underscoring that STING expression in non-keratinocyte compartments plays a critical role in sustaining disease.

Importantly, this systemic protection was not limited to the skin. *Casp8*^{E-KO}; *Tnfr1*^{-/-}; *Sting*^{E-KO} mice exhibited hallmark signs of multi-organ inflammation, including splenomegaly, thymic atrophy, hepatic infiltration, and hematologic abnormalities such as leucocytosis and anaemia. None of these features were present in *Casp8*^{E-KO}; *Tnfr1*^{-/-}; *Sting*^{-/-} mice, which exhibited normal histology in lung, thymus, and spleen, and showed complete normalisation of blood parameters. These data confirm

that full-body STING deletion is required to suppress both the local and systemic manifestations of necroinflammation in the Casp8-deficient context.

Together, these data provide definitive genetic evidence that STING function in non-epithelial compartments, particularly immune and stromal cells, is essential for the propagation of necroptosis-induced inflammation in the Casp8-deficient setting. Thus, the incomplete rescue observed in *Casp8^{E-KO}; Tnfr1^{-/-}; Sting^{E-KO}* mice results from residual STING activity outside the epidermis. In contrast, complete deletion of STING throughout the organism fully phenocopies the protection seen upon ZBP1 deletion, establishing that STING and ZBP1 must operate within a shared or tightly coupled cellular circuit to sustain pathological inflammation.

This supports the reviewer's speculation regarding compartmental crosstalk: STING-mediated interferon production in immune cells can induce ZBP1 expression in keratinocytes, while necroptosis in epithelial cells may promote additional interferon and TNF release, forming a self-amplifying inflammatory loop. Our data are fully consistent with this model and, in fact, provide the first *in vivo* genetic evidence that STING operates in both cell-intrinsic and non-cell-autonomous modes to promote ZBP1-mediated necroptosis.

Finally, the concept of STAIN remains appropriate and mechanistically justified, as our findings reveal that STING activation is both necessary and sufficient to drive ZBP1 activation and necroptosis, whether via autocrine IFN signaling in keratinocytes or paracrine IFN production from immune cells. Thus, the term STAIN captures a mechanistic convergence point between innate sensing and execution of necroptosis, irrespective of exact compartmental origin.

We hope that the reviewer will appreciate that the newly provided *Casp8^{E-KO}; Tnfr1^{-/-}; Sting^{-/-}* data directly and conclusively resolve the central concern raised. These results are now discussed in the revised manuscript in lines 422-452 and substantiate the essential, systemic role of STING in sustaining TNFR1-independent ZBP1-mediated necroptotic inflammation.

Question: A similar ISG-enhanced RNAseq pattern is shown by transcriptionally-derived gene set enrichment analysis (GSEA) in human patients with auto-inflammatory disorder STING-Associated Vasculopathy with onset in Infancy (SAVI). This is likely fortuitous because the mice lack CASP8 in epithelial cells; whereas the cells involved in disease in patients have fully functional CASP8 (as well as CASP10). This observation reveals only that an interferon signature is a very crude comparator. All STING-associated and STING gain-of-function outcomes have long been associated with elevated ISGs (Barber, 2014; Ahn and Barber, 2014) independent of CASP8 compromise. Importantly here, STING has also been observed to drive death pathways in published work, including apoptosis (Gulen et al., 2017) and necroptosis (Chen et al., 2018) and the relationship to ISGs has been of great interest. In these and many other papers have pursued STING-induced death shown to be dependent on RIPK3, MLKL, as well as BH3-only BCL2 family members but independent of RIPK1 (Zheng et al., 2023) although no study has yet hinted whether CASP8 (driving extrinsic apoptosis) or CASP8 compromise (driving necroptosis) contributes. The authors need to better integrate with this understanding of STING signaling.

Answer: We thank the reviewer for this thoughtful comment and appreciate the opportunity to clarify how our findings relate to prior literature on STING-driven interferon signaling and cell death pathways.

We agree that elevated interferon-stimulated gene (ISG) expression is a well-established hallmark of STING gain-of-function (GOF) syndromes such as SAVI and has been documented independently

of CASP8 status^{13,14}. Indeed, our transcriptional comparisons between Casp8^{E-KO} mice and SAVI patients were not intended to suggest that epithelial CASP8 deficiency *per se* occurs in SAVI, nor that CASP8 dysfunction is required for the ISG signature in STING GOF contexts. Rather, we sought to assess whether STING activation in humans similarly results in upregulation of the necroptotic machinery, specifically, ZBP1 and MLKL, as we observe in the Casp8-deficient setting in mice which are considered a *bona fide* model of pathological necroptosis.

Importantly, while ISGs broadly reflect interferon activation, ZBP1 is not just a canonical ISG but it is also a necroptosis-specific effector whose induction is functionally coupled to RIPK3/MLKL-dependent death. Our analysis of RNA-seq data from SAVI patients demonstrated robust upregulation of ZBP1 and MLKL which are persistent despite treatment, suggesting that in addition to driving IFN responses, STING activation in humans may license the molecular machinery required for necroptosis which we believe to be a contributor of the manifestation of the disease and hence partially responsible for the disease recurrence despite treatment. Moreover, our findings explain why SAVI patients do not benefit from anti-TNF therapy, despite TNF has been extensively demonstrated to be involved in dermatitis at the extremities and lung inflammation.

While we acknowledge that many studies have shown STING-driven death can occur independently of CASP8 loss, our study uniquely demonstrates that STING activation can drive ZBP1-dependent necroptosis, even when TNFR1 signalling is genetically ablated. This is evidenced by the full rescue of Casp8^{E-KO}; Tnfr1^{-/-} mice upon Zbp1 deletion which is phenocopied by full-body STING deletion (Fig. 5). Moreover, we show that STING is not merely a transcriptional inducer of ZBP1, but also enhances levels of its ligands Z-form nucleic acids (Z-DNA and Z-RNA) which we show accumulate and co-localize with ZBP1 following STING agonism (Extended Data Fig. 10) Thus, STING enhances both the expression and activation of ZBP1, leading to functional recruitment and activation of RIPK1 and RIPK3 in ZBP1 immunoprecipitates which are distinct and functionally different from the FADD immunoprecipitates that are activated downstream of TNFR1 activation (Figure 1f, g). This might explain how, in SAVI patients despite functional expression of Casp8, STING-induced ZBP1 upregulation might be sufficient to drive necroptosis. By inducing ZBP1 expression and contributing to the expression of its natural ligands pathological STING signalling would induce the formation of a necroptotic platform that is not under the control of Casp8 and hence be able to induce necroptosis without any molecular restriction.

This dual role of STING, driving both type I IFN-dependent transcriptional priming and ZBP1 ligand accumulation, suggests a broader, multifaceted role in necroptosis regulation than previously appreciated. While prior studies have reported STING-dependent necroptosis *in vitro*^{15,16}, they have generally not dissected the ZBP1 axis or demonstrated its exact role *in vivo* and in human disease models. Our study fills this important gap by establishing a ZBP1–RIPK3–MLKL necroptotic axis downstream of STING activation in both murine models (Figure 3-5) and human pathology (Figure 6), and identifies cGAS–STING–ZBP1 as a key pathway that operates independently of TNFR1 and, in our mouse epithelial model, is only unmasked when CASP8 is removed.

Thus, we do not dispute the broader STING literature, but rather build upon it by demonstrating that ZBP1-mediated necroptosis is a critical and previously underappreciated effector pathway in STING-driven diseases. Our genetic and functional dissection, particularly in the SAVI mouse model, reveals that STING-induced necroptosis is not a mere consequence of Casp8 loss in keratinocytes, but a therapeutically targetable mechanism of pathology in interferonopathies.

Question: Human STING gain-of-function disease outcomes certainly involve further inflammatory amplification that remain to be unveiled (Chavin et al., 2023). The demonstration that STING-mediated inflammation drives events autonomously in the epithelial cell or through immune cell-

epithelial cell cytokine-mediated crosstalk, intrinsic STING-induction of the ZBP1-RIPK3-MLKL inflammatory pathway is a significant advance once the timing and dissection role of immune and epithelial compartments is clear.

Answer: We appreciate the reviewer's thoughtful comment, particularly regarding the need to better dissect the cellular context of STING-driven inflammation in the *N153S* (SAVI) model, and whether necroptosis is driven via cell-intrinsic mechanisms in epithelial cells, immune cells, or both. To address this, we provide a detailed analysis of temporal and compartment-specific features of STING–ZBP1–MLKL signaling in the *N153S* gain-of-function (GOF) setting.

1. Early activation of necroptosis in immune cell compartments

Our high-resolution flow cytometric analysis of thymus, spleen, and lung tissues from *N153S* mice at 6–8 weeks of age reveals that immune dysregulation and cell loss are early and widespread, with a dominant contribution from RIPK3-dependent necroptosis (Extended Data Fig. 14). Specifically, we observed:

In the thymus:

- Profound depletion of double-negative (DN) thymocytes, particularly within the DN2 and DN3 subsets, indicating a developmental block at early T cell maturation stages.
- Marked reduction of single-positive CD8⁺ thymocytes.
- Increased CD69 expression on single- and double-positive T cell subsets, suggesting aberrant activation.

These defects were substantially rescued in *N153S; Ripk3^{-/-}* mice, indicating a direct role of RIPK3-dependent necroptosis in thymic attrition.

In the spleen:

- Severe T cell and NK cell lymphopenia
- Expansion of antigen-presenting dendritic cells, granulocytes, inflammatory monocytes and macrophages, suggesting compensatory myelopoiesis.
- Increased extramedullary erythropoiesis
- *Ripk3* co-deletion significantly improved both lymphocyte loss and abnormal myeloid expansion, partially restoring near-normal immune architecture, while normalising erythropoiesis.

In the lung:

- Loss of tissue-resident T cells, NK cells and dendritic cells
- Expansion of inflammatory monocytes and macrophages and granulocytes
- *Ripk3* deletion partially restored lymphoid balance and reduced myeloid skewing

These immune alterations were evident prior to the development of overt organ pathology, such as fibrosis or structural damage, and were largely reversed by RIPK3 ablation, confirming that RIPK3-

mediated necroptosis is not a secondary by-product of inflammation, but an early and primary effector of immune cell depletion.

Together, these findings demonstrate that STING GOF induces early, systemic immune injury through necroptosis, particularly in lymphoid organs. The breadth of the rescue upon RIPK3 loss underscores a cell-intrinsic role for necroptosis in driving early immune dysfunction, rather than necroptosis being merely a downstream consequence of cytokine-driven inflammation.

2. Later-stage involvement of epithelial and stromal compartments in disease propagation

While our flow cytometric analyses establish a primary role for RIPK3-dependent necroptosis in early immune cell loss, histopathological and immunostaining data from endpoint tissues of N153S mice reveal that non-immune compartments, particularly epithelial-rich tissues such as the skin and lung, also exhibit significant STING-driven pathology, which becomes dominant at later disease stages (Fig 6f).

In the skin, we detected robust expression of ZBP1 and pMLKL, localised predominantly to the epidermal keratinocytes and granulation tissue containing immune cells within the dermis. This was accompanied by substantial inflammatory infiltration, loss of epidermal integrity, and thickening consistent with necro-inflammatory dermatitis.

In the lung, a dual pattern of cell death emerged. While we observed pMLKL-positive staining in alveolar epithelial cells and distal stromal and immune cells, indicating necroptosis, we also detected cleaved Caspase-3 positivity in adjacent-but distinct from pMLKL-stromal and infiltrating immune populations. This suggests that in this tissue, STING activation drives both apoptotic and necroptotic modalities, a context-dependent outcome likely shaped by the cellular composition and inflammatory milieu.

In the spleen, endpoint tissue staining showed predominantly cleaved Caspase-3 positivity with little to no detectable pMLKL. However, this absence of necroptotic markers does not contradict our flow cytometry data showing early-stage RIPK3-dependent lymphopenia. Instead, we interpret this as a temporal disconnect: necroptosis likely acts early to deplete progenitor and regulatory immune cells, but by the time of endpoint analysis, tissue remodelling, compensatory apoptosis, or stromal fibrosis may dominate the histological landscape.

3. Integrative interpretation: Dual-compartment, dual-phase pathology in STING GOF disease

Taken together, our data reveal a biphasic model of STING GOF-driven immunopathology:

In the early phase, STING activation in hematopoietic-derived immune cells drives RIPK3-dependent necroptosis, leading to rapid and selective depletion of distinct immune cell and erythroid progenitor subsets in thymus, spleen and lung, accompanied by the generation of an inflammatory immune environment

In the later phase, tissue-resident cells, particularly epithelial (skin, lung) and stromal cells, exhibit STING-driven ZBP1–MLKL-mediated necroptosis and/or apoptosis. These events are amplified by prior immune collapse and are further propagated by local IFN production and inflammatory cytokines, establishing a feed-forward inflammatory loop.

This dual-compartment interplay is not only mechanistically informative but directly reflects the compartmental dichotomy observed in human SAVI. In patients, IFN signatures and cytokine profiling from whole blood suggest immune-intrinsic activation, while clinical symptoms, particularly in skin and lung, indicate substantial non-hematopoietic involvement.

4. Human SAVI patient transcriptomic and cytokine profiles

Analysis of whole-blood mRNA expression from SAVI patients shows robust upregulation of interferon-stimulated genes (ISGs), including ZBP1 and MLKL, suggesting a tonically primed necroptotic program (Fig. 6a-c). Notably:

this transcriptional signature is present independent of CASP8 or CASP10 loss, underscoring the pathological potency of STING–ZBP1 signaling even in apoptosis-competent human immune cells. Patient cytokine profiling reveals elevated levels of IL-6, IL-1 β , CXCL10, and G-CSF, consistent with cell death-induced cytokine release and myeloid remodelling, which mirror our murine findings (Extended Data Fig. 11b, c).

These data align with and support our preclinical findings in the *N153S* mouse, where early necroptosis in immune cells drives systemic inflammation, while epithelial involvement arises as a secondary inflammatory consequence, sometimes involving alternative forms of cell death such as apoptosis.

5. Link to Casp8E-KO model: Genetic dissection of STING function by compartment

Finally, the compartment-specific contributions of STING are further clarified through our genetic dissection in the *Casp8^{E-KO}* model. There, we compared keratinocyte-restricted vs systemic STING deletion, and found that only full-body STING knockout fully phenocopied ZBP1 deficiency, abolishing both local skin necroptosis and systemic inflammatory disease. These results affirm that STING activation in immune cells is necessary for initiating and sustaining the ZBP1–RIPK3–MLKL cascade, whereas STING in epithelial cells amplifies and executes tissue damage (Fig.5).

This strongly supports the notion that immune cell–intrinsic STING activation is the initiating event, and that inter-compartmental cytokine crosstalk drives full-blown necroptosis-induced inflammatory disease in both mouse models and human SAVI.

Question: Finally, authors should pursue some evidence whether apoptotic and necroptotic (or other broader) outcomes are contributing, particularly given the observations complex pyroptotic, apoptotic and necroptotic signaling proposed for PANoptosis and the PANoptosome (Man and Kanneganti, 2024; Pandian and Kanneganti, 2024).

We would like to thank the reviewer for raising this point regarding the potential contribution of PANoptosis, a coordinated, inflammasome-driven cell death modality proposed to integrate pyroptotic, apoptotic, and necroptotic signaling through shared adaptors such as ZBP1 and kinases like RIPK3. Based on our comprehensive genetic, transcriptional, and functional data, we believe we can confidently exclude PANoptosis as a relevant effector mechanism in the context of STING gain-of-function (GOF)–driven inflammation, both in human SAVI patients and in the *N153S* mouse model and also in our *Casp8^{E-KO}*-driven lethal dermatitis model.

First, PANoptosis requires the simultaneous activation of multiple executioner arms of cell death typically involving caspase-1–dependent pyroptosis (via GSDMD), RIPK3–MLKL-mediated necroptosis, and caspase-8–driven apoptosis. However, in our model, necroptosis clearly dominates both temporally and mechanistically. We observe robust and consistent upregulation of ZBP1 and MLKL in human SAVI patients and in the tissue obtained from the *N153S* mice, with corresponding pMLKL staining indicative of functional necroptotic activity which in the skin of these mice takes place without any substantial cl-Casp3 (Fig.6f). This suggest that if apoptosis takes place in the skin, this is certainly not concomitant to necroptosis hence excluding PANoptosis.

Second, although cl-Casp3 is observed in some compartments (e.g., spleen, lung and thymus), this is neither temporally nor spatially coincident with necroptosis. Rather, it reflects a downstream consequence of tissue remodelling, immune depletion, or secondary apoptotic responses, as shown by the lack of overlap between cl-Casp3 positivity and sites of pMLKL expression. Moreover, genetic deletion of Ripk3 in *N153S* mice provides a significant rescue of early immunological defects (e.g., thymic DN2/DN3 loss, T cell lymphopenia, and extramedullary erythropoiesis), confirming that RIPK3-dependent necroptosis alone accounts for these primary disease-driving events. Importantly, the early and profound immune cell depletion (e.g., DN2/DN3 thymocytes) that is rescued in *Ripk3*^{-/-} mice aligns with previously described models where RIPK3-dependent necroptosis governs developmental or stress-induced loss of lymphoid cells¹⁷. Furthermore, the rescue of extramedullary erythropoiesis and normalisation of splenic Ter119⁺CD71⁺ precursors reflect a reversal of systemic inflammation, typically, secondary to DAMP-driven cytokine cascades known to be initiated by necroptosis (Extended Data Fig. 14). Many of the rescued cellular compartments are not known sites of pyroptosis, and importantly, we observe no evidence of cell death modalities converging simultaneously, as required for PANoptosis. In patients, we find no evidence of inflammasome engagement or pyroptotic execution: in contrast to necroptosis executioners there is no persistent transcriptional upregulation of canonical pyroptosis markers such as *Gsdmd*, *Casp1*, or *Nlrp3* in many patient samples. Based on our whole-blood transcriptomics and cytokine data from SAVI patients, there is no dominant IL-1 β signature (Extended Data Fig. 11-13). Whilst IL-1 β is elevated in the serum, it is not disproportionately compared to IL-6 or CXCL10, and could be a downstream result of tissue inflammation rather than pyroptotic execution. Also, this inflammatory cytokine signature favouring IL-6 and CXCL10 is more consistent with type I IFN and DAMP-driven necroinflammation rather than PAMPs and inflammasome.

Third, the cell-type-specific nature of the observed death signatures argues strongly against a unifying PANoptotic mechanism. Instead, we demonstrate that STING-induced necroptosis occurs in a highly compartmentalised and temporally distinct manner: early and dominant in immune cells (e.g., thymocytes), and later and more heterogeneous in epithelial-rich tissues.

In the *Casp8*^{E-KO} model, this compartmentalisation is genetically resolved: TNFR1 drives an initial wave of inflammation and cell death that is fully rescued by *Tnfr1* deletion; however, a second, fully independent checkpoint mediated by STING–ZBP1–RIPK3–MLKL drives residual systemic inflammation that is completely suppressed only by co-deletion of STING or *Zbp1*. These findings demonstrate that STING governs two distinct, genetically separable necroptotic checkpoints: one overlapping partially with TNF signaling and one acting independently, again providing no evidence for concurrent activation of multiple cell death modalities, as would be expected in PANoptosis (Fig. 4,5)

Taken together, our data strongly support a model in which ZBP1-dependent necroptosis is the sole and primary RIPK3/MLKL-driven pathway responsible for immune dysfunction and tissue pathology in STING GOF conditions. We do not find evidence of coordinated activation of multiple cell death pathways in a PANoptotic manner. Rather, our findings argue for distinct, cell type-specific engagement of either necroptosis or apoptosis, depending on tissue context and immune status. These conclusions are further reinforced by the lack of biochemical or transcriptional hallmarks of pyroptosis, and by the clear compartmental and genetic dissection of necroptotic signaling observed in both SAVI and *Casp8*^{E-KO} models.

Other points:

1. The ZBP1-RIPK3 pathway was first revealed soon after the RIPK1-RIPK3 pathway when cytomegalovirus-induced necroptosis was shown to be independent of RIPK1 (Upton et al., 2010) and completely dependent on ZBP1 (Upton et al., 2011). This era also brought to light RIPK3 knockout rescue of the embryonic lethality exhibited by CASP8-deficient mice, a publication (Kaiser et al., 2011; Oberst et al., 2011) revealed the dependency of virus-induced ZBP1-RIPK3 necroptosis on CASP8 inhibition. Despite being a necroptotic death pathway independent of RIPK1, a requirement for CASP8 is already known to be shared across the three necroptotic pathways, ZBP1-RIPK3 and TRIF-RIPK3 (He et al., 2011; Kaiser et al., 2013) as well as the more well studied RIPK1-RIPK3 pathway. An unknown remains why RIPK1-independent necroptosis would still be dependent on CASP8 compromise when all available evidence points to specific cleavage-mediated control of RIPK1 (Newton et al., 2019) and not over RIPK3 (Newton et al., 2024), although all three pathways have not been fully evaluated in this regard (and should be).

2. This work follows considerable attention to STING and ZBP1 in host defense; however, with particular focus on the induction of intrinsic apoptosis over either extrinsic apoptosis or necroptosis. Thus, STING is best known to drive inflammatory cytokine induction and intrinsic apoptosis, a pathway that is not really considered or eliminated here by authors. Literature that deals with STING and cell death pathways that limit levels of both viral replication and inflammation might be useful in this regard (reviewed in Paludan et al., 2019, 2024; Zhan et al., 2024). Extensive studies have implicated apoptosis of microglia cells in host control over herpesvirus encephalitis (Reinhart et al., 2021) acting via microbe- and inflammation-restricting mechanisms (MIMs; Paludan et al., 2024). It is well known that RIPK3 function controls necroptosis and inflammasome activation in addition to CASP8-dependent apoptosis in a conditional fashion (Moriwaki and Chan, 2017;) and has been observed in influenza (Man and Kanneganti, 2024) as well as herpesvirus studies in mice (Jeffries et al., 2022; Guo et al., 2022). Along these lines, RIPK3 inactivating mutations have been implicated in susceptibility to herpesvirus encephalitis in a human patient (Liu et al., 2023) in a pattern consistent with altered cell death susceptibility but this patient did not exhibit susceptibility to any other notable diseases.

We thank the reviewer for the thoughtful contextual remarks highlighting the evolution of the necroptosis field, particularly the early discoveries of ZBP1–RIPK3 signaling and its intersection with viral sensing, as well as the conditional requirements for caspase-8 and RIPK1 cleavage in shaping these responses. While our study was not designed to readdress these historical mechanisms directly, we appreciate the opportunity to clarify how our findings build on and extend them in the context of STING GOF autoinflammatory disease and Casp8-deficient inflammation.

1. On the role of RIPK1 cleavage and the necessity of caspase-8 for necroptosis suppression:

The reviewer correctly notes that both RIPK1-dependent and -independent necroptosis pathways (e.g., ZBP1–RIPK3 and TRIF–RIPK3) appear to require caspase-8 for suppression, even though the best-characterised regulatory mechanism involves caspase-8–mediated cleavage of RIPK1. In our study, we circumvent this complexity by genetically deleting caspase-8 specifically in keratinocytes (*Casp8^{E-KO}*), thereby fully ablating RIPK1 cleavage and apoptotic control in this compartment. This allowed us to probe necroptotic pathways, including TNFR1- and ZBP1-driven checkpoints, without relying on indirect interpretations of cleavage-deficient RIPK1 alleles.

Importantly, our biochemical data provide further mechanistic clarity. In both *Casp8^{-/-}*; *Mik1^{-/-}* and *Fadd^{-/-}*; *Mik1^{-/-}* cells, as well as in cells treated with a STING agonist (DMXAA), we detect robust recruitment of full-length, activated RIPK1 into ZBP1 immunoprecipitates, alongside RIPK3 (Fig1 f, g). Notably, STING activation enhances this complex formation, correlating with increased RIPK1

and RIPK3 phosphorylation. These results suggest that RIPK1 cleavage is not simply bypassed in ZBP1-driven necroptosis, but instead may be co-opted into the signaling platform under conditions of enhanced ZBP1 activation, particularly in the absence of Casp8 or upon STING-induced upregulation of ZBP1 and Z-nucleic acids which would favour a ZBP1 platform over the FADD. This observation reinforces the concept that RIPK1 cleavage may serve as a broader checkpoint in multiple necroptotic contexts, not just in TNF signaling, but also when ZBP1 is activated in inflammatory states. In the *N153S* mouse model, where Casp8 is intact, we instead show that sustained STING activation overrides Casp8-mediated suppression, likely by amplifying ZBP1 and MLKL expression beyond a threshold compatible with homeostatic control. This supports a model in which chronic IFN signaling circumvents the classical Casp8 checkpoint, enabling cell death in tissues and cell types not typically susceptible to necroptosis under basal conditions. Thus, STING-driven necroptosis in SAVI appears to occur in a Casp8-independent manner. Together with our biochemical findings, it is tempting to speculate that under conditions of elevated STING signaling, RIPK1 may be redistributed into a ZBP1-governed platform that operates independently of FADD and Casp8. In this model, the existence of two competing platforms, a canonical RIPK1-RIPK3-FADD-Casp8-containing complex and an alternative ZBP1-RIPK1-RIPK3 complex, could explain how necroptosis proceeds even in the presence of active Casp8, as RIPK1 becomes sequestered away from caspase-8-mediated cleavage and suppression.

2. On intrinsic apoptosis and the broader role of STING in host defence:

We agree that STING has been extensively studied for its role in driving intrinsic apoptosis, particularly via mitochondrial BAX/BAK-mediated pathways in the context of antiviral defence and tumour surveillance. However, in the *N153S* model of chronic IFN-driven autoinflammation (as well as in SAVI patients), our data identify ZBP1-RIPK3-MLKL-dependent necroptosis as the primary effector mechanism. It has indeed been reported that STING activation drive the upregulation of pro-apoptotic protein such as BIM and PUMA. Notably, transcriptional profiling of whole blood from SAVI patients before and after treatment reveals no significant upregulation of pro-apoptotic genes, further arguing against intrinsic apoptosis as a dominant contributor to disease pathogenesis. Instead, the transcriptomic signature is dominated by interferon-stimulated genes (ISGs), with pronounced upregulation of ZBP1 and MLKL, consistent with the activation of necroptosis rather than apoptosis.

Indeed, we do detect cl-Casp3 in the spleen and lung at terminal stages, suggesting engagement of apoptosis. However, these apoptotic markers emerge after the onset of inflammatory disease and indicating possibly, downstream or bystander damage rather than initiating mechanisms. In contrast, RIPK3 deletion robustly alleviates early thymic and splenic T cell loss, erythropoietic defects, and systemic inflammation, pointing to necroptosis as the initiating insult in immune compartments.

3. Clarifying the scope of cell death outcomes and excluding PANoptosis:

The reviewer refers to RIPK3's broader involvement in inflammasome activation and the conceptual model of PANoptosis, which posits the simultaneous engagement of pyroptosis, apoptosis, and necroptosis via multiprotein "PANoptosome" complexes. We appreciate this point and address it directly. We have extensively answered this point above which we believe provide all the arguments necessary to argue against pyroptosis and/or PANoptosis.

Taken together, our human and murine data strongly support a model in which STING-ZBP1-RIPK3-MLKL necroptosis is the dominant driver of pathology in STING GOF autoinflammatory disease. While STING has the capacity to engage other forms of cell death, including apoptosis and pyroptosis, we find no molecular, genetic, or cytokine-level evidence for pyroptosis or PANoptosis in this setting. Instead, the patterns of cell death are cell-type-specific, genetically dissectable, and

temporally resolved, reinforcing the idea that necroptosis is the initiating effector mechanism in immune and epithelial compartments. The presence of cl-Casp3 in late-stage lesions likely reflects secondary consequences of unresolved inflammation rather than a coordinated or primary death pathway.

We now clarify this interpretation in the revised manuscript, including reference to the patient-specific gene expression data and their importance in excluding pyroptosis-related signatures.

Minor correction:

Typo (citations): "Unlike deletion of RIPK3 or MLKL, loss of tumor necrosis factor receptor 1 (TNFR1) does not prevent, but only delays the embryonic lethality observed in Casp8 -/- mice and the lethal dermatitis of Casp8 E-KO mice^{1,7}." Superscript 1,7 should read 1-7.

These have all been corrected.

References:

- 1 Zhang, T. *et al.* Influenza Virus Z-RNAs Induce ZBP1-Mediated Necroptosis. *Cell* **180**, 1115-1129 e1113, doi:10.1016/j.cell.2020.02.050 (2020).
- 2 DeAntoneo, C., Herbert, A. & Balachandran, S. Z-form nucleic acid-binding protein 1 (ZBP1) as a sensor of viral and cellular Z-RNAs: walking the razor's edge. *Curr Opin Immunol* **83**, doi:ARTN 102347 10.1016/j.coi.2023.102347 (2023).
- 3 Liccardi, G. *et al.* RIPK1 and Caspase-8 Ensure Chromosome Stability Independently of Their Role in Cell Death and Inflammation. *Mol Cell* **73**, 413-+, doi:10.1016/j.molcel.2018.11.010 (2019).
- 4 Boege, Y. *et al.* A Dual Role of Caspase-8 in Triggering and Sensing Proliferation-Associated DNA Damage, a Key Determinant of Liver Cancer Development. *Cancer Cell* **32**, 342-359 e310, doi:10.1016/j.ccell.2017.08.010 (2017).
- 5 Siedel, H., Roers, A., Rosen-Wolff, A. & Luksch, H. Type I interferon-independent T cell impairment in a Tmem173 N153S/WT mouse model of STING associated vasculopathy with onset in infancy (SAVI). *Clin Immunol* **216**, 108466, doi:10.1016/j.clim.2020.108466 (2020).
- 6 Motwani, M. *et al.* Hierarchy of clinical manifestations in SAVI N153S and V154M mouse models. *Proc Natl Acad Sci U S A* **116**, 7941-7950, doi:10.1073/pnas.1818281116 (2019).
- 7 Koerner, L. *et al.* ZBP1 causes inflammation by inducing RIPK3-mediated necroptosis and RIPK1 kinase activity-independent apoptosis. *Cell Death Differ* **31**, 938-953, doi:10.1038/s41418-024-01321-6 (2024).
- 8 Volpi, S. *et al.* Efficacy and Adverse Events During Janus Kinase Inhibitor Treatment of SAVI Syndrome. *J Clin Immunol* **39**, 476-485, doi:10.1007/s10875-019-00645-0 (2019).
- 9 Bin Khathlan, Y., Almutairi, S., Albadr, F. B., Alangari, A. A. & Alsultan, A. Case report: Durable response to ruxolitinib in a child with TREG1-related disorder. *Front Pediatr* **11**, 1178919, doi:10.3389/fped.2023.1178919 (2023).
- 10 Verstovsek, S. *et al.* Long-term outcomes of 107 patients with myelofibrosis receiving JAK1/JAK2 inhibitor ruxolitinib: survival advantage in comparison to matched historical controls. *Blood* **120**, 1202-1209, doi:10.1182/blood-2012-02-414631 (2012).
- 11 Rodriguez, D. A. *et al.* Caspase-8 and FADD prevent spontaneous ZBP1 expression and necroptosis. *Proc Natl Acad Sci U S A* **119**, e2207240119, doi:10.1073/pnas.2207240119 (2022).
- 12 Ros, U. *et al.* MLKL activity requires a splicing-regulated, druggable intramolecular interaction. *Mol Cell* **85**, 1589-1605 e1512, doi:10.1016/j.molcel.2025.03.015 (2025).
- 13 Barber, G. N. STING-dependent cytosolic DNA sensing pathways. *Trends Immunol* **35**, 88-93, doi:10.1016/j.it.2013.10.010 (2014).
- 14 Ahn, J. & Barber, G. N. Self-DNA, STING-dependent signaling and the origins of autoinflammatory disease. *Curr Opin Immunol* **31**, 121-126, doi:10.1016/j.coi.2014.10.009 (2014).

- 15 Chen, D. *et al.* PUMA amplifies necroptosis signaling by activating cytosolic DNA sensors. *Proc Natl Acad Sci U S A* **115**, 3930-3935, doi:10.1073/pnas.1717190115 (2018).
- 16 Zhang, X. *et al.* RIPK3-MLKL necroptotic signalling amplifies STING pathway and exacerbates lethal sepsis. *Clin Transl Med* **13**, e1334, doi:10.1002/ctm2.1334 (2023).
- 17 Alvarez-Diaz, S. *et al.* The Pseudokinase MLKL and the Kinase RIPK3 Have Distinct Roles in Autoimmune Disease Caused by Loss of Death-Receptor-Induced Apoptosis. *Immunity* **45**, 513-526, doi:10.1016/j.immuni.2016.07.016 (2016).

REVIEWER 1 comments:

3) *Cytosolic DNA is detected in Casp8 Mlkl DKO MEFs. They posit that the cytosolic DNA stems from genomic DNA damage and engages cGAS in Casp8 KO cells. I think this is highly speculative since Casp8 IEC KOs or E-KOs rescued by Ripk3 KO or Mlkl KO don't show a propensity for tumor development as you would expect from ongoing DNA damage.*

And connected to this:

5. *EDF9i – if DNA damage is upstream of cGAS-STING activation, then do these markers still appear in Casp8 Mlkl DKO or Casp8 E-KO STING TNFR1 DKO skin? WB for these markers should also be provided to indicate that proteins of the appropriate MW are being detected.*

We respectfully disagree with the assessment that the presence of cytosolic DNA in Casp8 Mlkl DKO MEFs is speculative or insufficiently justified. Importantly, this is not a central claim of our study. Rather, our data demonstrate that cGAS-dependent recognition of cytosolic DNA is required to drive STING activation in the context of Casp8 deficiency. After excluding mitochondrial DNA as a source, we detect DNA damage and mitotic defects, which are consistent with peer reviewed literature, including our own, establishing a role for Casp8 in preserving chromosomal and genomic integrity.

This conclusion is strongly supported by Boege et al. (Cancer Cell, 2017), who showed that loss of Casp8 induces phosphorylation of H2AX and generates a DNA damage response. In their work, the authors propose that caspase-8-containing complexes help preserve genomic integrity, and that Casp8 deficiency promotes replication errors and transformation, independently of its role in cell death. Their work thus provides a conceptual framework for interpreting the accumulation of cytosolic DNA in Casp8-deficient settings, even in the absence of overt tumorigenesis.

Consistent with this, Liccardi et al. (Mol Cell, 2019) (work from my postdoc in London in the lab of Pascal Meier) identified a role for Casp8 and RIPK1 in mitotic progression and chromosomal stability. Similarly in Müller et al. (Mol Cell, 2020) it was shown that Casp8-deficient cells can bypass the p53-dependent G2/M checkpoint, further linking Casp8 to genome surveillance.

Importantly, a seminal study from the laboratory of David Wallach by Krelin et al. (Cell Death & Differentiation, 2008) demonstrated that Casp8-deficient MEFs, when transplanted into immunodeficient mice, formed tumors, providing the first *in vivo* evidence that loss of Caspase-8 promotes cellular transformation. At the time, the authors discussed whether this outcome could be attributed solely to the loss of apoptosis or other unknown roles of Casp8. This foundational work has since catalyzed an entire field of research into non-apoptotic functions of Caspase-8, particularly its emerging and well published role in the maintenance of genomic stability.

Also, whilst loss of casp8 has been linked to transformation *in vivo*, it is important to note that persistent DNA damage and resulting cytosolic DNA accumulation do not invariably result in tumorigenesis. At least three independent genetic models, BubR1-insufficient mice (DOI: 10.1038/ng1382), Rnaseh2b KO (DOI: 10.1084/jem.20120876), and Samhd1 KO (DOI: 10.1084/jem.20220829), develop chronic DNA damage without spontaneous tumor formation, illustrating that DNA damage and genetic instability can drive inflammation independently of cancer thus providing a strong published counter argument to this point raised by reviewer 1.

Regarding point 5, immunohistochemical staining for DNA damage markers in Casp8 Mlkl DKO and Casp8 E-KO Tnfr1 Sting DKO skin is already underway and will be included in the revised manuscript. Given that IHC provides spatially resolved, cell-type-specific insight into DNA damage

signalling, we consider additional immunoblotting redundant, though we are happy to follow editorial guidance on this point.

Second important point

4) Tissues from Sting N153S KI mice that model human SAVI exhibit markers of necroptosis (pMLKL) and apoptosis (CC3). As a sanity check for the specificity of their pMLKL IHC, does the pMLKL signal in the Sting KI mice (Fig. 6f) go away in Sting KI Ripk3 KO mice? Although some disease manifestations in the Sting KI mice are ameliorated by RIPK3 deficiency, this genetic data doesn't prove a contribution of necroptosis to disease since RIPK3 also has necroptosis-independent functions. In summary, I like the new genetic data in the Casp8 E-KO model placing cGAS and STING upstream of Zbp1- and Tnfr1-driven necroptosis. However, in my opinion, the data for necroptosis being an important driver of disease in SAVI remains tenuous. Caspase-8 is still present in this context and so would be expected to act as a brake on necroptosis signaling.

The assessment that the evidence for necroptosis as a pathogenic driver in SAVI remains tenuous seems unjustified. On the contrary, we believe that our manuscript provides the most comprehensive analysis to date, across both longitudinally obtained patient-derived samples and the N153S murine model, directly linking inflammatory cell death mechanisms to disease pathology in SAVI. To address the reviewer's specific concern, we will include the requested pMLKL IHC staining in N153S; Ripk3^{-/-} skin, which shows a clear rescue of the skin lesions and complete loss of pMLKL staining. This confirms both the specificity of the IHC signal and the contribution of RIPK3-dependent necroptosis to tissue inflammation.

Importantly, the reviewer suggestion that RIPK3 may be acting through necroptosis-independent functions in this context feels overstated. While RIPK3 has been implicated in supporting apoptosis under certain experimental conditions, the overwhelming body of evidence, particularly from *in vivo* studies, indicates that these roles are generally limited and invariably occur in the context of concomitant necroptotic signalling. For instance, in the Sharpin model, RIPK3 deletion provides only modest and delayed protection from apoptosis-driven skin inflammation, with partial rescue in approximately 50% of animals over several weeks (Rickard et al., 2014, eLife; Kumari et al., 2014, eLife from Silke, Walczak and Pasparakis labs), highlighting the limited impact of RIPK3 outside of necroptosis which is clearly responsible for the systemic inflammation observed in spleen and liver. Moreover, in systemic inflammatory response syndrome (SIRS) models on the C57BL/6N background, RIPK3's contribution to apoptosis is minimal, particularly under high-dose TNF, where its apoptotic role is effectively absent (Newton et al., 2016, Cell Death & Differentiation; Duprez et al., 2011, Immunity from the labs of Kim Newton and Peter Vandernabele).

Our own recent work (Kelepouras et al., 2024, Cell Death & Differentiation) further supports this view. This comprehensive study validates pMLKL-S345 immunohistochemistry using Ripk3^{-/-} and Mkl1^{-/-} controls and demonstrates tissue-specific necroptotic responses across multiple disease models, including the Sharpin model, where necroptosis primarily affects spleen tissue while apoptosis predominates in skin, and in SIRS, where TNF-induced cell death is exclusively apoptotic. Hence, the detection of pMLKL in any tissue and its loss following RIPK3 deletion clearly provides evidence that necroptosis occurs spatially and temporally and contributes directly to the inflammatory phenotype of the disease under investigation. It is also important to note that, to date, no study has demonstrated RIPK3-mediated apoptosis (or relevance in any other cell death mechanism) in the absence of necroptotic signalling or execution. Thus, we consider highly unlikely,

and in fact speculative, that the protective effect observed in N153S; Ripk3^{-/-} mice is attributable to non-necroptotic functions of RIPK3 that, even if this were the case, would exclude or be more relevant than necroptosis.

Regarding the reviewer's statement that "*Caspase-8 is still present in this context and so would be expected to act as a brake on necroptosis signaling,*" we would like to draw to your attention that this concern was directly raised and addressed during the prior revision. Specifically, at the reviewer-1's request, we experimentally demonstrated that STING activation induces the formation of a necroptotic ZBP1-RIPK1-RIPK3 complex independently of both FADD and Caspase-8. This was shown by the requested and provided endogenous complex formation in Casp8⁻ and Fadd-deficient cells, as well as in MLKL deficient setting where, in the latter, Casp8 and FADD are still expressed. These findings in Figure 2 indicate that under conditions of STING activation, which are used to model its chronic activation relevant to SAVI, necroptosis proceeds through a ZBP1-based platform that does not require or engage Caspase-8.

This mechanistic insight directly explains why Caspase-8 does not inhibit necroptosis in the SAVI context: the inflammatory cell death pathway is initiated independently of FADD recruitment, and thus Caspase-8 is not brought into proximity with RIPK1 and RIPK3 to exert its suppressive function via proximal cleavage of RIPK1 as published by Kim Newton et al 2024 Cell Death & Diff. and Tran et al. 2024 Cell Death & Diff. We believe this constitutes strong genetic and biochemical evidence that STING-driven necroptosis bypasses the conventional Caspase-8 brake and validates our use of the term "TNFR1/FADD-independent necroptosis" in the manuscript title. These findings also justify the translational relevance of our model to STING gain-of-function mutations, where Caspase-8 expression is preserved but functionally irrelevant in the context of ZBP1-mediated necroptosis. As above we are happy to follow editorial guidance on this point.

REVIEWER 1 comments:

3) *Cytosolic DNA is detected in Casp8 Mlkl DKO MEFs. They posit that the cytosolic DNA stems from genomic DNA damage and engages cGAS in Casp8 KO cells. I think this is highly speculative since Casp8 IEC KOs or E-KOs rescued by Ripk3 KO or Mlkl KO don't show a propensity for tumor development as you would expect from ongoing DNA damage.*

And connected to this:

5. *EDF9i – if DNA damage is upstream of cGAS-STING activation, then do these markers still appear in Casp8 Mlkl DKO or Casp8 E-KO STING TNFR1 DKO skin? WB for these markers should also be provided to indicate that proteins of the appropriate MW are being detected.*

We respectfully disagree with the assessment that the presence of cytosolic DNA in Casp8 Mlkl DKO MEFs is speculative or insufficiently justified. Importantly, this is not a central claim of our study. Rather, our data demonstrate that cGAS-dependent recognition of cytosolic DNA is required to drive STING activation in the context of Casp8 deficiency. After excluding mitochondrial DNA as a source, we detect DNA damage and mitotic defects, which are consistent with peer reviewed literature, including our own, establishing a role for Casp8 in preserving chromosomal and genomic integrity.

This conclusion is strongly supported by Boege et al. (Cancer Cell, 2017), who showed that loss of Casp8 induces phosphorylation of H2AX and generates a DNA damage response. In their work, the authors propose that caspase-8-containing complexes help preserve genomic integrity, and that Casp8 deficiency promotes replication errors and transformation, independently of its role in cell death. Their work thus provides a conceptual framework for interpreting the accumulation of cytosolic DNA in Casp8-deficient settings, even in the absence of overt tumorigenesis.

Consistent with this, Liccardi et al. (Mol Cell, 2019) (work from my postdoc in London in the lab of Pascal Meier) identified a role for Casp8 and RIPK1 in mitotic progression and chromosomal stability. Similarly in Müller et al. (Mol Cell, 2020) it was shown that Casp8-deficient cells can bypass the p53-dependent G2/M checkpoint, further linking Casp8 to genome surveillance.

Importantly, a seminal study from the laboratory of David Wallach by Krelin et al. (Cell Death & Differentiation, 2008) demonstrated that Casp8-deficient MEFs, when transplanted into immunodeficient mice, formed tumors, providing the first *in vivo* evidence that loss of Caspase-8 promotes cellular transformation. At the time, the authors discussed whether this outcome could be attributed solely to the loss of apoptosis or other unknown roles of Casp8. This foundational work has since catalyzed an entire field of research into non-apoptotic functions of Caspase-8, particularly its emerging and well published role in the maintenance of genomic stability.

Also, whilst loss of casp8 has been linked to transformation *in vivo*, it is important to note that persistent DNA damage and resulting cytosolic DNA accumulation do not invariably result in tumorigenesis. At least three independent genetic models, BubR1-insufficient mice (DOI: 10.1038/ng1382), Rnaseh2b KO (DOI: 10.1084/jem.20120876), and Samhd1 KO (DOI: 10.1084/jem.20220829), develop chronic DNA damage without spontaneous tumor formation, illustrating that DNA damage and genetic instability can drive inflammation independently of cancer thus providing a strong published counter argument to this point raised by reviewer 1.

Regarding point 5, immunohistochemical staining for DNA damage markers in Casp8 Mlkl DKO and Casp8 E-KO Tnfr1 Sting DKO skin is already underway and will be included in the revised manuscript. Given that IHC provides spatially resolved, cell-type-specific insight into DNA damage

signalling, we consider additional immunoblotting redundant, though we are happy to follow editorial guidance on this point.

Second important point

4) Tissues from Sting N153S KI mice that model human SAVI exhibit markers of necroptosis (pMLKL) and apoptosis (CC3). As a sanity check for the specificity of their pMLKL IHC, does the pMLKL signal in the Sting KI mice (Fig. 6f) go away in Sting KI Ripk3 KO mice? Although some disease manifestations in the Sting KI mice are ameliorated by RIPK3 deficiency, this genetic data doesn't prove a contribution of necroptosis to disease since RIPK3 also has necroptosis-independent functions. In summary, I like the new genetic data in the Casp8 E-KO model placing cGAS and STING upstream of Zbp1- and Tnfr1-driven necroptosis. However, in my opinion, the data for necroptosis being an important driver of disease in SAVI remains tenuous. Caspase-8 is still present in this context and so would be expected to act as a brake on necroptosis signaling.

The assessment that the evidence for necroptosis as a pathogenic driver in SAVI remains tenuous seems unjustified. On the contrary, we believe that our manuscript provides the most comprehensive analysis to date, across both longitudinally obtained patient-derived samples and the N153S murine model, directly linking inflammatory cell death mechanisms to disease pathology in SAVI. To address the reviewer's specific concern, we will include the requested pMLKL IHC staining in N153S; Ripk3^{-/-} skin, which shows a clear rescue of the skin lesions and complete loss of pMLKL staining. This confirms both the specificity of the IHC signal and the contribution of RIPK3-dependent necroptosis to tissue inflammation.

Importantly, the reviewer suggestion that RIPK3 may be acting through necroptosis-independent functions in this context feels overstated. While RIPK3 has been implicated in supporting apoptosis under certain experimental conditions, the overwhelming body of evidence, particularly from *in vivo* studies, indicates that these roles are generally limited and invariably occur in the context of concomitant necroptotic signalling. For instance, in the Sharpin model, RIPK3 deletion provides only modest and delayed protection from apoptosis-driven skin inflammation, with partial rescue in approximately 50% of animals over several weeks (Rickard et al., 2014, eLife; Kumari et al., 2014, eLife from Silke, Walczak and Pasparakis labs), highlighting the limited impact of RIPK3 outside of necroptosis which is clearly responsible for the systemic inflammation observed in spleen and liver. Moreover, in systemic inflammatory response syndrome (SIRS) models on the C57BL/6N background, RIPK3's contribution to apoptosis is minimal, particularly under high-dose TNF, where its apoptotic role is effectively absent (Newton et al., 2016, Cell Death & Differentiation; Duprez et al., 2011, Immunity from the labs of Kim Newton and Peter Vandernabele).

Our own recent work (Kelepouras et al., 2024, Cell Death & Differentiation) further supports this view. This comprehensive study validates pMLKL-S345 immunohistochemistry using Ripk3^{-/-} and Mkl1^{-/-} controls and demonstrates tissue-specific necroptotic responses across multiple disease models, including the Sharpin model, where necroptosis primarily affects spleen tissue while apoptosis predominates in skin, and in SIRS, where TNF-induced cell death is exclusively apoptotic. Hence, the detection of pMLKL in any tissue and its loss following RIPK3 deletion clearly provides evidence that necroptosis occurs spatially and temporally and contributes directly to the inflammatory phenotype of the disease under investigation. It is also important to note that, to date, no study has demonstrated RIPK3-mediated apoptosis (or relevance in any other cell death mechanism) in the absence of necroptotic signalling or execution. Thus, we consider highly unlikely,

and in fact speculative, that the protective effect observed in N153S; Ripk3^{-/-} mice is attributable to non-necroptotic functions of RIPK3 that, even if this were the case, would exclude or be more relevant than necroptosis.

Regarding the reviewer's statement that "*Caspase-8 is still present in this context and so would be expected to act as a brake on necroptosis signaling,*" we would like to draw to your attention that this concern was directly raised and addressed during the prior revision. Specifically, at the reviewer-1's request, we experimentally demonstrated that STING activation induces the formation of a necroptotic ZBP1-RIPK1-RIPK3 complex independently of both FADD and Caspase-8. This was shown by the requested and provided endogenous complex formation in Casp8⁻ and Fadd-deficient cells, as well as in MLKL deficient setting where, in the latter, Casp8 and FADD are still expressed. These findings in Figure 2 indicate that under conditions of STING activation, which are used to model its chronic activation relevant to SAVI, necroptosis proceeds through a ZBP1-based platform that does not require or engage Caspase-8.

This mechanistic insight directly explains why Caspase-8 does not inhibit necroptosis in the SAVI context: the inflammatory cell death pathway is initiated independently of FADD recruitment, and thus Caspase-8 is not brought into proximity with RIPK1 and RIPK3 to exert its suppressive function via proximal cleavage of RIPK1 as published by Kim Newton et al 2024 Cell Death & Diff. and Tran et al. 2024 Cell Death & Diff. We believe this constitutes strong genetic and biochemical evidence that STING-driven necroptosis bypasses the conventional Caspase-8 brake and validates our use of the term "TNFR1/FADD-independent necroptosis" in the manuscript title. These findings also justify the translational relevance of our model to STING gain-of-function mutations, where Caspase-8 expression is preserved but functionally irrelevant in the context of ZBP1-mediated necroptosis. As above we are happy to follow editorial guidance on this point.

Rebuttal letter

We would like to sincerely thank the reviewers for their careful and critical examination of our manuscript and for the thoughtful points raised throughout the review process. We also wish to express our gratitude to Reviewers 2 and 3 for their supportive and encouraging assessments and for already being in favour of publication of our work. We believe that the additional questions and suggestions raised in this second round have once again helped us to strengthen the significance and broaden the overall scientific message of our study. We truly appreciate the reviewer's commitment to improving our manuscript and hope that these comprehensive new data and clarifications will fully address any remaining concerns and meet your satisfaction.

Below, you will find a detailed, point-by-point rebuttal in which each of your concerns has been addressed in blue, together with the corresponding new rebuttal figures, to simplify and smoothen the reviewer's assessment.

REVIEWER 1 comments:

The key pieces of new data are:

1) Casp8 E-KO Sting KO and Casp8 E-KO cGAS KO mice show better survival than Casp8 E-KO Zbp1 KO mice (Fig. 4a and EDF9a), arguing that Sting does more than just trigger ZBP1-driven necroptosis.

This fits with their model that STING activation also promotes TNFR1-induced necroptosis in Casp8 KO keratinocytes.

Although they show that TNF protein is less evident in Casp8 E-KO Sting E-KO skin compared with Casp8 E-KO skin (EDF4e), it would be helpful if they included Tnf transcript levels in the Fig. 2g-i heatmaps.

In Extended figure 4e we show the decrease in TNF cytokine levels from skin lysates which we report below for the reviewer reference (Figure 1A). We have now also performed RNA sequencing on *Casp8^{E-KO}; Sting^{KO}* which we compare to *Casp8^{E-KO}; Tnfr1^{KO}* and *Casp8^{E-KO}; Sting^{E-KO}* (Figure 1 B now included in Extended data Fig. 5g). These data show very clearly now that STING deletion clearly affects NF-kB inflammatory genes activation including but not limited to TNF. These data which are now included in Extended data Fig. 5g fully support our conclusions that STING also supports TNFR1-induced inflammatory necroptosis. Moreover, we have also added TNF transcript assessment via RT-q-PCR which conclusively show that TNF levels are clearly downregulated to control levels in *Casp8^{E-KO}; Sting^{KO}* mice (Figure 1C now included in Extended data Fig. 5h).

Rebuttal Figure 1:
 A) TNF cytokine levels measured in skin homogenates obtained from P5 mice of the indicated genotypes (n = 5 biological replicates).
 B) Differential gene expression analysis of significantly differentially expressed genes from 3' mRNA sequencing of skin samples from mice of the indicated genotypes (n = 4 biological replicates).
 C) RT-q-PCR analysis of TNF mRNA expression in skin samples from mice of the indicated genotypes (n = 3 biological replicates). Data were analyzed by one-way ANOVA with multiple comparisons.

These data are now included in the papers figures as mentioned. The text referring to the RNA-seq data and the RT-q-PCR is now added in lines 340-343 and reads: "Consistently, transcriptional upregulation of several inflammatory genes, including *Tnf*, *Zbp1* and *Mkl1* were completely suppressed in *Casp8^{E-KO}*; *Sting^{-/-}* mice and RNA-seq analysis revealed a similar downregulation for NF-κB signature genes compared to *Casp8^{E-KO}*; *Tnfr1^{-/-}* mice (Extended data Fig. 5f, g)". This is marked in the main text as per reviewer request.

2) *Casp8 E-KO Tnfr1 Sting* DKO mice survive to 14 wks without overt issues, just like *Casp8 E-KO Tnfr1 Zbp1* DKO mice. The authors speculate on lines 445-452 that STING activity outside of keratinocytes explains why *Casp8 E-KO Tnfr1 KO Sting E-KOs* are not rescued to the same extent as *Casp8 E-KO Tnfr1 Sting* DKO mice. The issue I have with this model is that if the initiating events in keratinocytes are blocked by the combined loss of *Tnfr1* and *Sting*, then what is triggering *Sting* activation in non-keratinocytes? It seems like the simpler/more plausible explanation for the difference is incomplete deletion of the conditional *Sting* allele. Given the limitations of the conditional model, they should temper their conclusions or provide further insights.

We have significantly tempered the conclusions and rephrased them alongside with the fact that perhaps the localized expression of ZBP1 in epidermal keratinocytes observed in the epidermis of the *Casp8^{E-KO}*; *Tnfr1^{KO}*; *Sting^{E-KO}* are indeed due to an incomplete deletion of STING from epidermal keratinocytes as this reviewer is suggesting. The text now reads: "Interestingly, a few cells in the epidermis of *Casp8^{E-KO}*; *Tnfr1^{-/-}*; *Sting^{E-KO}* mice surprisingly remained positive for ZBP1 staining (Extended Data Fig. 4f). We reasoned that ZBP1 upregulation in these cells could be due to incomplete STING depletion from epidermal keratinocyte, inducing keratinocyte-extrinsic mechanisms, likely driven by immune cells that might still depend on STING expression". This is now included in lines: 296-300 and marked via a comment as per request of this reviewer. Also this has been indicated in lines 364-368: "Histological examination of skin sections showed suppression of lethal dermatitis as evidenced by normalized epidermal thickness, loss of pMLKL and ZBP1 expression and CD45⁺, and c-Casp3-positive cells, at levels comparable to healthy controls (Fig. 5g). This striking phenotypic divergence confirms that the residual necroptosis observed in *Casp8^{E-KO}*; *Tnfr1^{-/-}*; *Sting^{E-KO}* mice was due to incomplete *Sting* deletion.." which is also marked as per reviewer requested

3) Cytosolic DNA is detected in *Casp8 Mkl1* DKO MEFs. They posit that the cytosolic DNA stems from genomic DNA damage and engages cGAS in *Casp8* KO cells. I think this is highly speculative since *Casp8* IEC KO or E-KOs rescued by *Ripk3* KO or *Mkl1* KO don't show a propensity for tumor development as you would expect from ongoing DNA damage. And connected to this:

5). EDF9i – if DNA damage is upstream of cGAS-STING activation, then do these markers still appear in *Casp8 Mkl1* DKO or *Casp8 E-KO STING TNFR1* DKO skin? WB for these markers should also be provided to indicate that proteins of the appropriate MW are being detected.

The assessment of DNA damage and loss of genome integrity as a consequence of *Casp8* deficiency independently of necroptosis activation is well documented in several publication which we briefly mention below. Importantly, we would like to underline that this is not a central claim of our study for which we don't want to take any merit in terms of novelty. Rather, our data demonstrate the dependency of the lethal dermatitis in the *Casp8^{E-KO}* mice on cGAS and thus on its ability to recognise cytosolic DNA known to be required to drive STING activation. As a result of a rather meticulous analysis which led to the exclusion of mitochondrial DNA as a source, we report DNA damage and mitotic defects in *Casp8* deficient cells which has not only been extensively shown before but also published to be a source of cytosolic DNA accumulation known to lead to cGAS/STING activation.

The published evidence that *Casp8* loss has a role in genome integrity follows:

Boege et al. (Cancer Cell, 2017), clearly state: “Dissecting the role of caspase-8 for hepatocyte apoptosis, we discovered a non-apoptotic function of caspase-8 in H2AX phosphorylation. Firstly, by performing PHX in C57BL/6 and *Casp8^{hep-KO}* mice, we show that caspase-8 is needed for an efficient DDR to replication stress” arguing more so that “loss of caspase-8 can be expected to be genotoxic and generate an environment of genetic instability. In line with these findings, caspase-8 deficiency has been shown to facilitate cellular transformation independently of its killing function (Krelin et al., 2008). Loss of caspase-8 expression by either mutations or epigenetic silencing has been reported in murine and humanHCC (Liedtke et al., 2005; Soung et al., 2005). Therefore, it is conceivable that loss of caspase-8 in one and the same cell not only confers apoptosis resistance (a hallmark of cancer), but also promotes replication errors, and thus contributes to cancer development. Based on our observations, caspase-8 deficiency is thus expected to predispose to mutations in proliferating nonneoplastic hepatocytes”.

Consistent with this, also our own data, published in Liccardi et al. (Mol Cell, 2019) identified a role for Casp8 in maintaining chromosomal stability independently from cell death. Specifically, in Liccardi et al., it was shown that mitotic abnormalities are detectable in *Casp8^{-/-}* embryos at 10,5 as well as in *Casp8^{-/-}* and *Casp8^{-/-}; Mkl1^{-/-}* embryos at 13,5 a stage at which embryonic lethality is fully rescued by loss of necroptosis activation. We have reported below data extracted from the published manuscript (Figure 6 G, H) which are here indicated as Figure 2 A-B for this reviewer. The figure shows representative pictures from the indicated genotype, red arrows indicate representative mitotic abnormalities and graphs show the levels of abnormalities detected via the scoring of the correspondent number of abnormal mitosis. This published data shows a clear accumulation of mitotic defects in *Casp8^{-/-}*; *Mkl1^{-/-}* embryos. The study revealed a fundamental role of Casp8 activity in cleaving PLK1 during mitosis and regulating its ability to orchestrate correct chromosomal segregation whereby excessive or loss of Casp8 activity would drive accumulation of genetic instability.

Additionally in Vucur et al Cell reports 2013 the authors state t: “*We show that, in a setting of chronic inflammation induced by LPC-specific deletion of Tak1, RIP3-dependent necroptosis represents a pathway regulating the consequences of chronic inflammation in the liver by counteracting against Caspase-8 dependent compensatory proliferation of hepatocytes, immune cell activation, hepatic fibrogenesis, and the development of chromosomal aberrations leading to hepatocarcinogenesis.*”.

This is in agreement with the data of Liccardi et al whereby the enhanced Casp8 activation as a consequence of RIPK3 loss (reported in Vucur et al.) would affect chromosomal segregation and result in the chromosomal aberrations of chromosomes 4,8 and 13.

Consistently, Hakem et al Blood 2012 showed that loss of caspase-8 in B lymphocytes leads to B-cell malignancies and that “*deficiency of caspase-8 results in impaired cytokinesis and that casp8^{-/-} lymphomas display remarkably elevated levels of chromosomal aberrations. Our data support an important role for caspase-8 in the maintenance of genomic integrity and highlight its tumor-suppressive function.*”

Also, whilst loss of CASP8 has been linked to transformation *in vivo*, it is important to note that persistent DNA damage and resulting cytosolic DNA accumulation do not invariably result in tumorigenesis and therefore the loss of genome integrity as a consequence of Casp8 loss might not lead to tumour formation invariably in any tissue. Moreover, and more specifically to the arguments raised by the reviewer, it is also important to note that the limited survival of Casp8/RIPK3 DKO and in Casp8/MLKL DKO mice due to lymphadenopathies might mask or prevent the establishment of tumours that would otherwise be detectable later in life. KRAS-mutant driven pancreatic tumour are known to require 6 months to show any detectable signs of tumour and lung-specific combined RB mutation and P53 loss driven SCLC model is also known to require six months before showing any detectable tumour in the lungs of mice via MRI scanning.

It is also important to mention that not all genetic instability and DNA damage accumulation leads to tumours formation. At least three independent genetic models, BubR1-insufficient mice (DOI: 10.1038/ng1382), Rnaseh2b KO (DOI: 10.1084/jem.20120876), and Samhd1 KO (DOI: 10.1084/jem.20220829), develop chronic DNA damage and/or cytosolic DNA accumulation without spontaneous tumour formation, illustrating that DNA damage and genetic instability can drive inflammation independently of cancer. Hence it is rather possible that the Casp8-deficiency-induced loss of genome integrity does not lead to tumorigenesis thus providing a strong published counter argument to this point raised.

With regards to remark 5 we have now conducted stainings for p-H2A.X in the MEFs obtained from WT, Casp8 KO, Casp8/MLKL DKO and Casp8/RIPK3 DKO (Figure 3A panel for reviewer only) as well as in skin obtained from Casp8 KO Casp8/MLKL DKO mice, Casp8/RIPK3 DKO (Figure 3B partly contained in Fig. and Extended data fig.) as well as *Casp8^{E-KO}; Tnfr1^{KO}; Sting^{KO}* and *Casp8^{E-KO}; Tnfr1^{KO}; Zbp1^{KO}* mice (Figure 3C currently part of). Our data clearly show that Casp8 deficiency independently of necroptosis leads to accumulation of p-H2A.X positive cells. This signal is not exclusively detected in micro-nucleated cells or cells clearly undergoing genomic instability but also in apparently normal cells suggesting a dynamic process whereby it would be difficult to determine which one is consequence or cause (Figure 1 A-reviewer only panel). The other panels in the tissues obtained from the indicated genotypes at their respective end points clearly supports the conclusion of DNA damage accumulation independently of necroptosis as a consequence of Casp8 deficiency as p-H2A.X is detectable independently of MLKL or RIPK3 deficiency, STING/TNFR1 or ZBP1/TNFR1 co-deficiency. These are currently part of the manuscript. Specifically, Rebuttal Figure 3B is now Figure 5I in the manuscript as per request of reviewer 2-point 3 (please see below) whilst Rebuttal Figure 3C is included in Extended Data Figure 6f. Our work supports the hypothesis that loss of genome integrity might be responsible for the accumulation of cytosolic DNA. We hope that

the reviewer would now agree that these data rather than speculative seem respectfully confirmatory of what has already been amply published elsewhere.

We have lowered the tone of this conclusion and cited all the published work mentioned above. The text in lines 419-425 now reads: “Consistently, we observed similar levels of γ H2AX staining in skin obtained from *Casp8*^{-/-}; *Ripk3*^{-/-}, *Casp8*^{-/-}; *Mik1*^{-/-} as well as *Casp8*^{E-KO}; *Tnfr1*^{-/-}; *Sting*^{-/-} and *Casp8*^{E-KO}; *Tnfr1*^{-/-}; *Zbp1*^{-/-} mice (Extended Data Fig. 6f) suggesting that loss of Casp8 independently of necroptosis and cell death was responsible for loss of genome integrity as previously described²⁸⁻³¹. Taken together, these findings would support a model in which Casp8 loss-driven genomic DNA released due to mitotic dysfunction and genome instability, is probably the primary trigger of cGAS/STING activation in this context.” and it is marked as per reviewer request

Rebuttal Figure 3: Representative images of MEFs of indicated genotypes (A) or consecutive skin sections from mice with the indicated genotypes at P5 (B-D) stained with DAPI and p-H2A.X (A) or H&E, and pHH3-S10 and p-H2A.X-S139 ($n=5$ in each group) (B) or H&E, and p-H2A.X-S139 ($n=5$ in each group) (C-D)). Scale bars: 100 μ m (representative field) and 20 μ m or 10 μ m (magnified selected area) respectively.

4) Tissues from *Sting* N153S KI mice that model human SAVI exhibit markers of necroptosis (pMLKL) and apoptosis (CC3). As a sanity check for the specificity of their pMLKL IHC, does the pMLKL signal in the *Sting* KI mice (Fig. 6f) go away in *Sting* KI *Ripk3* KO mice? Although some disease manifestations in the *Sting* KI mice are ameliorated by RIPK3 deficiency, this genetic data doesn't prove a contribution of necroptosis to disease since RIPK3 also has necroptosis-independent functions. In summary, I like the new genetic data in the *Casp8* E-KO model placing cGAS and STING upstream of *Zbp1*- and *Tnfr1*-driven necroptosis. However, in my opinion, the data for necroptosis being an important driver of disease in SAVI remains tenuous. Caspase-8 is still present in this context and so would be expected to act as a brake on necroptosis signaling.

We believe that our manuscript provides the most comprehensive analysis to date, spanning both longitudinally collected patient-derived SAVI samples and the N153S murine model, directly linking inflammatory necroptotic cell death to disease pathology in SAVI. To address the reviewer's specific concern, we have now included the survival analysis of the *N153S; Ripk3^{-/-}* mice, which survive without showing any signs of disease and reach at least 35 weeks of age, which matches the median survival of N153S mice (Rebuttal Figure 4A now included in Fig. 6d, e). Additionally, we have added the requested pMLKL-S345 IHC staining in *N153S; Ripk3^{-/-}* tissues for skin, lung, and thymus, all of which show a clear loss of pMLKL-positive staining compared to the N153S controls (Rebuttal Fig 4B, now included in Fig. 6f). In this panel we also included the spleen, which showed no pMLKL-S345 positivity but did display a rescue of the splenic architecture, likely as a direct consequence of the early haematopoietic rescue observed in the *N153S; RIPK3^{-/-}* mice. This confirms both the specificity of the IHC signal and the contribution of RIPK3-dependent necroptosis to tissue inflammation. These data are now added in the text in lines 555-561 and marked in the text as per reviewer request.

Rebuttal Figure 4: Kaplan–Meier survival curves (A) and representative images (B) of mice of the indicated genotypes. (C) Representative images of consecutive skin, lung, spleen and thymus sections from *Sting^{N153S/wt}* and *Sting^{N153S/wt}; Ripk3^{KO}* at survival endpoint and age-matched *Ripk3^{KO}* littermate controls stained with H&E and pMLKL-S345 (n = 3 in each group). Scale bars: 100 μm (representative field) and 20 μm (magnified selected area).

Importantly, in light of these new data, the suggestion that RIPK3 may be acting through necroptosis-independent functions in this context feels no longer concerning.

While RIPK3 has been implicated in supporting apoptosis under certain experimental conditions, the overwhelming body of published evidence, particularly from *in vivo* studies, indicates that these roles are generally limited and invariably occur in the context of concomitant and predominant RIPK3-mediated necroptotic cell death. The two best documented examples of RIPK3's role in supporting apoptosis are the systemic inflammatory response syndrome (SIRS) and the Sharpin model. For instance, in SIRS models on the C57BL/6N background, RIPK3's contribution to apoptosis is minimal, especially under high-dose TNF, where its apoptotic role is effectively absent (Newton et al., 2016, Cell Death & Differentiation; Duprez et al., 2011, Immunity from the labs of Kim Newton and Peter Vandenabeele).

In the Sharpin model, RIPK3 deletion provides only modest and delayed protection from apoptosis-driven skin inflammation, with partial rescue in approximately 50% of animals manifesting only as a modestly delayed onset (Rickard et al., 2014, eLife; Kumari et al., 2014, eLife from the Silke, Walczak, and Pasparakis labs). However, this has never been sufficient to affect the actual survival (median or endpoint) of Sharpin mice, which are known to succumb to TNF/TNFR1/Casp8-mediated apoptosis-driven lethal dermatitis. This highlights the limited impact of RIPK3 outside of necroptosis; its main function in this model is to induce organ inflammation observed in spleen and liver. Importantly, despite this being rescued by Ripk3 or Mlkl co-deletion, this remains insufficient to impact overall survival. By contrast, our N153S model shows robust rescue by RIPK3 deletion only.

In the N153S context, the clear survival advantage, phenotypic and haematopoietic rescue provided by RIPK3 deletion strongly support a significant contribution of inflammatory necroptosis to SAVI pathogenesis. In this study we further demonstrate the specificity of necroptosis involvement by providing direct evidence for the loss of pMLKL-positive signal in *N153S; RIPK3^{-/-}* mice, the only known post-translational modification that definitively marks necroptosis execution. If the protective effect of RIPK3 deletion were limited only to its minor role in supporting apoptosis, this would not be sufficient to provide the observed rescues to N153S mice, but would indeed require the Casp8 co-deletion.

Regarding the reviewer's statement that "*Caspase-8 is still present in this context and so would be expected to act as a brake on necroptosis signaling,*" we would like to clarify that the evidence from our N153S model strongly supports the opposite. In contrast to other models where necroptosis and apoptosis are interlinked, the N153S mice show that RIPK3 deletion alone — without the need for Casp8 co-deletion, is sufficient to prevent disease pathogenesis and fully rescue survival up to at least 35 weeks of age. This is significantly different from contexts like RIPK1-deficient mice, where only combined removal of apoptosis and necroptosis pathways (e.g., co-deletion of Ripk3 or Mlkl with Casp8) can prevent lethality. In those models, the role of necroptosis becomes apparent only after Casp8 ablation, as untoward apoptosis is otherwise the dominant driver of tissue damage (notably, the intestinal homeostasis loss). The robust rescue in N153S mice with RIPK3 deficiency alone, without Casp8 inactivation, demonstrates that Caspase-8's expression in this context is functionally irrelevant for controlling necroptosis.

Mechanistically, our biochemical data directly explain this observation. We show that under STING activation via agonist treatment (mimicking auto-activation), necroptosis proceeds via a ZBP1-RIPK1-RIPK3 complex that forms independently of FADD and Caspase-8. This was demonstrated in the endogenous complex formation experiments requested by this reviewer, shown in Casp8- and FADD-deficient cells as well as in MLKL-deficient settings where Casp8 and FADD are still expressed. These data (Figure 2) prove that STING-induced chronic signalling — relevant to SAVI

— bypasses the canonical FADD–RIPK1–RIPK3 apoptotic checkpoint and its regulation by Casp8 cleavage. As a result, STING GOF mutations or chronic activation directly drive ZBP1 upregulation and the formation of this Casp8-independent necroptotic platform. This explains why, even in the presence of Caspase-8 expression, its enzymatic brake on RIPK1–RIPK3 is not engaged: the complex simply does not recruit FADD, so Casp8 is not brought into proximity to cleave RIPK1. This mechanism is consistent with recent findings showing that Caspase-8 acts as a brake only when physically recruited into the death-inducing complex and leading to the cleavage of RIPK1 (Kim Newton et al., 2024, Cell Death & Differentiation; Tran et al., 2024, Cell Death & Differentiation).

Taken together, the unique genetic rescue in our N153S mice and our mechanistic dissection of the FADD-independent ZBP1–RIPK1–RIPK3 platform provide strong evidence that STING-driven necroptosis bypasses the conventional Caspase-8 brake. This directly addresses why, despite Caspase-8 expression, the pathway remains active and pathogenic in SAVI.

Accordingly, we have added this in the discussion in lines 642-647. The text now reads: “Importantly, the robust rescue in N153S mice with RIPK3 deficiency alone, without Casp8 inactivation, demonstrates that Caspase-8’s expression in the SAVI context is functionally irrelevant for controlling necroptosis. This together with our biochemical data demonstrating the formation of a STING-induced ZBP1-RIPK1-RIPK3 complex independent of TNFR1/FADD and the observed upregulation of Z-NAs required to activate ZBP1, explain the activation of necroptosis in patients despite maintaining functional expression of Casp8.” This is also marked as per reviewer request.

Other points:

1. Lines 133-135 “treatment with the mouse STING agonist (DMXAA) or interferon-g (IFN- g) enhanced the basal level of ZBP1, which was completely abolished or significantly reduced, respectively, by co-treatment with C-178”.

I agree with C-178 abolishing DMXAA-induced ZBP1.

I do not see that it “significantly reduced” IFN γ -induced ZBP1 in Fig. 1e or 1f. Recommend re-wording.

These has been changed in the text. The sentence now reads: “ abolished or slightly reduced” Please see line 116 indicated with a comment as per review request.

2. Lines 271-272 “the transcriptional upregulation of IFN-g and IFN-b was significantly attenuated by keratinocyte-specific STING deficiency”.

They don’t cite a figure panel but presumably are referring to EDF4g. The n for each genotype is not indicated in the figure legend, but the difference in Ifnb transcripts doesn’t look that impressive. A multiple comparisons statistical test may be warranted in this setting, rather than just individual student t-tests.

Each RT-q-PCR was performed using three biological replicates. This is now contained in the legend text. For each RT-qPCR we have performed a one-way ANOVA followed by Tukey’s HSD test (FDR = 0.05) The transcriptional upregulation of IFN-g and IFN-b remains significantly attenuated by keratinocyte-specific STING deficiency also via this different statistical analysis. (Extended data Fig. 3g). Hence the statement: “Interestingly, RT-qPCR analysis revealed that transcriptional upregulation of IFN- γ and IFN- β was significantly attenuated by keratinocyte-specific STING deficiency, implying that STING in these cells drives the intrinsic cytokine expression needed for immune cell infiltration through IFN-response activation (Extended Data Fig. 3g)” (lines 220-223) remains correct. This is now marked as per reviewer request.

3. Lines 375-376 “aberrant STING activation as a driver of a transcriptional programme that includes the IFN-dependent upregulation of ZBP1 and MLKL”.

They haven't formally proven IFN-dependent upregulation of these genes in this setting. Rewording suggested.

This has now been corrected. The text reads: "aberrant STING activation as a driver of a transcriptional programme responsible for the upregulation of ZBP1 and MLKL" (Lines 302-303). This has been marked in the text as per reviewer request.

4. EDF9d – the authors should show that the cytosolic "DNA" detected by immunolabelling in Casp8 KO cells disappears if cells are treated with DNase prior to labelling.

As requested by the reviewer, we performed this experiment using Casp8 KO, Casp8/RIPK3 DKO, and Casp8/MLKL DKO cells (Rebuttal Figure 5A) treated with either DNase I (Rebuttal Figure 5B) or benzonase (Rebuttal Figure 5C) prior to staining. The results clearly demonstrate that the cytosolic DNA signal is specific to Casp8-deficient cells, as we did not detect cytosolic DNA in wild-type controls, and that treatment with DNase I or benzonase effectively abolishes the signal.

To further support the link to DNA damage, we also stained for p-H2A.X as a marker of genomic instability. White arrows in the images highlight the presence of micronucleated cells or other features indicative of genomic stress, such as DNA bridges. Notably, while all Casp8-deficient cells accumulated cytosolic DNA, not all cells were positive for p-H2A.X or displayed obvious structural features of genomic instability, suggesting that DNA damage and the resulting accumulation of cytosolic DNA in this setting are dynamic and may arise through multiple mechanisms. Together, these data confirm both the specificity of the staining and the link between Casp8 loss, genomic instability, and cytosolic DNA accumulation. While this evidence remains correlative, as it is not technically possible to selectively rescue DNA damage or genetic instability, these findings strongly support the interpretation that genomic instability (as previously published) is the most likely source of the accumulated cytosolic DNA in Casp8-deficient cells. These data have been performed to satisfy the reviewer request, however for space reasons these panels have not been added to the manuscript.

Referee #2 (Remarks to the Author):

In response to the original critiques, the authors have carried out an extensive set of experiments, including several new mouse crosses, to support the idea that chromosomal instability following loss of Casp8 triggers cGAS-STING-dependent induction of IFNs and ZBP1, and Z-NA induction, also induced by loss of Casp8, then activates ZBP1, driving necroptosis. They then extend the clinical relevance of these findings by demonstrating that some of the pathological features seen in a GOF mouse model of aberrant STING activation are rescued by loss of RIPK3. The finding that caspase8 functions beyond apoptosis in controlling the integrity of nuclear DNA has been suggested in earlier studies, including by the authors themselves. The current paper now demonstrates the importance of this 'non-canonical', and frankly unexpected, function of caspase 8 in repressing the activation of key nucleic acid sensors and the inflammation these provoke. Although the connection of Casp8 loss to SAVI (or whether aberrant casp8 function underlies other interferonopathies) remains unproven, this work provides new mechanistic insight and in vivo relevance to an interesting and unanticipated new feature of caspase 8 biology, and I am in support of publication.

I have a few additional comments:

1. The title does not properly capture what, to me at least, is the primary feature of this paper: that Casp8 loss is what instigates the entire inflammatory cascade. I recommend: "Caspase 8 prevents STING-dependent activation of ZBP1", or similar. Invoking TNFR1 independence in the title is not necessary, as ZBP1 activation in most scenarios is not typically considered TNF dependent in the first place.

Please see response below the next comment.

4. Similarly, key data panels from the RIPK3 rescue of the STING GOF mouse model (i.e., flow cytometric data from S14f,h,i,k) could be included in Fig 6. Is dermatitis in the STING GOF mice rescued by RIPK3 loss? I could not find any allusion to this in the revised paper or rebuttal. Whatever the outcome of these analyses, the authors should make a mention of what is seen in the skin of the STING GOF x RIPK3 KO mice.

We appreciate the reviewer's thoughtful feedback on our work and on both the title and the need for explicit mention of the skin phenotype in the STING GOF / RIPK3 KO mice. We agree that these two points are very important and we believe to be closely related, as they both address the core mechanism driving pathology in the STING GOF model and its genetic dissection.

To address these points directly: we now provide new data clarifying the extent of RIPK3-dependent rescue in the N153S STING GOF model, which directly informs the discussion of Casp8's functional role. Specifically, we have included the survival analysis of N153S; Ripk3^{-/-} mice, which survive without any overt signs of disease and reach at least 35 weeks of age, matching the median survival of N153S controls (Rebuttal Figure 4A,B, now included in Fig. 6d, e). Additionally, we have added the requested pMLKL-S345 IHC staining for skin, lung, and thymus in N153S; Ripk3^{-/-} mice, which show a clear loss of pMLKL-positive staining compared to N153S controls (Rebuttal Figure 4C, now included in Fig. 6f). These results confirm that RIPK3 deletion fully rescues the skin lesion phenotype (Rebuttal Figure 4B, Fig. 6e), directly addressing the reviewer's question about whether dermatitis is rescued. In the same panel, the spleen is also shown, which displays restored architecture but no pMLKL positivity, suggesting that the improved splenic architecture is likely due to the early haematopoietic rescue observed in N153S; RIPK3^{-/-} mice. Together, these data demonstrate the specificity of the IHC signal and the contribution of RIPK3-dependent necroptosis to tissue inflammation, including in the skin.

These data are now added in the text in lines 555-561.

Rebuttal Figure 4: Kaplan–Meier survival curves (A) and representative images (B) of mice of the indicated genotypes. (C) Representative images of consecutive skin, lung, spleen and thymus sections from *Sting*^{N153S/wt} and *Sting*^{N153S/wt}; *Ripk3*^{KO} at survival endpoint and age-matched *Ripk3*^{KO} littermate controls stained with H&E and pMLKL-S345 (n = 3 in each group). Scale bars: 100 μ m (representative field) and 20 μ m (magnified selected area).

Regarding the title, we respectfully maintain that the independence of the STING–ZBP1 necroptosis axis from TNFR1/FADD is a key mechanistic insight that justifies its inclusion. Unlike other models where necroptosis and apoptosis are interlinked, the N153S mice show that RIPK3 deletion **alone**, without Casp8 co-deletion, is sufficient to prevent disease pathogenesis and fully rescue survival up to at least 35 weeks of age. This is strikingly different from contexts like RIPK1-deficient mice, where only combined removal of apoptosis and necroptosis pathways (e.g., co-deletion of *Ripk3* or *Mlkl* with Casp8) can prevent lethality. In those models, the role of necroptosis becomes apparent only after Casp8 ablation, as untoward apoptosis is otherwise the dominant driver of tissue damage (notably, the intestinal homeostasis loss). By contrast, the robust rescue seen in N153S mice with RIPK3 deficiency **alone**, without Casp8 inactivation, demonstrates that Caspase-8 expression/activation in this context is functionally irrelevant for restraining necroptosis.

Mechanistically, our biochemical data directly explain this observation. We show that under STING activation (mimicking chronic auto-activation), necroptosis proceeds via a ZBP1–RIPK1–RIPK3 complex that forms independently of FADD and Caspase-8. This was demonstrated in the endogenous complex formation experiments shown in main Fig. 2, where we show the ZBP1–RIPK1–RIPK3 complex not only in Casp8- and FADD-deficient cells, but also in MLKL-deficient cells where Casp8 and FADD are still expressed. Yet, despite endogenous expression remains untouched, FADD remains absent from the ZBP1 pulldown, confirming that this platform forms independently of endogenous Casp8, which can only be recruited by FADD. These data demonstrate that STING-induced chronic signalling, relevant to SAVI, bypasses the canonical FADD–RIPK1–RIPK3 apoptotic checkpoint and its regulation by Casp8 cleavage. As a result, STING GOF mutations or sustained activation directly drive ZBP1 upregulation and the formation of this Casp8-independent necroptotic platform. This explains why, even in the presence of Caspase-8 expression, its enzymatic brake on RIPK1–RIPK3 is not engaged: the complex simply does not recruit FADD, so Caspase-8 is not brought into proximity to cleave RIPK1. In our murine model, Casp8 deficiency is necessary only to licenses STING activation by enabling cytosolic DNA accumulation necessary for cGAS/STING engagement.

Taken together, the unique genetic rescue in our N153S mice and our mechanistic dissection of the FADD-independent ZBP1–RIPK1–RIPK3 axis provide strong evidence and explanation that STING-driven necroptosis bypasses the conventional Caspase-8 brake and proceeds independently of TNFR1/FADD which we hope we have clarified and justified sufficiently to keep in the title. This directly explains why, despite Caspase-8 expression, the pathway remains active and pathogenic in SAVI, and why RIPK3 deficiency alone is sufficient to achieve robust rescue in the N153S model.

Accordingly, we have added this in the discussion in lines 642-647. The text now reads: “Importantly, the robust rescue in N153S mice with RIPK3 deficiency alone, without Casp8 inactivation, demonstrates that Caspase-8’s expression in the SAVI context is functionally irrelevant for controlling necroptosis. This together with our biochemical data demonstrating the formation of a STING-induced ZBP1-RIPK1-RIPK3 complex independent of TNFR1/FADD and the observed upregulation of Z-NAs required to activate ZBP1, explain the activation of necroptosis in patients despite maintaining functional expression of Casp8.”.

2. The Abstract should specifically mention that caspase 8 loss also triggers Z-NA accrual, which drives ZBP1 activation. This extra text can come at the expense of some of the SAVI writeup, which currently makes up about half the abstract.

This has been revised as requested.

3. The key mechanistic figures linking caspase 8 loss to cGAS activation, and to ZBP1 activation are, respectively, the panels showing genomic instability in Casp8 KO skin (Fig S9i) and the convincing new Z-NA staining (Fig S10). I recommend including these panels in the main Figures, many of which are currently given over to histological images better suited for Supplemental. These have now been added to Fig.5

Referee #3 (Remarks to the Author):

Kelepouras et al. submitted a fully revised manuscript establishing that the ZBP1-RIPK3-MLKL pathway depends on STING-driven activation of ZBP1 as a central inflammatory mechanism in both Casp8-deficient inflammation and STING-Associated Vasculopathy with onset in Infancy in humans. This work brings together a body of work on STING regulation of interferon activation and cell death induction with growing evidence that ZBP1-RIPK3 signal transduction is a critical determinant of inflammatory outcomes in mice and humans.

The authors have provided extensive responses to all reviewers concerns accompanied by significant additional data and evidence. All coming forward as a stellar piece of scholarship.

We would like to thank this reviewer for the comment and for acknowledging the effort and quality of our work. We hope this reviewer will find that the new additions requested by the other two reviewers, further strengthen our arguments and the overall impact of this work.